# Efficient First-Order Optimization on the Pareto Set for Multi-Objective Learning under Preference Guidance

**Lisha Chen** [1]   **Quan Xiao** [1]   **Ellen Hidemi Fukuda** [2]   **Xinyi Chen** [3]   **Kun Yuan** [3]   **Tianyi Chen** [1]

## Abstract

Multi-objective learning under user-specified preference is common in real-world problems such as multi-lingual speech recognition under fairness. In this work, we frame such a problem as a semivectorial bilevel optimization problem, whose goal is to optimize a pre-defined preference function, subject to the constraint that the model parameters are weakly Pareto optimal. To solve this problem, we convert the multi-objective constraints to a single-objective constraint through a merit function with an easy-to-evaluate gradient, and then, we use a penalty-based reformulation of the bilevel optimization problem. We theoretically establish the properties of the merit function, and the relations of solutions for the penalty reformulation and the constrained formulation. Then we propose algorithms to solve the reformulated single-level problem, and establish its convergence guarantees. We test the method on various synthetic and real-world problems. The results demonstrate the effectiveness of the proposed method in finding preference-guided optimal solutions to the multi-objective problem.

## 1. Introduction

Many machine learning tasks naturally involve multiple objectives, which may include diverse performance metrics such as accuracy, fairness, and privacy, or even the same metrics evaluated across different datasets (Sener & Koltun, 2018). A common approach to tackling such multi-objective problems is to learn a shared model that performs well across all objectives simultaneously. Compared to training separate models for each objective, this approach offers

[1]Rensselaer Polytechnic Institute, Troy, United States [2]Kyoto University, Kyoto, Japan [3]Peking University, Beijing, China. Correspondence to: Lisha Chen <lishachen9577@gmail.com>, Tianyi Chen <chentianyi19@gmail.com>.

*Proceedings of the 42$^{nd}$ International Conference on Machine Learning*, Vancouver, Canada. PMLR 267, 2025. Copyright 2025 by the author(s).

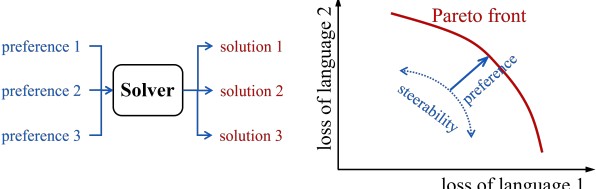

*Figure 1.* Example on multi-lingual speech or language processing problem under user-specified preference guidance. The red curve represents the Pareto front, which is, informally, the set of objective values that achieve the best trade-offs among multiple objectives.

significant benefits – most notably, it reduces model size and inference time, making it more efficient and scalable. Multi-objective optimization facilitates this by enabling the learning of models that minimize vector-valued objectives (Miettinen, 1998; Ehrgott, 2005). In practical scenarios, it is often desirable to obtain solutions that provide controlled trade-offs or reflect specific preferences among competing objectives, rather than treating all objectives equally.

To further illustrate, we use one example on multi-lingual speech or language processing problem in Figure 1. The goal of this problem is to minimize multiple losses from different languages, while satisfying the user-specified preferences. Preferences can control trade-offs among multiple losses and enhance steerability, enabling the solver to return diverse solutions on the Pareto front. Analytically, the preferences can be defined as constraints or objectives, see, e.g., (Lin et al., 2019; Curtis et al., 2023; Chen et al., 2024a). To prioritize finding the optimal solutions of the multi-lingual losses over satisfying the preferences, we model the preference as a secondary scalar-valued objective. We first optimize the vector-valued objective formed by concatenating the multi-lingual losses, and then optimize the scalar-valued preference objective. More formally, let $f_1, \ldots, f_M : \mathcal{X} \to \mathbb{R}$ be the objective functions, and $f_0 : \mathcal{X} \to \mathbb{R}$ be the preference function, e.g., the discrepancy between different objectives, with $\mathcal{X} \subseteq \mathbb{R}^q$ being nonempty, closed and convex. Depending on the problem, $\mathcal{X}$ can be compact or $\mathbb{R}^q$. Let $F = (f_1, \ldots, f_M) : \mathcal{X} \to \mathbb{R}^M$ be a vector-valued function. Then the optimization problem is

$$\min_{x \in \mathcal{X}} f_0(x) \ \text{s.t.} \ \ x \in \arg\min_{x \in \mathcal{X}} \ F(x) \qquad \text{(OPS)}$$

where for minimizing the vector-valued objective $F(x)$, we consider the widely used Pareto optimality, whose formal definition is deferred to Section 2. Then problem (OPS) is also known as Optimization on the Pareto Set (OPS), or a semivectorial simple bilevel optimization problem (Bolintineanu, 1993a;b).

The OPS or semivectorial bilevel optimization problem is generally very difficult to solve (Bolintineanu, 1993b). Existing studies usually make strong assumptions such as the convexity of the objectives $F$ (Roy et al., 2023), or the algorithms require evaluating the second-order derivatives of the objectives, and is thus inefficient for large-scale problems (Chen et al., 2022; Roy et al., 2023). To address these challenges, we introduce reformulations of the OPS problem, establish relations of the reformulations and the original problem, and propose algorithms to solve the reformulated problem with convergence guarantees.

Our contributions can be summarized as follows:
First, we propose a smoothed merit function to convert the vector-valued objective $F$ to a scalar-valued objective, with easy-to-evaluate gradient. We prove that the smoothed merit function preserves the equivalence with approximate weak Pareto optimality. Then we use the smoothed merit function as a penalty function, prove its error bound properties, and establish the relation of the global/local/stationary solutions of the penalty-based reformulation and the original problem. Based on the reformulation, we propose an efficient first-order algorithm with convergence rate guarantees. Experiments are conducted on various synthetic and real datasets with possibly nonconvex objectives to demonstrate its effectiveness.

Technically, we address the following challenges:

T1 The original OPS problem in general has a non-convex non-smooth structure, posing challenges to evaluating the subdifferential and developing convergent algorithms. We propose to use a smoothed merit function as a penalty so that it has easy-to-evaluate gradient, and the convergence of the developed algorithm can be analyzed.

T2 Instead of directly assuming the lower-level merit (penalty) function satisfies the error bound as in existing (simple) bilevel optimization literature, we prove the proposed merit (penalty) function satisfies the desired error bound when the objectives satisfy certain conditions such as subanalyticity.

T3 We define a stationary condition for the simple bilevel problem with provable calmness condition under the Kurdyka-Łojasiewicz inequality, weaker than the assumptions in existing results, and thus applicable to a wider range of problems including ours. Based on this, we establish the relation of the stationary solution

to the penalty problem and that to the simple bilevel problem.

## 2. Problem Setup and Preliminaries

For the optimization problem $\min_{x \in \mathcal{X}} F(x)$ in (OPS), we use the standard definitions for Pareto optimality (Miettinen, 1998). Given two vectors $v$ and $w$, we use $v < w$ and $v \leq w$ to denote $v_i < w_i$ and $v_i \leq w_i$ for all $i$, respectively. We use $v \lneq w$ to denote $v \leq w$ and $v \neq w$, and define $>, \geq, \gneq$ analogously. We use $\mathbf{1}_M$ to denote an all-one vector with dimension $M$, where $M$ is sometimes ommitted if it is clear from the context. Then Pareto dominance and weak Pareto optimality are formally defined below.

**Definition 2.1** (Pareto dominance and optimality)**.** Given $v, w \in \mathbb{R}^M$, we say $v$ strictly dominates $w$ if and only if $v - w < 0$. Correspondingly, a point $x \in \mathcal{X}$ is *weakly Pareto optimal* if there is no $x' \in \mathcal{X}$ such that, $F(x') < F(x)$. In addition, a point $x \in \mathcal{X}$ is $\epsilon$-*weakly Pareto optimal* if there exists no $x' \in \mathcal{X}$ and $x' \neq x$ such that, $F(x') < F(x) - \epsilon\mathbf{1}$.

Throughout the paper, we assume $f_m(x), m = 0, \ldots, M$ are proper and bounded below. And we assume they are twice continuously and directionally differentiable. Denote the directional derivative of $f_m$ at point $x$ along direction $d$ as $f'_m(x; d)$, defined as

$$f'_m(x; d) := \lim_{\alpha \downarrow 0} \frac{f_m(x + \alpha d) - f_m(x)}{\alpha}. \qquad (1)$$

Then the Pareto stationarity is defined as follows.

**Definition 2.2** (Pareto stationarity e.g. (Ehrgott, 2005))**.** A point $x \in \mathcal{X}$ is Pareto stationary if $\max_{m \in [M]} f'_m(x; z - x) \geq 0$ for all $z \in \mathcal{X}$, where $[M] = \{1, \ldots, M\}$.

Denote $WP(F) \subseteq \mathcal{X}$ as the weak Pareto set of $F$, which contains all the weakly Pareto optimal solutions for $\min_{x \in \mathcal{X}} F(x)$. Then problem (OPS) is equivalent to $\min_{x \in WP(F)} f_0(x)$. We say that the solutions to (OPS) are *preferred Pareto optimal*. We then discuss in the next section the problem formulation to find these solutions.

## 3. Problem Reformulation

In this section, we first convert the original problem (OPS) with multiple lower-level objectives to an equivalent problem with a single lower-level objective using a merit function. Then we discuss its penalty-based reformulation.

### 3.1. A smoothed merit function and its properties

A merit function associated with the multi-objective optimization problem $\min_{x \in \mathcal{X}} F(x)$ is non-negative, and returns zero only at the weakly Pareto optimal solutions (Auslender, 1976; Hearn, 1982). Under this require-

ment, assuming lower semicontinuity of $F$, then

$$\bar{u}(x) := \sup_{y \in \mathcal{X}} \min_{m \in [M]} \{f_m(x) - f_m(y)\} \qquad (2)$$

is a merit function in the sense of weak Pareto optimality (Tanabe et al., 2024, Theorem 3.1). In other words, $\bar{u}(x) = 0$ if and only if $x$ is weakly Pareto optimal. Given this equivalence, it is desirable to convert the original lower-level multi-objective optimization problem to minimizing the scalar-valued function $\bar{u}(x)$. However, $\bar{u}(x)$ in general can be non-differentiable due to its max-min structure, posing challenges to directly applying gradient-based approaches to minimize $\bar{u}(x)$. To address this challenge, we propose the following smoothed and regularized merit function $v_{l,\tau}(x)$ given $l \geq 0, \tau > 0$.

$$h_{l,\tau}(x, y) := \tau \ln \Big( \sum_{m=1}^{M} e^{\frac{f_m(y) - f_m(x)}{\tau}} \Big) + \frac{l}{2} \|x - y\|^2 \quad (3a)$$

$$v_{l,\tau}(x) := -\min_{y \in \mathcal{X}} h_{l,\tau}(x, y). \qquad (3b)$$

Note that, $v_{l,\tau}$ can be seen as a smoothed and regularized function of $\bar{u}$. Specifically, when $l = 0$, $v_{0,\tau}$ smoothes the maximization operation over $m \in [M]$ in $\bar{u}$ with the log-sum-exponential (LSE) function (Nesterov, 2005). And it uniformly converges to $\bar{u}$ as $\tau$ converges to zero. Besides, adding the regularization with $l > 0$ can further lift a weakly convex objective to a strongly convex one, so that not only the minimization $\min_{y \in \mathcal{X}} h_{l,\tau}(x, y)$ in (3b) enjoys a unique solution, but also $v_{l,\tau}(x)$ is smooth and has easy-to-evaluate gradient. To further illustrate, we provide a visualization of $\bar{u}, v_{l,\tau}$ on a simple example in Figure 2. It shows that $v_{l,\tau}$ with smaller $\tau$ or $l$ approximates $\bar{u}$ better, while larger $\tau$ or $l$ makes $v_{l,\tau}$ smoother. Too large $\tau$ or $l$ could possibly change the shape of $v_{l,\tau}$ significantly compared to $\bar{u}$.

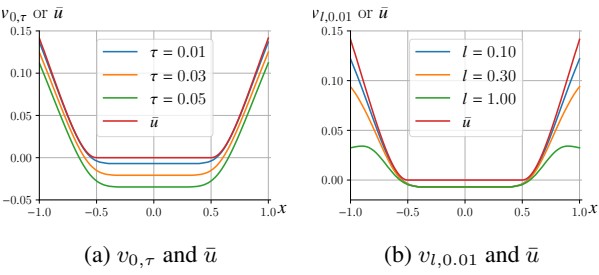

(a) $v_{0,\tau}$ and $\bar{u}$        (b) $v_{l,0.01}$ and $\bar{u}$

*Figure 2.* Illustration of $v_{l,\tau}$ and $\bar{u}$ with different values of $\tau$ and $l$ with $F(x) = \Big( \sqrt[6]{(x + \frac{1}{2})^2 + \frac{1}{8}}, \sqrt[6]{(x - \frac{1}{2})^2 + \frac{1}{8}} \Big)^\top$.

Then we discuss the properties of the smoothed merit function and outline the procedure for computing its gradient. **Properties of the smoothed merit function.** We establish how the value of the smoothed merit function changes when $x$ reaches the weak Pareto optimality condition. In the prior work (Tanabe et al., 2024), such properties have

been established for the merit functions $\bar{u}$ and $u_l$ (c.f. Appendix B.1). We will show that the smoothed version $v_{l,\tau}$ can still preserve these properties approximately depending on the hyperparameters $l$ and $\tau$. For our analysis, we make the following assumptions that are common in multi-objective optimization (Fliege et al., 2019; Liu & Vicente, 2021). Note that we do not require all the following assumptions to hold for all our results, which will be specified correspondingly.

**Assumption 1.** For all $m \in \{0, \dots, M\}$, $f_m(x)$ is locally Lipschitz on any bounded set in $\mathcal{X}$.

**Definition 3.1** (Weak convexity). A locally Lipschitz function $f : \mathcal{X} \to \mathbb{R}$ is $\mu$-weakly convex if $f(x) - \frac{\mu}{2}\|x\|^2$ is convex for $x \in \mathcal{X}$.

**Assumption 2.** For all $m \in [M]$, $f_m(x)$ is locally Lipschitz and $\mu$-weakly convex on $\mathcal{X}$.

Before proceeding to the theoretical results, we introduce the following definition, which relaxes the commonly used convexity assumption of the objective functions.

**Definition 3.2** (Point strong quasar-convex functions (Hardt et al., 2018, Definition 2.1)). A function $f : \mathcal{X} \to \mathbb{R}$ is $(c_q, \mu)$-point strong quasar-convex with $c_q \in (0, 1], \mu \geq 0$ at $x^* \in \mathcal{X}$ if for all $x \in \mathcal{X}$,

$$f(x^*) \geq f(x) + \frac{1}{c_q}\nabla f(x)^\top (x^* - x) + \frac{\mu}{2}\|x^* - x\|^2. \quad (4)$$

The point (strong) quasar-convexity in Definition 3.2 is a relaxation of the (strong) convexity. When $c_q = 1$, the quasar-convexity implies point star-convexity (Lee & Valiant, 2016). And if the point star-convexity holds at all $x \in \mathcal{X}$, then it implies convexity. There are many examples of nonconvex but point quasar-convex functions, see the discussions in, e.g., (Hinder et al., 2020). In machine learning, typical examples that satisfy the quasar convexity include the linear dynamical systems identification (Hardt et al., 2018) and generalized linear models with leaky ReLU or logistic activation functions (Wang & Wibisono, 2023).

Based on the above assumptions and definition, we introduce the properties of the smoothed merit function $v_{l,\tau}$ in Proposition 3.3.

**Proposition 3.3** (Properties of $v_{l,\tau}$). *Suppose Assumption 2 holds. The merit function $v_{l,\tau}(x)$ defined in* (3b) *satisfies the following properties:*
*1. $\bar{u}(x) - \tau \ln M \leq v_{0,\tau}(x) \leq \bar{u}(x)$. Furthermore, $\min_{x \in \mathcal{X}} v_{l,\tau}(x) = -\tau \ln M$.*
*2. If $x$ is weakly Pareto optimal, then $v_{l,\tau}(x) \leq 0$. Conversely, if a) $l = 0$, $v_{l,\tau}(x) \leq 0$, then $x$ is $\epsilon$-weakly Pareto optimal with $\epsilon = \tau \ln M$; b) $l > 0$, $v_{l,\tau}(x) \leq -\tau \ln M$, and for all $m \in [M]$, $f_m$ are $(1, 0)$-point quasar-convex at $x$, then $x$ is weakly Pareto optimal.*

The proof of Proposition 3.3 is deferred to Appendix B.3. In Appendix B.4, we provide some examples of nonconvex $F$ that satisfies condition 2-b) in Proposition 3.3.

Next we discuss how to compute the gradient of the smoothed merit function $v_{l,\tau}$, so that we can use gradient-based methods to directly minimize $v_{l,\tau}$.

**Gradient of the smoothed merit function.** Under Assumption 2, all the objectives $f_m$ are $\mu$-weakly convex, i.e., $f_m(x) - \frac{\mu}{2}\|x\|^2$ is convex on $\mathcal{X}$. The LSE function preserves weak convexity (c.f. Lemma B.3), thus $h_{l,\tau}(x,y)$ is strongly convex w.r.t. $y$ if $l + \mu > 0$. Then $y_{l,\tau}^*(x) := \arg\min_{y \in \mathcal{X}} h_{l,\tau}(x,y)$ is a singleton, and is continuous w.r.t. $x$, so the Danskin-type theorem can be applied here to compute the gradient of $v_{l,\tau}$, given by

$$\nabla v_{l,\tau}(x) = \sum_{m=1}^{M} \pi_m(x)\nabla f_m(x) - l(x - y_{l,\tau}^*(x)), \quad \text{(5a)}$$

$$\text{with} \quad \pi_m(x) := \frac{e^{\frac{1}{\tau}(f_m(y_{l,\tau}^*(x)) - f_m(x))}}{\sum_{m=1}^{M} e^{\frac{1}{\tau}(f_m(y_{l,\tau}^*(x)) - f_m(x))}}. \quad \text{(5b)}$$

**Reformulation of problem** (OPS). We then consider the following optimization problem by approximating the Pareto set constraint $x \in WP(F)$ through a merit function constraint using $v_{l,\tau}$.

$$\min_{x \in \mathcal{X}} f_0(x), \text{ s.t. } x \in \mathcal{X}_{v_{l,\tau}}^* := \{x \in \mathcal{X} \mid v_{l,\tau}(x) + \tau \ln M \le 0\}. \quad \text{(CP)}$$

We name the above program a constrained program (CP) reformulation. By Proposition 3.3, $S(F) \subseteq WP(F)$, and these two sets become equal as $l, \tau \downarrow 0$. To solve (CP), we further consider a penalty-based program (PP$_\gamma$) with penalty parameter $\gamma, \theta > 0$ below

$$\min_{x \in \mathcal{X}} \varphi_\gamma(x) := f_0(x) + \gamma p(x) \quad \text{(PP}_\gamma\text{)}$$

$$\text{with} \quad p(x) := (v_{l,\tau}(x) + \tau \ln M)^\theta.$$

For (PP$_\gamma$), when $\gamma \to \infty$, any limit point of the sequence of solutions to the approximation problem (PP$_\gamma$) is a solution to the problem (CP).

## 3.2. Relation of different formulations

To establish the relations of the solutions to (PP$_\gamma$), (CP), and (OPS), without loss of generality, we assume there exists at least one $x^* \in \arg\min_{x \in \mathcal{X}} v_{l,\tau}(x)$ that $x^*$ is bounded, and that the function value $f_m$ and gradient $\nabla f_m$ at $x^*$ for $m = 0, \dots, M$ are also bounded. We also introduce the following global subanalyticity assumptions on the objectives below.

**Assumption 3** (Subanalyticity of $f_m(x)$). For all $m \in [M]$, $f_m(x)$ is subanalytic on $\mathcal{X}$.

Due to space limit, the definitions on (global) subanalytic functions and the related properties are provided in Appendix C. Subanalyticity can be generally satisfied by many

widely-used objective functions (Dries & Miller, 1996; Bolte et al., 2007). For example, the $\ell_p$-norm with $p \ge 1$, and the LSE and polynomial functions defined on a bounded set, all satisfy the subanalyticity. More discussions and examples are provided in Appendix C.2 and Table 6. Intuitively speaking, global subanalytic functions can be described by finite combinations of locally analytic functions. They exhibit a "tame" geometry, thus stability under basic operations, and desirable properties for optimization. One of them is the Hölderian error bound defined below.

**Definition 3.4** (($\varrho, \eta$)-Hölderian error bound). For a function $v: \mathcal{X} \to \mathbb{R}$, let $\mathcal{X}_v^* := \arg\min_{x \in \mathcal{X}} v(x)$. Then $v$ satisfies the ($\varrho, \eta$)-Hölderian error bound (HEB) if

$$\varrho\big(v(x) - \min_{x \in \mathcal{X}} v(x)\big) \ge \big(\text{dist}(x, \mathcal{X}_v^*)\big)^\eta \quad \text{(6)}$$

where $\text{dist}(x, S)$ is the Euclidian distance from a point $x$ to a set $S$, and $\varrho, \eta > 0$.

The HEB in Definition 3.4 generalizes the widely used Quadratic Growth (QG) condition with $\eta = 2$ in optimization (Karimi et al., 2016; Drusvyatskiy & Lewis, 2018), and the weak sharp minima condition with $\eta = 1$ (Burke & Ferris, 1993). This condition ensures the point is close to the solution set if the function value gap at the point is small. In our problem, it is desirable that the function $v_{l,\tau}$ also satisfies such a condition, so that it satisfies HEB near its solution set. This can be proved based on the properties of subanalytic functions, as described in Lemma 3.5 below.

**Lemma 3.5** (Subanalyticity of $\mathcal{X}_{v_{l,\tau}}^*$ and $v_{l,\tau}(x)$). *Under Assumption 3, and that $\mathcal{X}$ is subanalytic, given a compact subanalytic set $\mathcal{X}_C$, suppose $f_m(x)$ is continuous and bounded on $\mathcal{X}_C \cap \mathcal{X}$ for all $m \in [M]$. Then both $\mathcal{X}_{v_{l,\tau}}^* \cap \mathcal{X}_C$ and $v_{l,\tau}(x)$ on $\mathcal{X}_C \cap \mathcal{X}$ are globally subanalytic. Consequently, $v_{l,\tau}(x), p(x)$ satisfy the $(\varrho, \eta)$ and $(\varrho_p, \eta_p)$-HEB in Definition 3.4 on $\mathcal{X}_C \cap \mathcal{X}$, respectively, with some $\varrho, \eta > 0$, $\eta_p = \theta\eta$, and $\varrho_p = \varrho^\theta$.*

Definition 3.4 and Lemma 3.5 are crucial for establishing the relations of the global/local/stationary solutions of the penalty reformulation (PP$_\gamma$) and the constrained formulation (CP). Below we first define the global and local solutions to (CP), then we discuss their relation with global and local solutions to (PP$_\gamma$).

**Definition 3.6** (Global and local solutions). We say $x$ is an $(\epsilon, \delta)$-global solution to (CP) on $\mathcal{X}_S \subseteq \mathcal{X}$ if it satisfies

$$f_0(x) - \min_{x \in \mathcal{X}_S \cap \mathcal{X}_\delta} f_0(x) \le \epsilon, \quad x \in \mathcal{X}_S \cap \mathcal{X}_\delta, \quad \text{(7a)}$$

$$\text{with} \quad \mathcal{X}_\delta := \{x \in \mathcal{X} \mid p(x) \le \delta\}. \quad \text{(7b)}$$

Let $\mathbb{B}(x, r)$ denote the neighborhood of $x$ with radius $r$. We say $x$ is an $(\epsilon, \delta)$-local solution to (CP) on $\mathcal{X}$ if it is an $(\epsilon, \delta)$-global solution on $\mathcal{X}_S = \mathbb{B}(x, r) \cap \mathcal{X}$.

In Theorem 3.7, we establish the relation of solutions to (CP) with a smoothed merit function and those to the original problem (OPS). We also establish the relations of the global/local solutions of the penalty reformulation (PP$_\gamma$) and the constrained formulation (CP).

> **Theorem 3.7** (Relation of $\epsilon$-global/local solutions to (OPS), (CP) and (PP$_\gamma$)). *Suppose Assumptions 1 and 3 hold. Then with proper choices of $\epsilon'$, $\epsilon$, $\delta'$, $\gamma$ depending on $\delta$ for all $\delta > 0$, we have*
> *1. (Relation of (OPS) and (CP)) The $(\epsilon, \delta)$-global/local solutions to (CP) with $\delta \geq \tau \ln M$ are $(\epsilon', \delta)$-global/local solutions to (OPS). Conversely, the $(\epsilon, \delta)$-global/local solutions to (OPS) are $(\epsilon', \delta')$-global/local solutions to (CP).*
> *2. (Relation of (CP) and (PP$_\gamma$)) The $\epsilon$-global/local solution to (PP$_\gamma$), denoted as $x_\gamma$, is an $(\epsilon, \delta + 2\epsilon^*)$-global/local solution to (CP), where $\epsilon^* = 0$ for the global case, and $\epsilon^* = \inf_{x \in \mathbb{B}(x_\gamma, r) \cap \mathcal{X}} p(x)$ for the local case. Conversely, an $(\epsilon', \epsilon)$-global/local solution to (CP) is a $\delta$-global/local solution to (PP$_\gamma$).*

The proof of Theorem 3.7-2 is deferred to Appendix D.2. It extends the result from (Shen et al., 2025) using the HEB proved in Lemma 3.5 on a compact subanalytic set instead of directly assuming the QG condition on the whole domain $\mathcal{X}$. Furthermore, it does not rely on convexity assumptions of $v_{l,\tau}(x)$. The less restrictive assumptions make it applicable to a much wider set of problems. A more detailed comparison is given in Appendix A.1.

Theorem 3.7-2 states that if $v_{l,\tau}$ satisfies HEB, and there exists $p(x) \leq \epsilon^*$ in the neighborhood of the local solution, then the local solution of (PP$_\gamma$) is a local solution of (CP). In e.g., (Shen et al., 2025; Chen et al., 2024b), these conditions can be satisfied under certain assumptions on the lower-level objective, such as the PL inequality or convexity. However, in our problems (PP$_\gamma$) and (CP), we cannot directly assume such conditions hold for $v_{l,\tau}$. Therefore, next we will show that the conditions hold under additional conditions specified in Proposition 3.9. We first introduce the Kurdyka-Łojasiewicz inequality below.

**Definition 3.8** (Kurdyka-Łojasiewicz inequality). A proper and lower semicontinuous function $f : \mathbb{R}^q \to (-\infty, +\infty]$ satisfies the $(c, \alpha)$-Kurdyka-Łojasiewicz (KL) inequality at $\bar{x}$ if there exist $\nu \in (0, +\infty]$, $c > 0$, $\alpha > 1$, a neighborhood $\mathbb{B}(\bar{x})$, such that for all $x \in \mathbb{B}(\bar{x})$ and $f(\bar{x}) < f(x) < f(\bar{x}) + \nu$, the following inequality holds

$$c\big(\text{dist}(0, \partial f(x))\big)^\alpha \geq \|f(x) - f(\bar{x})\|. \tag{8}$$

Moreover, if $f$ satisfies the $(c, \alpha)$-KL inequality for every pair of points $(x, \bar{x})$ on a set $\mathcal{X}_C$ with $f(\bar{x}) = \min_{x \in \mathcal{X}_C} f(x)$, then we say $f$ is $(c, \alpha)$-KL on $\mathcal{X}_C$.

**Proposition 3.9.** *Let $x \in \mathcal{X}$ be a bounded $\epsilon$-stationary point of $\min_{x \in \mathcal{X}} v_{l,\tau}(x)$. If there exists $x^* \in \mathcal{X}_{v_{l,\tau}}^*$ and*

*$x \in \mathbb{B}(x^*)$ with KL inequality at $x^*$, then $v_{l,\tau}$ satisfies the condition in Theorem 3.7-2. The above condition holds if a) for all $m \in [M]$, $f_m$ satisfies the $(1, 0)$-point quasar-convexity at $x$, and $(1, \mu)$-point strong quasar-convexity at $y_{l,\tau}^*(x) = \arg \min_{y \in \mathcal{X}} h_{l,\tau}(x, y)$; or b) $v_{l,\tau}(x) + \tau \ln M \leq \nu$ in Lemma C.9.*

The condition in Theorem 3.7-2 requires a local solution to $\min_{x \in \mathcal{X}} v_{l,\tau}(x)$ is also $\epsilon$-globally optimal to this problem. Such a condition holds if $F$ satisfies point convexity or subanalyticity near the stationary points of $v_{l,\tau}(x)$, as shown in Proposition 3.9. The proof is deferred to Appendix D.2. Next, we define the KKT condition, its necessity, and establish the relation of stationary solutions.

**Definition 3.10** ($(\epsilon, \delta)$-stationary condition, e.g. (Liu et al., 2022; Xiao et al., 2023b)). Let $\mathcal{X} = \mathbb{R}^q$. For the function $p$ such that $\nabla p(x)$ and $\nabla^2 p(x)$ exist, and $\nabla p(x) = 0$ implies $p(x) = 0$, a gradient-based reformulation of the problem (CP) is

$$\min_{x \in \mathbb{R}^q} f_0(x), \text{ s.t. } \nabla p(x) = 0, \tag{9}$$

The $(\epsilon, \delta)$-KKT condition of the problem (9) is

$$\|\nabla f_0(x) + \nabla^2 p(x) w\| \leq \epsilon, \ \|\nabla p(x)\| \leq \delta \tag{10}$$

where $w \in \mathbb{R}^q$ is bounded.

> **Theorem 3.11** (Relation of $\epsilon$-stationary solutions). *Let $\mathcal{X} = \mathbb{R}^q$, and $\theta = 1$. Let $x_\gamma$ be a bounded $\epsilon$-stationary solution to (PP$_\gamma$). Then there exists a compact subanalytic set $\mathcal{X}_C \subset \mathbb{R}^q$ with $\mathcal{X}_C \cap \mathcal{X}_{v_{l,\tau}}^* \neq \emptyset$ and $x_\gamma \in \mathcal{X}_C$. Suppose Assumption 1 holds, and that on $\mathcal{X}_C$, $\nabla v_{l,\tau}(x)$ exists and is $\ell_{v,2}$-smooth, and $v_{l,\tau}(x)$ is $(c_v, \alpha_v)$-KL with $\alpha_v \geq 2$. Then with proper choice of parameter $\gamma$, $\epsilon$ depending on $\delta$, $x_\gamma$ is an $(\epsilon + \delta, \delta)$-KKT point to the problem (9).*

The $\ell_{v,2}$-smoothness of $\nabla v_{l,\tau}$ can be justified under additional assumptions of the Hessian of $f_m$ for $m \in [M]$. See a detailed discussion in Lemma D.13. Theorem 3.11 shows that for a simple bilevel problem where the lower-level objective $v_{l,\tau}(x)$ satisfies the KL inequality with exponent $\alpha_v \geq 2$, a stationary solution to the penalty reformulation approximates the KKT solution to the constrained formulation in (9). This indicates that though the KKT solution to a bilevel problem often requires the second-order information of the lower-level objective as shown in (10) and discussed in e.g., (Roy et al., 2023; Liu et al., 2022; Xiao et al., 2023b), one can still use first-order methods to approximate such solutions. The proof of Theorem 3.11 is provided in Appendix D.3. We first prove that the calmness condition holds in the above settings, ensuring the KKT conditions are necessary for global optimality.

## 3.3. Comparison with existing methods

To address the preference-guided MOL problem, one commonly used formulation is linear scalarization (LS) where

*Table 1.* A recipe to choose hyperparameters $\theta, \gamma$ to obtain solutions to (PP$_\gamma$) and thus the solutions to (CP). Denote $\eta_p, \alpha_p$ as the HEB, KL exponents of $p(x)$, and $\eta, \alpha_v$ as the HEB, KL exponents of $v_{l,\tau}(x)$ on a subanalytic and compact set, respectively.

| (PP$_\gamma$) | $\mathcal{X}$ | $v_{l,\tau}(x)$ property | $\theta$ | $p(x)$ property | $\gamma$ | (CP) |
|---|---|---|---|---|---|---|
| $\epsilon$-global/local (Theorem 3.7-2) | compact or $\mathbb{R}^q$ | $\eta > 0$ | $\theta = \frac{\eta_p}{\eta}$ | $\eta_p \geq 1$ | $\epsilon^{\frac{1}{\eta_p}-1}$ | $(\epsilon, \epsilon)$-global/local $(\epsilon, \epsilon)$-global/local |
| $\epsilon$-stat. (Theorem 3.11) | $\mathbb{R}^q$ | $\alpha_v \geq 2$ | $\theta = 1$ | $\alpha_p \geq 2$ | $\epsilon^{-1}$ | $(\epsilon, \epsilon)$-KKT to (9) |

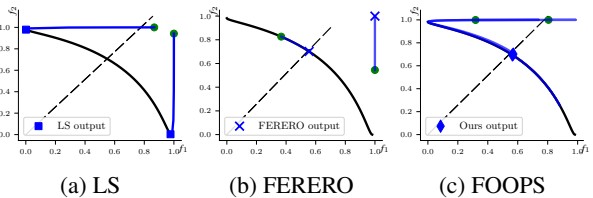

| (a) LS | (b) FERERO | (c) FOOPS |
|---|---|---|

*Figure 3.* Results of LS, FERERO, and FOOPS on Example 3.12. The black dashed lines is the preference defined by $H(x) = 0$. Green dots represent initial values, blue markers represent converged values for different methods.

the preference is modeled by different weights of the objectives. Another commonly used formulation is constrained vector optimization (Lin et al., 2019; Chen et al., 2024a), given by

$$\min_{x \in \mathcal{X}} F(x), \ \text{s.t.} \ G(x) \leq 0, H(x) = 0 \quad (11)$$

where $G, H$ are vector-valued functions defined by user-specified perferences. Though the above formulation has successful applications in, e.g., multi-task learning, it is only guaranteed to converge to a KKT point of (11), which may be far away from the optimal solutions to $\min_{x \in \mathcal{X}} F(x)$. See Example 3.12, and the corresponding results in Figure 3. An intuitive explanation for this suboptimality is that (11) puts constraints at the lower level, emphasizing more on satisfying the constraints rather than minimizing the objectives.

**Example 3.12.** Let $M = 2, q = 1$, and $M_g = 0, M_h = 1$. Let $\mathcal{X} = \mathbb{R}$. The objective $F$ and constraint $H$ for the preference-constained formulation (11) is defined as

$$F(x) = \left(1 - e^{-\|x - \mathbf{1}_q\|_2^2}, \ 1 - e^{-\|x + \mathbf{1}_q\|_2^2}\right), \quad (12a)$$
$$H(x) = 5f_1(x) - 4f_2(x). \quad (12b)$$

The corresponding preference function $f_0$ in formulation (OPS) to minimize the constraint violation of $H$ is defined as $f_0(x) = \|H(x)\|^2$. In Figure 3, it is easy to see that there exists a solution $x^*$ in the Pareto set whose objective $F(x^*)$ is at the intersection of the dashed line defined by $H(x) = 0$ and the solid curve representing the Pareto front. Therefore, $x^*$ is an optimal solution to both (OPS) and (11).

In this example, linear scalarization (LS) fails to converge to the preferred region that $H(x) = 0$, even after enumerating different weights. Indeed, we could prove that with

*Table 2.* Comparison with existing methods for OPS or semivectorial bilevel optimization. "NC", "C" and "SC" represent "nonconvex", "convex" and "strongly convex", respectively. "Ncs." represents whether the method converges to a necessary condition to the OPS problem.

| Method | $f_m(x)$, $m \in [M]$ | $f_0(x)$ | First order | Ncs. |
|---|---|---|---|---|
| PB-PDO (Kamani et al., 2021) | NC | NC | ✓ | ✗ |
| TAWT (Chen et al., 2022) | NC | NC | ✗ | ✗ |
| PNG (Ye & Liu, 2022) | NC | NC | ✓ | ✗ |
| PMM (Roy et al., 2023) | SC | SC | ✗ | ✓ |
| BSG (Giovannelli et al., 2024) | Stricly C | NC | ✗ | ✓ |
| Ours | NC | NC | ✓ | ✓ |

certain initialization, LS cannot converge to a point that satisfies (12b) with different weights, see Proposition 3.13. The intuition is that, in Example 3.12, a solution to both (OPS) and (11) that satisfies $\nabla F(x)\lambda = 0$ with $\lambda \in \Delta^M$ is a local maximum point of the objective $\lambda^\top F(x)$. It is also worth noting that, there are some other examples where LS cannot find all points on the Pareto front even after enumerating all possible weights (Osyczka, 1984; Athan, 1994; Hu et al., 2023). See a detailed discussion in Appendix A.2.

**Proposition 3.13.** *Under certain initializations, there exists no $\lambda \in \Delta^M$ such that gradient descent algorithm on LS objective with weight $\lambda$ converges to the solution of* (12)*.*

Moreover, algorithms developed under the preference-constrained formulation (11), such as PMTL (Lin et al., 2019) and FERERO (Chen et al., 2024a) (with the partial order cone being a nonnegative orthant cone $\mathbb{R}^M_+$), converge to a KKT point to (11), which is not necessarily Pareto optimal or Pareto stationary. For a more detailed discussion on this example, see Appendix A.2. Proposition 3.14 is provided to further support the claim that the KKT solution to (11) can be suboptimal for $\min_{x \in \mathcal{X}} F(x)$, with its proof in Appendix A.2.

**Proposition 3.14.** *The KKT solution to* (11) *is not necessarily Pareto stationary to $\min_{x \in \mathcal{X}} F(x)$.*

Besides modeling preference by weights or constraints, there are also some works which model the preference by objectives, and formulate the problem as optimization on the Pareto set (OPS), as in (OPS). Among these methods, Preference-Based Pareto Descent Optimization (PB-PDO) (Kamani et al., 2021) and Pareto Navigation Gradient descent (PNG) (Ye & Liu, 2022) use a descent-type algorithm to ensure the output of the algorithm satisfies the preference and decreases the objectives $F$ at each iteration. However, it has been shown in (Roy et al., 2023) that the

stationary condition derived in (Ye & Liu, 2022) is not a necessary optimality condition for (OPS). Furthermore, as discussed in (Roy et al., 2023, Proposition 3), a non-trivial stationarity condition for (OPS) typically requires second-order derivatives of the objectives $f_m, m \in [M]$. Different from these methods, Target-Aware Weighted Training (TAWT) (Chen et al., 2022) and Pareto Majorization-Minimization (PMM) (Roy et al., 2023) convert the lower-level vector-valued objective to a scalar-valued objective through linear scalarization (LS), and optimize both the scalarization weight and the model parameter. However, as discussed in Example 3.12 and Appendix A.2, optimality for LS is not necessary for (OPS) unless $f_m$ are convex for all $m \in [M]$.

We summarize in Table 2 the key differences of our work compared to existing methods for OPS. See also a detailed review in Section 5 and Appendix A.

## 4. Algorithms and Analysis

In this section, we introduce practical first-order gradient-based algorithms to solve (PP$_\gamma$). We first update $y$ to obtain an estimate for $y_{l,\tau}^*(x)$, and thus an estimate for $v_{l,\tau}(x) = -h_{l,\tau}(x, y_{l,\tau}^*(x))$. Then we update $x$ based on the estimated penalty function $\varphi_\gamma$.

At the $t$-th outer iteration and the $k$-th inner iteration, we iteratively update $x_t$ and $y_{t,k}$ as follows.

$$y_{t,k+1} = \mathrm{U}_y(y_{t,k}, \Delta y_{t,k}(y_{t,k}, x_t); \beta_{t,k}, k); \quad (13a)$$
$$x_{t+1} = \mathrm{U}_x(x_t, \Delta x_t(x_t, y_{t+1}); \alpha_t, t) \quad (13b)$$

where U is some gradient based oracle, and the gradient vectors with respect to $y$ and $x$ are defined as

$$\Delta y_{t,k}(y_{t,k}, x_t) = \nabla_y h_{l,\tau}(x_t, y_{t,k})$$
$$\Delta x_t(x_t, y_{t+1}) = \nabla f_0(x_t) - \gamma_t \theta \, \mathrm{sign}(v_t)|v_t|^{\theta-1} \nabla_x h_{l,\tau}(x_t, y_{t+1})$$

with $y_{t+1} = y_{t,K_t}$ and $v_t = \tau \ln M - h_{l,\tau}(x_t, y_{t+1})$.

A meta algorithm with the above updates is summarized in Algorithm 1. We name it First-Order Optimization on the Pareto Set (**FOOPS**) algorithm.

---
**Algorithm 1** The meta FOOPS algorithm with oracles
---
1: Initialize $t = 0$, $x_0, y_0$, set step sizes $\{\alpha_t, \beta_t\}$, penalty parameter $\{\gamma_t\}$, inner-loop iterations $\{K_t\}$.
2: **while** $\|\nabla \varphi_{\gamma_t}(x_t)\|^2 > \epsilon$ **do**
3:    Set $k = 0$;
4:    **for** $k = 0, \ldots, K_t - 1$ **do**
5:       Update $y_{t,k}$ by (13a);
6:    **end for**
7:    Set $y_{t+1} = y_{t,K_t}$;
8:    Update $x_t$ by (13b);
9:    Set $t = t + 1$;
10: **end while**
---

We then discuss the choices of update oracles and the non-asymptotic convergence rate of Algorithm 1 with different oracles below. Let $w$ represent the updated parameter, which can be either $x$ and $y$, $\Delta w$ denote the gradient vector, $\alpha$ the stepsize, and $t$ is the iteration number.

We give examples of oracles using projected gradient desent (PGD), and momentum updates (Momentum) in (14). The Nesterov's acceleration and Adam update are also applicable, which is detailed in Appendix E, one can also see in e.g., (Wang et al., 2024).

$$\text{PGD: } \mathrm{U}(w, \Delta w; \alpha_t, t) = \mathrm{Proj}_\mathcal{X}(w - \alpha_t \Delta w) \quad (14a)$$
$$\text{Momentum: } \mathrm{U}(w, \Delta w; \alpha_t, t) = \mathrm{Proj}_\mathcal{X}(w - \alpha_t v_t),$$
$$\text{with } v_t = \tilde{\alpha} v_{t-1} + \Delta w \quad (14b)$$

**Discussion about the convergence.** Since $h_{l,\tau}(x, y)$ is $\mu_{h_y}$-strongly convex w.r.t. $y$ as detailed in the proof of Lemma E.4, when we choose $\mathrm{U}_y$ as projected gradient descent, the momentum updates and Nesterov's acceleration, it gives linear convergence rate for the inner-loop of $y$. For the outer-loop w.r.t. $x$, if $\theta \geq 1$, then the objective $\varphi_\gamma(x)$ is differentiable and thus projected gradient descent, momentum, Nesterov's acceleration, Adam updates give $O(1/T)$ convergence rate in the deterministic setting and $O(1/\sqrt{T})$ convergence rate in the stochastic setting according to (Wang et al., 2024). Therefore, combining outer-loop and inner-loop update oracles together, Algorithm 1 converges. We provide a proof for the convergence of Algorithm 1 in Appendix E, Theorem E.6, when choosing both $\mathrm{U}_y$ and $\mathrm{U}_x$ as the PGD oracle and choosing $\theta \geq 1$, and assuming the objectives $f_m(x)$ for $m = 0, \ldots, M$ are smooth. For $\theta < 1$, $p(x)$ can be nonsmooth. The convergence for Algorithm 1 can be built upon nonsmooth optimization (Kiwiel, 2004; Davis et al., 2018), but possibly under additional assumptions. Some discussions about algorithms for nonsmooth lower-level objective are provided in (Chen et al., 2024b). When $f_0(x)$ and $p(x)$ are Lipschitz, the algorithm can be applied to our problem. We leave a more detailed study of algorithm development in nonsmooth cases for future research.

**Single-loop and stochastic variants.** When we take the exact penalty method with $\eta_p = 1$, $\gamma_t$ and $K_t$ can be upper bounded by a constant. In fact, under Assumption 2, when $l > \ell_{f,1}$, a single-loop variant of Algorithm 1 that takes $K_t = 1$ also has non-asymptotic convergence guarantees, see e.g. (Chen et al., 2021). Besides, stochastic variants of Algorithm 1 can also be derived which replace the deterministic gradients with their unbiased stochastic estimates. We leave the development and convergence analysis of such variants for future work.

## 5. Related Works

We discuss recent works that are most related to ours. An extended discussion is provided in Appendix A.1.

**Preference-vector-guided multi-objective learning.** Preferences in multi-objective optimization can be represented using weights, thresholds, or preference vectors. Scalarization methods, such as linear and Tchebycheff scalarization, convert vector objectives into scalar objectives by applying weighted norms (Miettinen, 1998). Alternatively, $\epsilon$-constraint methods impose thresholds on objectives to convert the problem to a constrained optimization problem (Curtis et al., 2023). There are also other approaches which represent preferences with vectors in the objective space, focusing on finding optimal solutions satisfying constraints defined by these vectors (Lin et al., 2019; Chen et al., 2024a) or minimizing distances to the vectors (Mahapatra & Rajan, 2020; Momma et al., 2022). See also a comprehensive review in (Chen et al., 2025).

**Optimization on the Pareto Set.** In machine learning, there are specific instantiations of the OPS problem. For example, EPO (Mahapatra & Rajan, 2020) and Preference-Based Pareto Descent Optimization (PB-PDO) (Kamani et al., 2021) find a Pareto model such that the objective values satisfy a ratio constraint by minimizing the non-uniformity score. Specifically, EPO finds update directions to improve the objective values, or to reduce the constraint violations, or both. PB-PDO finds common descent directions for the lower-level multi-objectives and the upper-level objective. They are not guaranteed to converge to a necessary condition of the OPS problem. Target-Aware Weighted Training (TAWT) (Chen et al., 2022) learns a multi-task model that minimizes the discrepancy of task representations to ensure they are similar. TAWT converts the lower-level objectives to a linearly scalarized objective, and optimizes the scalarization weights in the upper level. This approach has the limitation of introducing undesired local solutions.

**(Simple) bilevel optimization.** Problem (CP) is a constrained reformulation of the simple bilevel program with a generally nonconvex lower-level (LL) objective. For nonconvex LL objectives, algorithms are proposed in e.g., (Liu et al., 2021b; Huang, 2023; Xiao et al., 2023b). However, they usually require second-order derivatives in the algorithms, which can be expensive to implement. More recently, *first-order Hessian-free* approaches were proposed to address this by the value function reformulation of BLO. For example, sequential quadratic programming (Liu et al., 2022), and smoothed Lagrangian method (Lu, 2023) were used to solve the constrained problem. Later on, a *penalty-based* algorithm was proposed (Shen et al., 2025). Variants such as Moreau Envelope based algorithms (Kwon et al., 2024; Liu et al., 2024) and adaptive algorithms (Chen et al., 2024b) were proposed. However, none of these works tackle BLO with vector-valued LL objective. Moreover, even af-

ter converting the vector-valued LL objective to a scalar-valued one through the merit function $v_{l,\tau}$, the LL objective is generally nonconvex and non-PL even if the objectives $f_m, m \in [M]$ are all convex or PL. Therefore, it is difficult to directly apply the existing analysis or algorithms to our problem.

## 6. Experiments

In this section, we conduct experiments to verify our theory and show the applicability of the algorithms to preference-guided multi-task learning. We use LS, PMTL (Lin et al., 2019), EPO (Mahapatra & Rajan, 2020), XWC-MGDA (XM) (Momma et al., 2022), FERERO (Chen et al., 2024a) as baselines for comparison. For a preference vector-guided MOL problem, we define $f_0(x) = \|H(x)\|^2$, where $H(x)$ is the equality constraint function derived from the preference vector (Chen et al., 2024a).

**Metrics.** *Objective loss and accuracy.* We report the objective losses and accuracies in classification.

*Hypervolume.* Let $F' \in \mathbb{R}^M$ denote the Nadir point, i.e., the worst performance on single-task baselines, and $\mathcal{S}$ denote a set of objective function values of the obtained models. Hypervolume measures the size of the dominated space of $\mathcal{S}$ relative to $F'$, which can be computed by $H(\mathcal{S}) = \Lambda(\{q \in \mathbb{R}^M \mid \exists F \in \mathcal{S} : F \le q \le F'\})$, where $\Lambda(\cdot)$ denotes the Lebesgue measure.

**Additional details.** The implementation and additional experiments can be found in Appendix F.

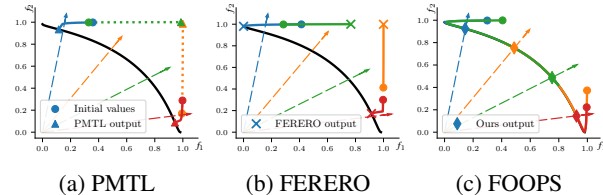

(a) PMTL     (b) FERERO     (c) FOOPS

*Figure 4.* Outputs (colored markers) and optimization trajectories (colored curves) of different methods when initial objectives are near the Pareto front. Dashed arrows with different colors represent different preferences.

**Example 3.12.** Following (Lin et al., 2019; Mahapatra & Rajan, 2020), the first objective we consider is (12a) in Example 3.12, but with $q = 20$. The results for the experiments with hard initialization are displayed in Figure 4. They show that under certain initializations and preferences, algorithms developed under (11) such as PMTL and FERERO (with $A = I$ therein) could fail to reach the Pareto front. It further justifies our Proposition 3.14 that the KKT solution to (11), with preferences modelled by the constraints at the lower level, can be suboptimal for $\min_{x \in \mathcal{X}} F(x)$. In contrast, FOOPS successfully converges to preferred Pareto optimal solutions under different preferences. It demonstrates the benefit of the OPS formulation over (11), by prioritizing the

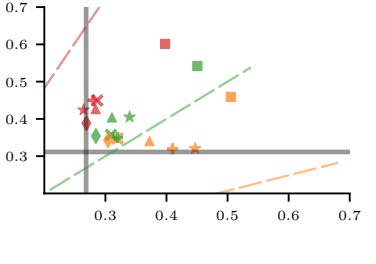

(a) Multi-MNIST loss

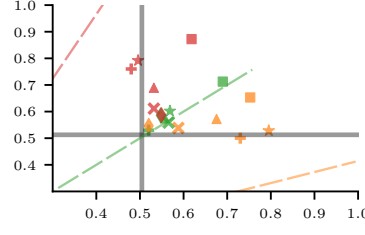

(b) Multi-Fashion loss

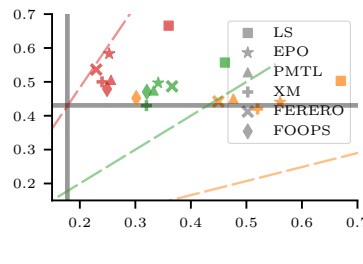

(c) Multi-F+M loss

*Figure 5.* Losses of various methods with different preferences across three image datasets. The horizontal and vertical axes represent results for objective 1 and objective 2, respectively. Different colored dashed arrows indicate various preference vectors. Different markers denote the solutions obtained by different methods, with marker colors matching the preferences.

*Table 3.* Hypervolumes of different methods $\uparrow (\times 10^{-2})$.

| Datasets | LS | PMTL | EPO | XM | FERERO | FOOPS |
|---|---|---|---|---|---|---|
| Mt-M loss | 1.68 | 1.41 | 1.35 | 1.42 | $1.95_{\pm 0.21}$ | $\mathbf{2.62}_{\pm 0.21}$ |
| Mt-F loss | 6.75 | 5.90 | 6.02 | 6.77 | $7.76_{\pm 0.18}$ | $\mathbf{8.32}_{\pm 0.37}$ |
| Mt-F+M loss | 3.63 | 3.03 | 3.76 | 3.89 | $3.82_{\pm 0.21}$ | $\mathbf{4.80}_{\pm 0.45}$ |
| Mt-M accuracy | 0.19 | 0.15 | 0.15 | 0.16 | $0.25_{\pm 0.04}$ | $\mathbf{0.33}_{\pm 0.02}$ |
| Mt-F accuracy | 0.99 | 0.87 | 0.87 | 0.99 | $1.13_{\pm 0.07}$ | $\mathbf{1.22}_{\pm 0.07}$ |
| Mt-F+M accuracy | 0.48 | 0.40 | 0.50 | 0.52 | $0.53_{\pm 0.04}$ | $\mathbf{0.72}_{\pm 0.06}$ |

*Table 4.* WERs $\downarrow$ (%) for speech recognition.

| Method | English | Chinese | Average |
|---|---|---|---|
| Komatsu et al. | 7.11 | - | - |
| w/o CPC | 11.8 | 10.2 | 11.0 |
| Init. (M2ASR) | 7.3 | 6.2 | 6.7 |
| LS-FT | 6.8 | 5.9 | 6.4 |
| FERERO-FT | **5.4** | 4.9 | **5.1** |
| FOOPS-FT | 5.7 | **4.7** | **5.1** |

attainment of weak Pareto optimality instead of the optimality of the preference function.

**Multi-patch image classification.** Following (Momma et al., 2022), we use Multi-MNIST (Mt-M), Multi-Fashion (Mt-F), and Multi-Fashion+MNIST (Mt-F+M) for image classification. The two tasks or objectives in all three datasets are to classify the top-left and the bottom-right images, respectively. We use LeNet as the backbone neural network. The losses of different methods given different preference vectors are plotted in Figure 5. Experiments for our method are repeated 5 times. Hypervolumes with means and standard deviations are reported in Table 3. The results for other methods in Table 3 are referenced from (Momma et al., 2022; Chen et al., 2024a). The results show that FOOPS is better at obtaining large hypervolumes, see Table 3, but worse at aligning with preferences compared to other methods, see Figure 5.

**Multi-lingual speech recognition.** We use the proposed method to fine-tune a pre-trained multi-lingual speech recognition model. The datasets include Librispeech (100 hours) (Panayotov et al., 2015), and AISHELL v1 (Bu et al., 2017). The model architecture is a conformer with 8 blocks. The speech recognition Connectionist Temporal Classification (CTC) losses in Chinese and English are denoted as $f_t^{\text{ch}}$ and $f_t^{\text{en}}$, respectively. We also use the self-supervised Contrastive Predictive Coding (CPC) loss $f_p$ for representation learning, i.e.,

$$\min f_0(x) := \| f_t^{\text{ch}}(x) - f_t^{\text{en}}(x) \|^2 \tag{15a}$$

$$\text{s.t.} \quad x \in \arg\min \ F(x) := \left( f_p(x), f_t^{\text{ch}}(x), f_t^{\text{en}}(x) \right)^\top \tag{15b}$$

where the lower-level objective $f_p$ ensures the model learns

a good representation, and the upper-level objective ensures the difference of the performances on both languages is small; see more details in Appendix F. The results are reported in Table 4, which show that FOOPS demonstrate competitive average performance on different languages, but is less good at optimizing the fairness preference function $f_0$. This observation is consistent with that in Figure 5.

## 7. Conclusions

In this work, we cast preference-guided multi-objective learning as an optimization on the Pareto set (OPS) problem, which is essentially a semivectorial simple bilevel optimization problem, with a lower-level vector-valued objective, and an upper-level scalar-valued preference objective. We propose a first-order penalty method to solve the problem, where the penalty function is the polynomial of a smoothed merit function. For the theoretical analysis, first we establish the properties of the merit function, including its relation to weak Pareto optimality, and the Hölderian error bound. Then we discuss the relation of solutions to the penalty reformulation and the original OPS problem. Part of our theoretical analysis is of independent interest to simple bilevel optimization problem. Interestingly and perhaps surprisingly, our analysis shows that although the stationary condition of the OPS problem usually requires second-order derivative information, it can be approximated using first-order methods. Based on the results, we develop first-order algorithms to solve the penalty problem and provide their convergence guarantees. For the empirical experiments, we apply the proposed method to synthetic and real-world problems, which demonstrate the effectiveness of the proposed method in finding preference-guided optimal solutions.

## Acknowledgement

The work of Lisha Chen, Quan Xiao, and Tianyi Chen was supported by Cisco Research Gifts and the IBM through the IBM-Rensselaer Future of Computing Research Collaboration. The work of Ellen Hidemi Fukuda was supported by the Japan Society for the Promotion of Science, Grant-in-Aid for Scientific Research (C) (JP25K15002). The work of Xinyi Chen was done while she was a remote intern at Rensselaer Polytechnic Institute.

## Impact Statement

This paper presents work whose goal is to advance the field of Machine Learning, especially algorithms and theory for optimization on the Pareto set with applications to preference-guided multi-objective learning. There are many potential societal consequences of our work, none which we feel must be specifically highlighted here.

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

# Appendix

Throughout the paper, we assume $f_m(x), m = 0, \ldots, M$ are twice continuously differentiable and bounded below, and the minimizer of $\lambda^\top F(x)$ exists for all $\lambda \in \Delta^M$.

For notation simplicity, we use $\mathrm{LSE}_\tau : \mathbb{R}^M \to \mathbb{R}$ to denote the Log-sum-exp function with parameter $\tau$. We say a function is $\mu$-(weakly) convex, with $\mu \in \mathbb{R}$. If $\mu > 0$, the function is strongly convex, if $\mu = 0$, the function is convex, and if $\mu < 0$, the function is weakly convex.

**Organization of the appendix.** We organize the proof in the appendix as follows.

In Appendix A, we discuss additional related work on (semivectorial) bilevel optimization and limitation of linear scalarization and preference as constraint formulations.

- Appendix A.1: additional related work on bilevel and multi-objective optimization

- Appendix A.2: limitations of linear scalarization and preference as constraint

In Appendix B, we prove the basic properties of the merit function.

- Appendix B.2: continuity of the merit function

- Appendix B.3: relations of $v_{l,\tau}$ and weak Pareto optimality

In Appendix C, we discuss (global) subanalyticity and prove related properties such as the subanalyticity, Hölderian error bound (HEB), and KL inequality of the merit function.

- Appendix C.1: proof of global subanalyticity of $v_{l,\tau}$

- Appendix C.2: examples of globally subanalytic functions and their HEB

- Appendix C.3: relations of proximal error bound (EB), proximal KL, and HEB

In Appendix D, we prove the relations of different formulations.

- Appendix D.1: $\epsilon$-global/local solutions relation of the smoothed problem

- Appendix D.2: $\epsilon$-global/local solutions relation between the constrained problem and the penalty problem

- Appendix D.3: $\epsilon$-stationary solutions relation between the constrained problem and the penalty problem

In Appendix E, we prove the convergence of the proposed algorithm.

In Appendix F, we provide implementation details and additional experiment results.

## A. Extended discussion of related works

In this section, we provide an extended review of recent works that are closely related to ours.

### A.1. Bilevel optimization

Bilevel optimization (BLO) is a classical problem that dates back to (Stackelberg, 1952; Vicente & Calamai, 1994; Luo et al., 1996a). Gradient-based approaches with non-asymptotic convergence analysis and applications to machine learning were studied in e.g., (Ghadimi & Wang, 2018; Chen et al., 2021; Ji et al., 2021; Hong et al., 2023). These works focus on problems with (strongly) convex lower-level (LL) objectives. Similarly, in simple bilevel optimization with shared optimization variable in the upper- and lower-level objectives, a line of works focus on convex LL objectives, including e.g., (Jiang et al., 2023; Merchav & Sabach, 2023; Giang-Tran et al., 2024; Samadi et al., 2024; Doron & Shtern, 2023).

For nonconvex LL objectives, a pessimistic algorithm with asymptotic convergence guarantee was proposed in (Liu et al., 2021b), the stationary metric was studied for BLO with lower-level PL objective, and an alternating descent algorithm with non-asymptotic analysis was proposed in (Xiao et al., 2023b). In Table 5, we provide a summary of the bilevel optimization works with possibly nonconvex lower-level objectives which may satisfy the Hölderian error bound (HEB).

**Semivectorial BLO.**   OPS can be seen as a semivectorial simple BLO problem, where the bilevel program has a vector-valued lower-level (LL) objective and a scalar-valued upper-level (UL) objective. To solve such problems, one straightforward approach is to convert the LL vector-valued objective to a scalar-valued objective through scalarization, and to optimize the scalarization parameter in the upper level, see, e.g., (Roy et al., 2023). However, it has been shown that this reformulation could induce additional local minimizers or stationary solutions (Dempe & Mehlitz, 2019; Benko & Mehlitz, 2021). Furthermore, this reformulation might require stronger constraint qualifications than the original problem (Benko & Mehlitz, 2021). Alternatively, penalty-based reformulations have been considered (Bonnel & Morgan, 2006), where the penalty function is defined as the maximum improvement amount of the vector-valued objective. However, no practical implementation, or relation of solutions to the original problem in nonconvex settings, or non-asymptotic convergence analysis are provided for the reformulation. Recently, in (Giovannelli et al., 2024), deterministic and stochastic risk-neutral and risk-averse algorithms are proposed, under the assumption that the LL objective is strictly convex w.r.t. the LL variable, and requiring the second-order derivative of the objective.

Another line of research study the problem with vector-valued upper-level (UL) objective, and scalar-valued LL objective (Ye et al., 2021; Gu et al., 2023; Ye et al., 2024; Yang et al., 2024). It is sometimes also referred to as the multi-objective BLO problem. For a more detailed review of multi-objective BLO algorithms, see a survey (Mejía-De-Dios et al., 2023) and the references therein. Besides, single-level multi-objective learning algorithms have been extensively studied, these include the variants of the multi-gradient descent algorithm (Fliege & Svaiter, 2000; Sener & Koltun, 2018; Liu et al., 2021a; Liu & Vicente, 2021; Fernando et al., 2023; Chen et al., 2023; Xiao et al., 2023a) that are designed to avoid gradient conflicts during the optimization procedure. For a more detailed review of multi-objective learning algorithms, see a survey (Chen et al., 2025) and the references therein.

**Comparison of the theory in Section 3.2 to existing works.**   The works most related to ours regarding the theory in Section 3.2 include (Ye et al., 1997; Luo et al., 1996b) and recent works (Shen et al., 2025; Chen et al., 2024b). However, a major difference is that they directly assume the lower-level objective satisfies HEB, while in our work, we prove the property holds on a bounded set for $v_{l,\tau}(x)$ and $p(x)$ when the objective $F$ is subanalytic. The two works (Ye et al., 1997; Luo et al., 1996b) focus on the cases with exact penalty. Other differences include that the results in (Shen et al., 2025) only consider LL objective satisfies HEB with $\eta = 2$, while we consider more general $\eta$. Also, we do not require the global convexity or Lipschitz assumption as in (Chen et al., 2024b), which is generally not satisfied by our merit function. Furthermore, we provide the relation between the stationary solution of the penalty formulation and the KKT solution of the constrained formulation under the general KL inequality, which is not discussed in either of the two works.

### A.2. Limitation of linear scalarization and preference as constraint

Besides using empirical results on Example 3.12, we also provide theoretical justifications to show the limitations of LS and KKT solutions for preference-guided MOL.

**Limitation of linear scalarization.**   We first discuss the limitation of linear scalarization (LS) for preference-guided MOL. It is known that LS is not good at handling nonconvex Pareto front. Prior works have shown that the optimality condition of LS is not a necessary condition for Pareto optimality, see e.g., (Athan, 1994, Proposition 3.3). As a result, the solution set of LS, even by enumerating all possible weights of objectives, does not include all Pareto optimal solutions, and thus it does not include Pareto optimal solutions under certain preferences.

Below we provide the proof of Proposition 3.13, which shows that under certain initializations, gradient descent on the LS objective does not converge to a preferred Pareto optimal solution in Example 3.12.

*Proof of Proposition 3.13.*  In Example 3.12, there exists a solution $x^* \in (-1, 1)$, which is an optimal solution to both (OPS) and (11). And there exists $\lambda = [\lambda_1, \lambda_2]^\top \in \Delta^2$ with $\lambda_2 = \lambda_1 \frac{(1-x^*)e^{-(x^*-1)^2}}{(1+x^*)e^{-(x^*+1)^2}}$ that $\nabla F(x^*)\lambda = 0$. To show this, let $c$ denote

*Table 5.* Comparison with existing methods for (simple) bilevel optimization with lower-level scalar objective, "SC" and "C" represent "strongly convex" and "convex", respectively; "comp" represents "compact set"; "Lip" represents "Lipschitz continuous". For non-simple bilevel optimization problem, the lower-level and upper-level properties are all w.r.t. the lower-level variable for a meaningful comparison. The lower-level objective in our problem is $v_{l,\tau}(x) + \tau \ln M$.

| Method | lower-level HEB | other lower-level properties | upper-level | first-order |
|---|---|---|---|---|
| | | non-simple bilevel optimization | | |
| IAPTT-GM (Liu et al., 2021b) | - | smooth, comp | smooth, comp | ✗ |
| BOME (Liu et al., 2022) | $\eta = 2$ | PL, Lip, smooth | Lip, smooth | ✓ |
| PBGD (Shen et al., 2025) | $\eta = 2$ | PL, smooth 
 C, smooth | Lip, smooth 
 Lip, smooth | ✓ |
| GALET (Xiao et al., 2023b) | $\eta = 2$ | PL, smooth | Lip, smooth | ✗ |
| AGILS (Bai et al., 2024) | $\eta \geq 1$ | KL, weakly C (composite) | smooth | ✓ |
| MEHA (Liu et al., 2024) | - | smooth 
 weakly C (composite) | smooth 
 smooth | ✓ |
| SLM (Lu, 2023) | $\eta = 2$ | PL, smooth | smooth, comp | ✓ |
| | | simple bilevel optimization | | |
| CG-BiO (Jiang et al., 2023) | $\eta \geq 1$ | C, smooth 
 C, smooth | C, smooth 
 non-C, smooth | ✓ |
| R-APM (Samadi et al., 2024) | $\eta = 1$ | C, composite | C, smooth | ✓ |
| PB-APG (Chen et al., 2024b) | $\eta \geq 1$ | C, composite 
 C, composite 
 nonsmooth, Lip | C, composite 
 SC, composite 
 nonsmooth, Lip | ✓ |
| FOOPS (ours) | $\eta > 0$, 
 (modified by $\theta = \frac{\eta_p}{\eta} \geq \frac{1}{\eta}$) | subanalytic (provable) 
 KL (provable) | locally Lip 
 locally Lip | ✓ |

a positive constant, note that

$$
\begin{aligned}
\nabla F(x)\lambda &= 2\lambda_1 e^{-(x-1)^2}(x-1) + 2\lambda_2 e^{-(x+1)^2}(x+1) \\
&= c\Big(2e^{-(x-1)^2}(x-1)(1+x^*)e^{-(x^*+1)^2} + 2e^{-(x+1)^2}(x+1)(1-x^*)e^{-(x^*-1)^2}\Big) \\
&= 2c(x+1)(x^*+1)e^{-(x+1)^2-(x^*+1)^2}\Big(e^{4x}\frac{x-1}{x+1} - \frac{x^*-1}{x^*+1}e^{4x^*}\Big).
\end{aligned}
\tag{16}
$$

Let $r(x) = \frac{x-1}{x+1}e^{4x}$. Then $r(x) > r(x^*) \iff \nabla F(x)\lambda > 0$ and vice versa for $r(x) < r(x^*)$. The above equation implies that when $r(x') = r(x^*)$, $\nabla F(x')\lambda = 0$. Therefore, $\nabla F(x^*)\lambda = 0$. Also observing that there exists different points $-1 < x_1 < x^* < x_2 < 1$ that

$$
r(x_1) = r(x_2) = r(x^*).
\tag{17}
$$

This means that $x_1, x_2$ are all stationary points of $\lambda^\top F(x)$. Furthermore,
1. for $x' \in (x_1, x^*) \cup (x_2, 1)$, $r(x') > r(x^*)$, thus $\nabla F(x')\lambda > 0$, then gradient descent (GD) on $\lambda^\top F(x)$ starting from $x' \in (x_1, x^*)$ with sufficiently small step size converges to $x_1$, and it converges to $x_2$ if starting from $x' \in (x_2, 1)$;
2. for $x' \in (-1, x_1) \cup (x^*, x_2)$, $r(x') < r(x^*)$, thus $\nabla F(x')\lambda < 0$, then GD on $\lambda^\top F(x)$ starting from $x' \in (-1, x_1)$ with sufficiently small step size converges to $x_1$, and it converges to $x_2$ if starting from $x' \in (x^*, x_2)$.

This proves that they will not converge to $x^*$. □

**Limitation of preference as constraint.** We then discuss in more detail of the limitation of modeling preference by constraints. We verify Proposition 3.14 by constructing an example with a KKT but non-Pareto stationary solution. Using Example 3.12, and noticing that one solution that PMTL converges to, denoted as $x^*$, has objective value that is approximately $F(x^*) \approx [0.80; 0.99]$, which satisfies the feasibility condition that $H(x^*) = 0$. Furthermore, its gradient at $x^*$ can be computed approximately as

$$
\nabla f_1(x^*) \approx 0.5062;
\tag{18a}
$$
$$
\nabla f_2(x^*) \approx 0.0002.
\tag{18b}
$$

Clearly, $x^*$ is not Pareto stationary. By the KKT stationarity condition, we further have

$$\nabla F(x)\lambda_f + \nabla H(x)\lambda_h = \nabla F(x)\Big(\lambda_f + \begin{bmatrix} 5 \\ -4 \end{bmatrix}\lambda_h\Big)$$

$$=\nabla F(x)\Big(\lambda_f + \begin{bmatrix} 5\lambda_h \\ -4\lambda_h \end{bmatrix}\Big) = 0, \text{ for some } \lambda_f \in \Delta^2, \lambda_h \in \mathbb{R}. \tag{19}$$

It can be verified that $0.5062(\lambda_{f,1}+5\lambda_h)+0.0002(1-\lambda_{f,1}-4\lambda_h)=0$ has solutions, e.g., $\lambda_f = [0;1]$, $\lambda_h \approx -7.9 \times 10^{-5}$. Therefore, $x^*$ is a KKT point.

We use another example with strongly convex objectives to prove Proposition 3.14 that a KKT point to (11) is not necessarily Pareto stationary.

**Example A.1.** Let $M = 2, q = 2$, and $M_g = 0, M_h = 1$. Let $\mathcal{X} = \mathbb{R}^2$, and $x = [x_1; x_2]$. The objective $F$ and constraint $H$ is defined as

$$F(x) =\big((x_1 - 1)^2 + x_2^2, \ 0.5x_1^2 + x_2^2\big) \tag{20a}$$
$$H(x) =9f_1(x) - 8f_2(x) \tag{20b}$$

*Proof of Proposition 3.14.* In Example A.1, the gradient of $F$ can be computed by

$$\nabla F(x) = \begin{bmatrix} 2(x_1 - 1) & x_1 \\ 2x_2 & 2x_2 \end{bmatrix}. \tag{21}$$

For $x = [3;0]$, $\nabla F(x) = \begin{bmatrix} 4 & 3 \\ 0 & 0 \end{bmatrix}$. Let $\Delta^M$ denote the $(M-1)$-simplex. Apparently, there exists no $\lambda \in \Delta^2$ such that $\nabla F(x)\lambda = 0$. Therefore, $x$ is not Pareto stationary.

Then we check whether $x$ satisfies the KKT condition. First, it satisfies the feasiblity condition since $H(x) = 0$. Second, by invoking the KKT stationarity condition, for some $\lambda_f \in \Delta^2, \lambda_h \in \mathbb{R}$, we have

$$\nabla F(x)\lambda_f + \nabla H(x)\lambda_h = \nabla F(x)\Big(\lambda_f + \begin{bmatrix} 9 \\ -8 \end{bmatrix}\lambda_h\Big) = \begin{bmatrix} 4 & 3 \\ 0 & 0 \end{bmatrix}\Big(\lambda_f + \begin{bmatrix} 9\lambda_h \\ -8\lambda_h \end{bmatrix}\Big) = 0. \tag{22}$$

It can then be verified that the above holds true when $\lambda_f = [0, 1] \in \Delta^2$, and $\lambda_h = -0.25$.

Therefore, $x$ is a KKT point but not a Pareto stationary point. The proof is complete. $\qquad\square$

## B. Proof of the properties of the merit functions

For convenience, we define the merit function $u_l(x)$ and restate the smoothed merit function $v_{l,\tau}(x)$ below.

$$u_l(x) := \max_{y \in \mathcal{X}} \min_{m \in [M]} \Big\{ f_m(x) - f_m(y) - \frac{l}{2}\|x - y\|^2 \Big\} \tag{23}$$

$$v_{l,\tau}(x) := -\min_{y \in \mathcal{X}} \Big\{ \tau \ln\Big( \sum_{m=1}^{M} e^{\frac{f_m(y)-f_m(x)}{\tau}} \Big) + \frac{l}{2}\|x - y\|^2 \Big\}. \tag{24}$$

Correspondingly, we define $h_{l,\tau}(x,y)$ below for analysis. Note that $v_{l,\tau}(x) = -\min_{y \in \mathcal{X}} h_{l,\tau}(x,y)$.

$$h_{l,\tau}(x,y) := \tau \ln\Big( \sum_{m=1}^{M} e^{\frac{f_m(y)-f_m(x)}{\tau}} \Big) + \frac{l}{2}\|x - y\|^2. \tag{25}$$

### B.1. Auxiliary lemmas

**Lemma B.1** (Restatement of (Tanabe et al., 2024, Theorems 3.1 and 3.3))**.** *For $l \geq 0$, consider the merit function $u_l(x)$ defined in (23). Then $u_l(x) \geq 0$ for $l \geq 0$. $u_0(x) = 0$ if and only if $x$ is weakly Pareto optimal. For $l > 0$, $x$ is weakly Pareto optimal implies $u_l(x) = 0$; furthermore, if $f_m(x)$ is convex for all $m \in [M]$, then $u_l(x) = 0$ implies $x$ is weakly Pareto optimal.*

**Proposition B.2** (Smoothness implies weak convexity). *If a locally Lipschitz function $f$ is $\ell_{f,1}$-smooth, then it is also $-\ell_{f,1}$-weakly convex.*

**Lemma B.3** (Log-sum-exp function preserves weak convexity). *Let $f_m(x), m \in [M]$ be weakly convex with modulus $\mu_m \in \mathbb{R}$. Let $\bar{\mu} = \min_{m \in [M]} \mu_m$. Then $\ln\left(\sum_{m=1}^{M} e^{f_m(x)}\right)$ is weakly convex with modulus $\bar{\mu}$.*

*Proof of Lemma B.3.* By definition, and since $\bar{\mu} = \min_{m \in [M]} \mu_m$, we have $f_m(x) - \frac{\bar{\mu}}{2}\|x\|^2$ is convex for all $m \in [M]$. Also because the Log-sum-exp function preserves convexity, we have that $\ln\left(\sum_{m=1}^{M} e^{f_m(x) - \frac{\bar{\mu}}{2}\|x\|^2}\right)$ is convex. Further rearranging this function, we have

$$\ln\left(\sum_{m=1}^{M} e^{f_m(x) - \frac{\bar{\mu}}{2}\|x\|^2}\right) = \ln\left(e^{-\frac{\bar{\mu}}{2}\|x\|^2}\left(\sum_{m=1}^{M} e^{f_m(x)}\right)\right) = \ln\left(\sum_{m=1}^{M} e^{f_m(x)}\right) - \frac{\bar{\mu}}{2}\|x\|^2 \tag{26}$$

which is convex. By definition, this implies that $\ln\left(\sum_{m=1}^{M} e^{f_m(x)}\right)$ is weakly convex with modulus $\bar{\mu}$. The proof is complete. $\qquad\square$

**Corollary B.4.** *If $f_m(x), m \in [M]$ is weakly convex with modulus $\mu_m \in \mathbb{R}$, and $l + \min_{m \in [M]} \mu_m > 0$, then $h_{l,\tau}(x,y)$ defined in (25) is strictly convex w.r.t. $y$, and the solution to $\min_{y \in \mathcal{X}} h_{l,\tau}(x,y)$ is a singleton.*

*Proof of Corollary B.4.* By the $\mu_m$-weak convexity, $\frac{1}{\tau} f_m(y)$ is $\frac{\bar{\mu}}{\tau}$-weakly convex w.r.t. $y$. Therefore, combining with Lemma B.3, the function $\tau \ln\left(\sum_{m=1}^{M} e^{\frac{f_m(y) - f_m(x)}{\tau}}\right)$ is $\bar{\mu}$-weakly convex w.r.t. $y$, where $\bar{\mu} = \min_{m \in [M]} \mu_m$. Since $l + \min_{m \in [M]} \mu_m = l + \bar{\mu} > 0$, $h_{l,\tau}(x,y)$ is strictly convex w.r.t. $y$, and the solution to $\min_{y \in \mathcal{X}} h_{l,\tau}(x,y)$ is unique. The proof is complete. $\qquad\square$

**Lemma B.5.** *If $f_m(x), m \in [M]$ is continuous and weakly convex with modulus $\mu_m$, and $l \geq -\min_{m \in [M]} \mu_m$, then there exists $x \in \mathcal{X}$ such that $v_{l,\tau}(x) = -\tau \ln M$.*

*Proof of Lemma B.5.* From Corollary B.4, since $l \geq -\min_{m \in [M]} \mu_m$, $h_{l,\tau}(x,y)$ is convex w.r.t. $y$. Let $P$ be the indicator function defined on $\mathcal{X}$. Then, $0 \in \nabla_y h_{l,\tau}(x,y) + \partial P(y)$ if and only if $y = \arg\min_{y \in \mathcal{X}} h_{l,\tau}(x,y)$.

By the definition of $h_{l,\tau}(x,y)$, the gradient $\nabla_y h_{l,\tau}(x,y)$ can be derived as

$$\nabla_y h_{l,\tau}(x,y) = \sum_{m=1}^{M} \frac{e^{\frac{f_m(y) - f_m(x)}{\tau}}}{\sum_{m=1}^{M} e^{\frac{f_m(y) - f_m(x)}{\tau}}} \nabla f_m(y) + l(y - x). \tag{27}$$

When $y = x$, it can be further derived that

$$\nabla_y h_{l,\tau}(y,y) = \frac{1}{M} \sum_{m=1}^{M} \nabla f_m(y). \tag{28}$$

Recall that $\lambda^\top F(x)$ is lower bounded for all $\lambda \in \Delta^M$. And $\lambda^\top F(x)$ is continuous since $f_m$ are continuous for all $m \in [M]$. We assume either $\mathcal{X}$ is compact, or $\mathcal{X} = \mathbb{R}^q$ and $f_m$ is coercive for all $m \in [M]$. Then the solution to $\min_{x \in \mathcal{X}} \lambda^\top F(x)$ exists. Let $x^* = \arg\min_{x \in \mathcal{X}} \frac{1}{M} \sum_{m=1}^{M} f_m(x)$, which implies

$$0 \in \frac{1}{M} \sum_{m=1}^{M} \nabla f_m(x^*) + \partial P(x^*). \tag{29}$$

Combining (29) with (28), we have that $0 \in \nabla_y h_{l,\tau}(x,y) + \partial P(y) \big|_{(x,y)=(x^*,x^*)}$. Therefore, $x^* \in \arg\min_{y \in \mathcal{X}} h_{l,\tau}(x^*,y)$, and thus

$$\min_{y \in \mathcal{X}} h_{l,\tau}(x^*,y) = h_{l,\tau}(x^*,x^*) = \tau \ln\left(\sum_{m=1}^{M} e^{\frac{f_m(x^*) - f_m(x^*)}{\tau}}\right) = \tau \ln M. \tag{30}$$

By definition, $v_{l,\tau}(x^*)$ can be computed by

$$v_{l,\tau}(x^*) = -\min_{y \in \mathcal{X}} h_{l,\tau}(x^*,y) = -\tau \ln M \tag{31}$$

which completes the proof. $\qquad\square$

## B.2. Continuity of the merit function

**Lemma B.6** (Continuity of $h_{l,\tau}$, $v_{l,\tau}$, and $p$). *1) If $f_m(x)$ is continuous for all $m \in [M]$, then the merit function $v_{l,\tau}(x)$ is lower semi-continuous.*
*2) Given a bounded set $\mathcal{X}_C$, suppose $f_m$ is $\ell_f$-Lipschitz continuous on $\mathcal{X}_C$ for $m = 0, \ldots, M$. Let $\ell_x = \sup_{x \in \mathcal{X}_C} \|x\|$. Then $h_{l,\tau}(x, y)$ is $(\ell_f + 2l\ell_x)$-Lipschitz continuous w.r.t. both $x \in \mathcal{X}_C$ and $y \in \mathcal{X}_C$, $v_{l,\tau}(x)$ is $\ell_{v_{l,\tau}}$-Lipschitz continuous on $\mathcal{X}_C$ with $\ell_{v_{l,\tau}} = \ell_f + 2l\ell_x$, and for $\theta \geq 1$, $p(x)$ is $\ell_p$-Lipschitz continuous on $\mathcal{X}_C$ with $\ell_p = \theta(2\ell_x)^{\theta-1}\ell_{v_{l,\tau}}^{\theta}$.*

*Proof of Lemma B.6. Proof of 1).* Recall that $v_{l,\tau}(x) = -\min_{y \in \mathcal{X}} h_{l,\tau}(x, y)$, and $h_{l,\tau}(x, y) = \mathrm{LSE}_\tau\big(f_m(y) - f_m(x)\big) + \frac{l}{2}\|x - y\|^2$. Since $f_m(x)$ is continuous for all $m \in [M]$, and the LSE function is continuous, we have $h_{l,\tau}(x, y)$ is continuous w.r.t. $x$ and $y$.

For any sequence $\{x_t\} \subseteq \mathcal{X}$ satisfying $\lim_{t \to \infty} x_t = \bar{x} \in \mathcal{X}$, given any $\epsilon > 0$, let $\bar{y} \in \mathcal{X}$ satisfy $h_{l,\tau}(\bar{x}, \bar{y}) \leq \min_{y \in \mathcal{X}} h_{l,\tau}(\bar{x}, y) + \epsilon$. As $h_{l,\tau}$ is continuous at $(\bar{x}, \bar{y})$, there exists $T > 0$ such that

$$\min_{y \in \mathcal{X}} h_{l,\tau}(x_t, y) \leq h_{l,\tau}(x_t, \bar{y}) \leq h_{l,\tau}(\bar{x}, \bar{y}) + \epsilon \leq \min_{y \in \mathcal{X}} h_{l,\tau}(\bar{x}, y) + 2\epsilon, \quad \forall t > T, \tag{32}$$

and thus

$$\limsup_{t \to \infty} \min_{y \in \mathcal{X}} h_{l,\tau}(x_t, y) \leq \min_{y \in \mathcal{X}} h_{l,\tau}(\bar{x}, y) + 2\epsilon. \tag{33}$$

As the above inequality holds for any $\epsilon > 0$, we obtain,

$$\limsup_{t \to \infty} \min_{y \in \mathcal{X}} h_{l,\tau}(x_t, y) \leq \min_{y \in \mathcal{X}} h_{l,\tau}(\bar{x}, y) \tag{34}$$

which proves that $v_{l,\tau}(x)$ is lower semi-continuous.

*Proof of 2).* We prove the Lipschitz continuity of $h_{l,\tau}(x, y)$ below. We define

$$\pi_m(x, y) := \frac{e^{\frac{1}{\tau}(f_m(y) - f_m(x))}}{\sum_{m=1}^{M} e^{\frac{1}{\tau}(f_m(y) - f_m(x))}}. \tag{35}$$

Note that

$$\|\nabla_x h_{l,\tau}(x, y)\| \leq \left\| \sum_{m=1}^{M} \pi_m(x, y) \nabla f_m(x) \right\| + \|l(x - y)\| \leq \ell_f + l(\|x\| + \|y\|) \leq \ell_f + 2l\ell_x, \tag{36}$$

$$\|\nabla_y h_{l,\tau}(x, y)\| \leq \left\| \sum_{m=1}^{M} \pi_m(x, y) \nabla f_m(y) \right\| + \|l(x - y)\| \leq \ell_f + l(\|x\| + \|y\|) \leq \ell_f + 2l\ell_x. \tag{37}$$

Therefore, $h_{l,\tau}(x, y)$ is $(\ell_f + 2l\ell_x)$-Lipschitz continuous w.r.t. both $x \in \mathcal{X}_C$ and $y \in \mathcal{X}_C$.

Next we prove the Lipschitz continuity of the merit function $v_{l,\tau}$ under additional assumptions. From Assumption 1, the functions $f_m(x), m = 0, \ldots, M$ are $\ell_f$-Lipschitz on a bounded set $\mathcal{X}_C$ where $\|x\| \leq \ell_x$. From (5), we can compute $\nabla v_{l,\tau}(x)$. We then derive the bound of $\|\nabla v_{l,\tau}(x)\|$ below.

$$\begin{aligned} \|\nabla v_{l,\tau}(x)\| &= \left\| \sum_{m=1}^{M} \pi_m(x) \nabla f_m(x) - l(x - y^*_{l,\tau}(x)) \right\| \\ &\leq \ell_f + l\|x\| + l\|y^*_{l,\tau}(x)\| \leq \ell_f + 2l\ell_x \end{aligned} \tag{38}$$

which proves that $v_{l,\tau}(x)$ is $(\ell_f + 2l\ell_x)$-Lipschitz continuous on $\mathcal{X}$.

Recall that $p(x) = \big(v_{l,\tau}(x) + \tau \ln M\big)^{\theta}$. For $\theta \geq 1$, the gradient of $p(x)$ is given by

$$\nabla p(x) = \theta\big(v_{l,\tau}(x) + \tau \ln M\big)^{\theta-1} \nabla v_{l,\tau}(x) \tag{39}$$

Note that $v_{l,\tau}(x) + \tau \ln M$ is bounded on a compact set since $v_{l,\tau}(x)$ is Lipschitz on this set, i.e.,

$$v_{l,\tau}(x) + \tau \ln M \leq \ell_{v_{l,\tau}} \|x - x^*\| \leq 2\ell_{v_{l,\tau}} \ell_x \quad \text{with} \quad \ell_{v_{l,\tau}} = \ell_f + 2l\ell_x. \tag{40}$$

Then $\|\nabla p(x)\|$ can be bounded by

$$\|\nabla p(x)\| \leq \theta \ell_{v_{l,\tau}} \big(v_{l,\tau}(x) + \tau \ln M\big)^{\theta - 1} \leq \theta \ell_{v_{l,\tau}} \big(v_{l,\tau}(x) + \tau \ln M\big)^{\theta - 1} \tag{41}$$

$$\leq \theta \ell_{v_{l,\tau}} \big(2\ell_{v_{l,\tau}} \ell_x\big)^{\theta - 1} = \theta \big(2\ell_x\big)^{\theta - 1} \ell_{v_{l,\tau}}^{\theta} \tag{42}$$

which proves that $p(x)$ is $\Big(\theta \big(2\ell_x\big)^{\theta - 1} \ell_{v_{l,\tau}}^{\theta}\Big)$-Lipschitz continuous on $\mathcal{X}$. $\qquad\square$

**Corollary B.7** (Lipschitz continuity of $\varphi_\gamma$). *Under the same settings as Lemma E.1, let $\ell_x = \sup_{x \in \mathcal{X}_C} \|x\|$. Given $\gamma > 0$, $\varphi_\gamma(x)$ is $\big(\gamma \ell_p + \ell_f\big)$-Lipschitz continuous on $\mathcal{X}_C$.*

*Proof of Corollary B.7.* Recall that $\varphi_\gamma(x) = f_0(x) + \gamma p(x)$. The proof directly follows by applying the $\ell_p$-Lipschitz continuity of $p$ from Lemma B.6, and the $\ell_f$-Lipschitz continuity of $f_0$ from Assumption 1. $\qquad\square$

**Lemma B.8** (Lipschitz continuity of $y_{l,\tau}^*(x)$). *Under the same settings as Lemma E.1, recall that*

$$y_{l,\tau}^*(x) := \arg\min_{y \in \mathcal{X}} h_{l,\tau}(x, y) = \arg\min_{y \in \mathcal{X}} \Big\{ \tau \ln \Big( \sum_{m=1}^{M} e^{\frac{f_m(y) - f_m(x)}{\tau}} \Big) + \frac{l}{2} \|x - y\|^2 \Big\}. \tag{43}$$

*For $l - \ell_{f,1} \geq \mu_{h_y} > 0$, there exists $\ell_{y_{l,\tau}^*} = \frac{2M\ell_f}{\tau}\Big(\frac{\ell_f^2}{\tau} + \ell_{f,1}\Big) + \frac{4M\ell_f^3}{\tau^2} > 0$ that for all $x, x' \in \mathcal{X}_C$, the following holds*

$$\|y_{l,\tau}^*(x) - y_{l,\tau}^*(x')\| \leq \ell_{y_{l,\tau}^*} \|x - x'\|. \tag{44}$$

*Proof.* By Corollary B.4, for $l + \min_{m \in [M]} \mu_m \geq \mu_{h_y} > 0$, the function $h_{l,\tau}(x, y)$ is $\mu_{h_y}$-strongly convex w.r.t. $y$. Therefore, from (Dontchev & Rockafellar, 2009, Theorem 2F.7), or using similar arguments for the proof in (Chen et al., 2023, Lemma 15), we can derive that

$$\|y_{l,\tau}^*(x) - y_{l,\tau}^*(x')\| \leq \mu_{h_y}^{-1} \|\nabla_{yy}^2 h_{l,\tau}(x, y) - \nabla_{yy}^2 h_{l,\tau}(x', y)\|. \tag{45}$$

Let $I_q \in \mathbb{R}^{q \times q}$ denote the identity matrix, then $\nabla_{yy}^2 h_{l,\tau}(x, y)$ can be further computed by

$$\nabla_{yy}^2 h_{l,\tau}(x, y) = \nabla_y \Bigg( \sum_{m=1}^{M} \underbrace{\frac{e^{\frac{f_m(y) - f_m(x)}{\tau}}}{\sum_{m=1}^{M} e^{\frac{f_m(y) - f_m(x)}{\tau}}}}_{\pi_m(x,y)} \nabla f_m(y) + l(y - x) \Bigg)$$

$$= \nabla_y \Big( \underbrace{\nabla F(y) \pi(x, y)}_{S(x,y)} + l(y - x) \Big) \quad \text{with} \quad \pi(x, y) = [\pi_1(x, y), \ldots, \pi_M(x, y)]^\top$$

$$= \frac{1}{\tau} \sum_{m=1}^{M} \pi_m(x, y) \nabla f_m(y) \nabla f_m(y)^\top - \frac{1}{\tau} S(x, y) S(x, y)^\top + \sum_{m=1}^{M} \pi_m(x, y) \nabla^2 f_m(y) + l I_q. \tag{46}$$

We first bound $\|S(x, y) S(x, y)^\top - S(x', y) S(x', y)^\top\|$ by

$$\|S(x, y) S(x, y)^\top - S(x', y) S(x', y)^\top\| \leq \Big( \|S(x, y)\| + \|S(x', y)\| \Big) \|S(x, y) - S(x', y)\|. \tag{47}$$

Then from Assumptions 4 and 1, we can bound $\|\nabla_{yy}^2 h_{l,\tau}(x, y) - \nabla_{yy}^2 h_{l,\tau}(x', y)\|$ by

$$\|\nabla_{yy}^2 h_{l,\tau}(x, y) - \nabla_{yy}^2 h_{l,\tau}(x', y)\|$$

$$\leq \frac{1}{\tau} \sum_{m=1}^{M} \|\pi_m(x, y) - \pi_m(x', y)\| \|\nabla f_m(y)\|^2 + \frac{1}{\tau} \Big( \|S(x, y)\| + \|S(x', y)\| \Big) \|S(x, y) - S(x', y)\|$$

$$+ \sum_{m=1}^{M} \|\pi_m(x, y) - \pi_m(x', y)\| \|\nabla^2 f_m(y)\|$$

$$\leq \sum_{m=1}^{M} \|\pi_m(x, y) - \pi_m(x', y)\| \Big(\frac{\ell_f^2}{\tau} + \ell_{f,1}\Big) + \frac{2\ell_f}{\tau} \|S(x, y) - S(x', y)\| \tag{48}$$

where $\|\pi_m(x, y) - \pi_m(x', y)\|$ can be further bounded by

$$\|\pi_m(x, y) - \pi_m(x', y)\| \leq \frac{2\ell_f}{\tau} \|x - x'\|. \tag{49}$$

Similarly, $\|S(x, y) - S(x', y)\|$ can be further bounded by

$$\|S(x, y) - S(x', y)\| \leq \sum_{m=1}^{M} \big(\pi_m(x, y) - \pi_m(x', y)\big) \nabla f_m(y) \leq \frac{2M\ell_f^2}{\tau} \|x - x'\|. \tag{50}$$

The proof is complete with $\ell_{y_{l,\tau}^*} = \frac{2M\ell_f}{\tau}\Big(\frac{\ell_f^2}{\tau} + \ell_{f,1}\Big) + \frac{4M\ell_f^3}{\tau^2}$. $\qquad\square$

### B.3. Proof of Proposition 3.3: relations of $v_{l,\tau}$ and weak Pareto optimality

*Proof of Proposition 3.3.* We prove each property as follows.

*Property 1.* For the first argument, by the property of the Log-sum-exp function (Nesterov, 2005), and since taking $\min_{y \in \mathcal{X}}$ preserves inequality, we have that

$$u_l(x) - \tau \ln M \leq v_{l,\tau}(x) \leq u_l(x). \tag{51}$$

This implies that, as $\tau \downarrow 0$, $v_{l,\tau}(x)$ uniformly converges to $u_l(x)$. Also recall from Lemma B.1 that $x$ is weakly Pareto optimal if and only if $u_l = 0$. Therefore, $x$ is weakly Pareto optimal if and only if $\lim_{\tau \downarrow 0} v_{l,\tau}(x) = 0$. The first argument is proved.

For the second argument, from (Nesterov, 2005), we have that

$$\tau \ln \Big( \sum_{m=1}^{M} e^{\frac{f_m(y) - f_m(x)}{\tau}} \Big) + \frac{l}{2} \|x - y\|^2 \leq \tau \ln M + \max_{m \in [M]} \{f_m(y) - f_m(x)\} + \frac{l}{2} \|x - y\|^2. \tag{52}$$

Since taking $\min_{y \in \mathcal{X}}$ preserves inequality, it implies that

$$\min_{y \in \mathcal{X}} \Big\{ \tau \ln \Big( \sum_{m=1}^{M} e^{\frac{f_m(y) - f_m(x)}{\tau}} \Big) + \frac{l}{2} \|x - y\|^2 \Big\} \leq \tau \ln M + \min_{y \in \mathcal{X}} \Big\{ \max_{m \in [M]} \{f_m(y) - f_m(x)\} + \frac{l}{2} \|x - y\|^2 \Big\} \tag{53}$$

which proves that

$$v_{l,\tau}(x) \geq - \min_{y \in \mathcal{X}} \Big\{ \max_{m \in [M]} \{f_m(y) - f_m(x)\} + \frac{l}{2} \|x - y\|^2 \Big\} - \tau \ln M$$

$$= u_l(x) - \tau \ln M \geq -\tau \ln M \tag{54}$$

where the last inequality holds because $u_l(x) \geq 0$. Furthermore, there exists $x \in \mathcal{X}$ such that $v_{l,\tau}(x) = -\tau \ln M$ by Lemma B.5. Then $\min_{x \in \mathcal{X}} v_{l,\tau}(x) = -\tau \ln M$.

*Property 2.* For the first argument, by Lemma B.1, if $x$ is weakly Pareto optimal, then $u_l(x) = 0$. Furthermore, by Property-1, $u_l(x) \geq v_{l,\tau}(x)$, which proves $v_{l,\tau}(x) \leq 0$.

Conversely, for the second argument, *for condition a)*, $l = 0$, $v_{0,\tau}(x) \leq 0$ implies that

$$\bar{u}(x) \leq v_{0,\tau}(x) + \tau \ln M \leq \tau \ln M. \tag{55}$$

By the definition of $\bar{u}$, for all $z \in \mathcal{X}$, it holds that

$$\min_{m \in [M]} \{f_m(x) - f_m(z)\} \le \bar{u}(x) \le \tau \ln M. \tag{56}$$

In other words, there exists no $z \in \mathcal{X}$ and $z \neq x$ such that, $F(z) < F(x) - \tau \ln M$.

*For condition b)*, $l > 0$, $v_{l,\tau}(x) \le -\tau \ln M$ implies that

$$0 \le u_l(x) \le v_{l,\tau}(x) + \tau \ln M \le 0. \tag{57}$$

By the definition of $u_l(x)$, it implies

$$\min_{m \in [M]} \{f_m(x) - f_m(y)\} - \frac{l}{2}\|x - y\|^2 \le 0 \tag{58}$$

By the convexity of $\mathcal{X}$, take $z \in \mathcal{X}$, and $t \in (0, 1)$, then $(1 - t)x + tz \in \mathcal{X}$. Let $y = (1 - t)x + tz$, and plug it into the above inequality, we have

$$\min_{m \in [M]} \{f_m(x) - f_m((1 - t)x + tz)\} - \frac{l}{2}\|x - y\|^2 \le 0 \tag{59}$$

By the $(1, 0)$-point-quasar convexity of $f_m$ at $x$ for all $m \in [M]$, $f_m((1 - t)x + tz) \le t f_m(z) + (1 - t) f_m(x)$. Therefore,

$$\min_{m \in [M]} \{t(f_m(x) - f_m(z))\} - \frac{l}{2}\|t(z - x)\|^2 \le 0. \tag{60}$$

Dividing both sides by $t$ and letting $t \downarrow 0$, we have that for all $z \in \mathcal{X}$,

$$\min_{m \in [M]} \{f_m(x) - f_m(z)\} \le 0. \tag{61}$$

This proves $x$ is weakly Pareto optimal.

The proof of the properties of the smoothed merit function is complete. $\square$

### B.4. Examples of point quasar convex functions

By definition, $\mu$-(strongly) convex functions are $(1, \mu)$-(strongly) point quasar convex everywhere. This quasar convexity property is also closely related to star convexity, restricted secant condition, variational coherence, PL condition, invexity, etc. For a more detailed discussion and more examples, refer to e.g., (Hinder et al., 2020, Appendix A, D.2, D.3).

B.4.1. EXAMPLES CONDITION 2-B) IN PROPOSITION 3.3

**Example B.9.** For all $m \in [M]$, $f_m$ are strongly convex on $\mathcal{X}$.

Example B.9 is the simplest case covered by our method. But existing works which focus on such cases usually require second-order information in the algorithm, as summarized in Table 2.

**Example B.10.** For $x = [x_1; x_2] \in \mathbb{R}^2$, let $\mathcal{X} = \mathbb{R}^2$. Define $f_1(x) = x_1^2 x_2^2$, $f_2(x) = x_1^4 x_2^4$.

In Example B.10, $f_1, f_2$ are nonconvex but star-convex (Hinder et al., 2020) at $x^* = [0; 0]$. Furthermore, $x^*$ satisfies that $v_{l,\tau}(x^*) = -\tau \ln M$. And both $f_1, f_2$ satisfies the $(1, 0)$-point quasar-convexity at $x^*$ within $\mathcal{X}$.

### B.5. Proof of gradient of the smoothed merit function

**Lemma B.11** (Gradient and directional derivative of $v_{l,\tau}$). *The gradient of $v_{l,\tau}$ can be computed by*

$$\nabla v_{l,\tau}(x) = \sum_{m=1}^{M} \pi_m(x) \nabla f_m(x) - l(x - y_{l,\tau}^*(x)), \ \ with \ \ \pi_m(x) := \frac{e^{\frac{1}{\tau}(f_m(y_{l,\tau}^*(x)) - f_m(x))}}{\sum_{m=1}^{M} e^{\frac{1}{\tau}(f_m(y_{l,\tau}^*(x)) - f_m(x))}}. \tag{62}$$

*For all $z \in \mathcal{X}$, the directional derivative of $v_{l,\tau}$, denoted as $v'_{l,\tau}(x; z - x)$, can be computed by*

$$v'_{l,\tau}(x; z - x) = \sum_{m=1}^{M} \pi_m(x, y^*_{l,\tau}(x)) f'_m(x; z - x) - l\left(x - y^*_{l,\tau}(x)\right)^\top (z - x). \tag{63}$$

*Proof of Lemma B.11.* Recall that we have defined

$$h_{l,\tau}(x, y) = \tau \ln \left( \sum_{m=1}^{M} e^{\frac{f_m(y) - f_m(x)}{\tau}} \right) + \frac{l}{2}\|x - y\|^2. \tag{64}$$

By definition, $v_{l,\tau}(x) = -\min_{y \in \mathcal{X}} h_{l,\tau}(x, y) = -h_{l,\tau}(x, y^*_{l,\tau}(x))$, with $y^*_{l,\tau}(x) \in \arg\min_{y \in \mathcal{X}} h_{l,\tau}(x, y)$. If $l + \min_{m \in [M]} \mu_m \geq c > 0$, by Corollary B.4, $y^*_{l,\tau}(x)$ is unique. Furthermore, $y^*_{l,\tau}(x)$ is continuous w.r.t. $x$.

By the extended Danskin-type theorem in e.g., (Shen et al., 2025, Proposition 5), $v_{l,\tau}(x)$ is differentiable. Its gradient can be computed by

$$\begin{aligned}
\nabla v_{l,\tau}(x) &= -\nabla_x h_{l,\tau}(x, y^*_{l,\tau}(x)) \\
&= \sum_{m=1}^{M} \frac{e^{\frac{f_m(y^*_{l,\tau}(x)) - f_m(x)}{\tau}}}{\sum_{m=1}^{M} e^{\frac{f_m(y^*_{l,\tau}(x)) - f_m(x)}{\tau}}} \nabla f_m(x) - l(x - y^*_{l,\tau}(x)).
\end{aligned} \tag{65}$$

Then given all $x, z \in \mathcal{X}$, the directional derivative of $v_{l,\tau}(x)$ can be computed by

$$v'_{l,\tau}(x; z - x) = \sum_{m=1}^{M} \pi_m(x, y^*_{l,\tau}(x)) f'_m(x; z - x) - l\left(x - y^*_{l,\tau}(x)\right)^\top (z - x) \quad \text{for all } z \in \mathcal{X} \tag{66}$$

where $\pi_m(x, y) = \frac{e^{\frac{f_m(y) - f_m(x)}{\tau}}}{\sum_{m=1}^{M} e^{\frac{f_m(y) - f_m(x)}{\tau}}}$. The proof is complete. $\qquad\square$

## C. Subanalyticity and related properties

In this section, we first discuss some preliminaries on subanalyticity, and then prove the global subanalyticity of the merit function $v_{l,\tau}(x)$.

**Definition C.1** (Subanalyticity (Bierstone & Milman, 1988))**.** 1) A subset $S \subset \mathbb{R}^q$ is called *semianalytic* if each point of $\mathbb{R}^q$ admits a neighborhood $V$ for which $S \cap V$ assumes the following form

$$\bigcup_{i=1}^{I} \bigcap_{j=1}^{J} \{x \in V : f_{ij}(x) = 0, g_{ij}(x) > 0\}, \tag{67}$$

where $f_{ij}, g_{ij} : V \to \mathbb{R}$ are real analytic functions for $1 \leq i \leq I, 1 \leq j \leq J$.
2) A subset $S \subset \mathbb{R}^q$ is called *subanalytic* if each point of $\mathbb{R}^q$ admits a neighborhood $V$ such that

$$S \cap V = \{x \in \mathbb{R}^q \mid (x, y) \in B\} \tag{68}$$

where $B$ is a bounded semianalytic subset of $\mathbb{R}^q \times \mathbb{R}^m$.
3) A function $f : \mathbb{R}^q \to \mathbb{R} \cup \{+\infty\}$ is called *subanalytic* if its graph is a subanalytic subset of $\mathbb{R}^q \times \mathbb{R}$.

**Definition C.2** (Global subanalyticity (Dries & Miller, 1996, p. 506))**.** Let $x = [x_1, \ldots, x_q]^\top \in \mathbb{R}^q$. Define the function

$$\Phi_q(x) := \left( \frac{x_1}{1 + x_1^2}, \ldots, \frac{x_q}{1 + x_q^2} \right). \tag{69}$$

1) A subset $S$ of $\mathbb{R}^q$ is called *globally subanalytic* if its image under $\Phi_q$ is a subanalytic subset of $\mathbb{R}^q$.
2) A function $f : \mathbb{R}^q \to \mathbb{R} \cup \{+\infty\}$ is called *globally subanalytic* if its graph is a globally subanalytic subset of $\mathbb{R}^q \times \mathbb{R}$.

**Proposition C.3** ((Bolte et al., 2007)). *Globally subanalytic sets are subanalytic, and conversely, any bounded subanalytic set is globally subanalytic.*

**Lemma C.4** ((Dries & Miller, 1996)). *The image or the preimage of a globally subanalytic set by a globally subanalytic function (respectively, globally subanalytic multivalued operator) is globally subanalytic.*

**Lemma C.5** (Projection theorem (Dries & Miller, 1996)). *Let $\Pi(x_1, \ldots, x_{n+1}) = (x_1, \ldots, x_n)$ be the canonical projection from $\mathbb{R}^{n+1}$ onto $\mathbb{R}^n$. If $S$ is a globally subanalytic subset of $\mathbb{R}^{n+1}$, then so is $\Pi(S)$ in $\mathbb{R}^n$.*

**Lemma C.6** (Lojasiewicz factorization lemma (Bierstone & Milman, 1988, Theorem 6.4)). *If $\mathcal{X}_{v_{l,\tau}}^* \coloneqq \arg\min_{x \in \mathcal{X}} v_{l,\tau}(x)$ is globally subanalytic, and $v_{l,\tau}(x)$ is continuous and globally subanalytic, then the $(\varrho, \eta)$-Hölderian error bound holds for $v_{l,\tau}(x)$ for some $\varrho, \eta > 0$.*

### C.1. Proof of global subanalyticity of $v_{l,\tau}$, $p$, and $u_l$

**Lemma C.7.** *Let $\mathcal{X}$ and $\mathcal{Y}$ be two bounded subanalytic subsets in $\mathbb{R}^q$. Let $f : \mathcal{X} \times \mathcal{Y} \to \mathbb{R}$ be a bounded subanalytic function. Then 1) the function $\phi$ below is bounded and subanalytic, thus globally subanalytic.*

$$\phi : \mathcal{X} \ni x \mapsto \max_{y \in \mathcal{Y}} f(x, y) \in \mathbb{R}. \tag{70}$$

*2) Given $x$, the solution set $\mathcal{Y}^*(x) \coloneqq \arg\max_{y \in \mathcal{Y}} f(x, y)$ is bounded and subanalytic, thus globally subanalytic.*

*Proof.* The proof mainly adopts the proof idea of (Kosiba, 2025, Lemma 4.18). The key difference is that instead of using local boundedness of the functions, we start from global boundedness of $f$ on $\mathcal{X}$ and derive the global boundedness of $\phi$, which, combined with subanalyticity, leads to the global subanalyticity of $\phi$ on $\mathcal{X}$.

Let $\mathrm{Gr}(\cdot)$ denote the graph of a function. Since $f$ is bounded, let $\overline{c_f}, \underline{c_f}$ denote its upper and lower bound, respectively. And define $\mathcal{W} \coloneqq \{w \in \mathbb{R} \mid \underline{c_f} \leq w \leq \overline{c_f}\}$. Then $\mathcal{W} \subset \mathbb{R}$ is bounded. Consider the following set

$$A = \{(x, y, z, w) \in \mathcal{X} \times \mathcal{Y} \times \mathbb{R} \times \mathcal{W} \mid (x, y, z) \in \mathrm{Gr}(f(x, y)), z \leq w\}. \tag{71}$$

Note that $x, y$ are bounded, and so is $z$ because of the boundedness of $f$. Furthermore, $w$ is bounded by definition. Thus $A$ is bounded. Also note that $A$ is subanalytic because $f$ is subanalytic, its graph is subanalytic, and the Cartesian product of two subanalytic sets is subanalytic. Therefore, $A$ is globally subanalytic.

Define the following projection $\Pi_A$ and the set $B$ by canonical projection of $A$.

$$\Pi_A : \mathcal{X} \times \mathcal{Y} \times \mathbb{R} \times \mathcal{W} \ni (x, y, z, w) \mapsto (x, y, w) \in \mathcal{X} \times \mathcal{Y} \times \mathcal{W}. \tag{72}$$

$$B \coloneqq \Pi_A(A) = \{(x, y, w) \in \mathcal{X} \times \mathcal{Y} \times \mathcal{W} \mid f(x, y) \leq w\}. \tag{73}$$

Then $B$ is globally subanalytic based on Lemma C.5.

We also define the following auxiliary sets

$$B_R = \{(x, y, w) \in \mathcal{X} \times \mathcal{Y} \times \mathcal{W}\}, \tag{74}$$

$$B_R \backslash B = \{(x, y, w) \in \mathcal{X} \times \mathcal{Y} \times \mathcal{W} \mid f(x, y) > w\}. \tag{75}$$

Then we define the projection $\Pi_B$ and define the following set $C$ by canonical projection of $B$.

$$\Pi_B : \mathcal{X} \times \mathcal{Y} \times \mathcal{W} \ni (x, y, w) \mapsto (x, w) \in \mathcal{X} \times \mathcal{W}, \tag{76}$$

$$C \coloneqq \Pi_B(B) \backslash \Pi_B(B_R \backslash B) = \{(x, w) \in \mathcal{X} \times \mathcal{W} \mid \sup_{y \in \mathcal{Y}} f(x, y) \leq w\}. \tag{77}$$

Since $C$ is bounded and subanalytic, it is globally subanalytic.

We further define the following subanalytic set and projection

$$D \coloneqq \{(x, w_1, w_2) \in \mathcal{X} \times \mathcal{W} \times \mathcal{W} \mid (x, w_1) \in C, (x, w_2) \in C, w_1 > w_2\}, \tag{78}$$

$$\Pi_D : \mathcal{X} \times \mathcal{W} \times \mathcal{W} \ni (x, w_1, w_2) \mapsto (x, w_1) \in \mathcal{X} \times \mathcal{W}. \tag{79}$$

Finally, observe that

$$C\backslash\Pi_D(D) \ni (x, w) \iff \text{there exists no } w' \text{ such that } (x, w'), (x, w) \in C \text{ and } w > w' \tag{80}$$

$$\iff \text{there exists no } w' \text{ such that } \sup_{y \in \mathcal{Y}} f(x, y) \leq w' \text{ and } \sup_{y \in \mathcal{Y}} f(x, y) \leq w \text{ and } w > w' \tag{81}$$

which means that $\mathrm{Gr}(\phi(x)) = C\backslash\Pi_D(D)$. Since $\mathrm{Gr}(\phi(x))$ is bounded and subanalytic, thus it is globally subanalytic. By definition, $\phi$ is also globally subanalytic.

Next we proceed to prove the global subanalyticity of $\mathcal{Y}^*(x)$ given $x$. Define the following set $E$ and projection $\Pi_E$

$$E := \{(x, y, w) \in \mathcal{X} \times \mathcal{Y} \times \mathcal{W} \mid (x, w) \in \mathrm{Gr}(\phi(x)), f(x, y) = w\}, \tag{82}$$

$$\Pi_E : \mathcal{X} \times \mathcal{Y} \times \mathcal{W} \ni (x, y, w) \mapsto (x, y) \in \mathcal{X} \times \mathcal{Y}. \tag{83}$$

Then $\Pi_E(E) = \{(x, y) \in \mathcal{X} \times \mathcal{Y} \mid y \in \mathcal{Y}^*(x)\}$ is globally subanalytic. Furthermore, given $x = c_x \in \mathcal{X}$ for any $c_x \in \mathcal{X}$, we have

$$\Pi_E(E) \cap \{(x, y) \in \mathcal{X} \times \mathcal{Y} \mid x = c_x\} = \{(x, y) \in \mathcal{X} \times \mathcal{Y} \mid y \in \mathcal{Y}^*(x), x = c_x\} \tag{84}$$

which is globally subanalytic. Taking projection $\Pi : \mathcal{X} \times \mathcal{Y} \ni (x, y) \mapsto y \in \mathcal{Y}$ yields that $\mathcal{Y}^*(x)$ given $x = c_x$ for any $c_x \in \mathcal{X}$ is also globally subanalytic. $\qquad\square$

*Proof of Lemma 3.5. Part 1:* By Assumption 3, $f_m$ is globally subanalytic for all $m = 0, \ldots, M$. By Definition C.1 and Lemma C.4, subanalyticity is preserved under the subanalytic LSE function.

Recall the definition of $h_{l,\tau}(x, y)$ in (25). By Assumption 1 and Lemma B.6, $h_{l,\tau}(x, y)$ is Lipschitz continuous w.r.t. both $x, y \in \mathcal{X}_C$, thus $h_{l,\tau}(x, y)$ is bounded for $(x, y) \in \mathcal{X}_C \times \mathcal{X}_C$. Combining the above arguments, $h_{l,\tau}(x, y)$ is bounded and subanalytic on $\mathcal{X}_C$, thus globally subanalytic on $\mathcal{X}_C$ by Proposition C.3.

*Part 2:* Next, we prove that $v_{l,\tau}(x) = -\min_{y \in \mathcal{X}} h_{l,\tau}(x, y)$ and $p(x) = (v_{l,\tau}(x) + \tau \ln M)^\theta$ are both globally subanalytic on $\mathcal{X}_C$. This directly follows by applying Lemma C.7-1).

*Part 3:* Finally, we prove that $\mathcal{X}^*_{v_{l,\tau}} = \arg\min_{x \in \mathcal{X}} v_{l,\tau}(x)$ is also globally subanalytic on $\mathcal{X}_C$. This directly follows by applying Lemma C.7-2).

Consequently, the merit function $v_{l,\tau}(x)$ satisfies the $(\varrho, \eta)$-Hölderian error bound for some $\varrho, \eta > 0$ on a bounded set $\mathcal{X}_C$ by Lemma C.6. The penalty function $p(x)$ also satisfies the $(\varrho_p, \eta_p)$-Hölderian error bound with $\varrho_p = \varrho^\theta$ and $\eta_p = \theta\eta$. $\qquad\square$

Following similar arguments as the above proof, we can obtain that the original merit function $u_l(x)$ is globally subanalytic on a bounded subanalytic set, and thus satisfies the HEB on the set. This result is formally stated below.

**Corollary C.8.** *Under Assumption 3, and that $\mathcal{X}$ is subanalytic, given a compact subanalytic set $\mathcal{X}_C$, suppose $f_m(x)$ is continuous and bounded on $\mathcal{X}_C \cap \mathcal{X}$ for all $m \in [M]$. Then both $\mathcal{X}^*_{v_{l,\tau}} \cap \mathcal{X}_C$ and $v_{l,\tau}(x)$ on $\mathcal{X}_C \cap \mathcal{X}$ are globally subanalytic. Consequently, the merit function $u_l(x)$ in (23) without smoothing satisfies the $(\varrho_u, \eta_u)$-HEB in Definition 3.4 on $\mathcal{X}_C \cap \mathcal{X}$, with some $\varrho_u, \eta_u > 0$.*

**Lemma C.9** (KL inequality (Kurdyka, 1998, Theorem 1)). *Let $f : \Omega \to \mathbb{R}$ be a subanalytic function which is differentiable in $\Omega \backslash f^{-1}(0)$, where $\Omega$ is an open bounded subset of $\mathbb{R}^q$. Then there exist $c > 0, \nu > 0$ and $\alpha > 1$ such that: $c\|\nabla f(x)\|^\alpha \geq |f(x)|$, for each $x \in \Omega$ such that $|f(x)| \in (0, \nu)$. If in addition $\lim_{x \to a} f(x) = 0$ for some $a \in \bar{\Omega}$, then the above inequality holds for each $x \in \Omega \backslash f^{-1}(0)$ close to $a$.*

## C.2. Examples of globally subanalytic functions

We summarize some commonly used globally subanalytic functions and their corresponding Hölderian error bound (HEB) in Table 6 below. The first few examples of convex functions in Table 6 are directly referenced from (Doron & Shtern, 2023, Table 2) and (Chen et al., 2024b, Table 2).

Below, we prove the HEB of the last four bounded functions in Table 6.

*Table 6.* Summary of some functions satisfying Hölderian error bound with corresponding exponents.

| Functions | Remarks | Names | $\eta$ | $\varrho$ |
|---|---|---|---|---|
| | | Convex functions | | |
| $\max_{i \in [m]}\{\langle a_i, x \rangle - b_i\}$ | $a_i \in \mathbb{R}^q, b \in \mathbb{R}^m$ | Polytope | 1 | $\max_{i \in [m]}\{\|a_i\|^{-1}\}$ |
| $\|x - x_0\|_Q = \sqrt{(x-x_0)^\top Q (x - x_0)}$ | $Q \in \mathbb{S}^q, Q \succ 0, x_0 \in \mathbb{R}^q$ | $Q$-norm (Ellipsoid) | 1 | $(\lambda_{\min}(Q))^{-\frac{1}{2}}$ |
| $\|x - x_0\|_p$ | $x_0 \in \mathbb{R}^q, p \geq 1$ | $\ell_p$-norm | 1 | 1 |
| $\frac{1}{m}\sum_{i=1}^m \log(1 + \exp(-a_i^\top x b_i))$ | $a_i \in \mathbb{R}^q, b \in \mathbb{R}^m, A \in \mathbb{R}^{m \times q}$ | Logistic loss | 2 | $(\lambda_{\min}(Q))^{-1}$, $Q$ is a function of $a_i, b$ |
| $f(x) + \frac{\sigma}{2}\|x\|^2$ | $f$ convex, $\sigma > 0$ | strongly-convex | 2 | $\frac{2}{\sigma}$ |
| | | Bounded and possibly nonconvex functions | | |
| $\sin(x)$ | $x \in [0, \frac{1}{2}\pi]$ | Sine | $\geq 1$ | $(\frac{\pi}{2})^\eta$ |
| $x^p$ | $x \in [0,1], p > 0$ | Polynomial | $p$ | 1 |
| $e^x$ | $x \in [0,1]$ | Exponential | $\geq 1$ | 1 |
| $\ln(x+1)$ | $x \in [0,1]$ | Logarithmic | $\geq 1$ | $\frac{1}{\ln(2)}$ |

*Proof of HEB of bounded functions in Table 6.* We prove the $(\varrho, \eta)$-Hölderian error bound (HEB) of the last four functions in Table 6 one by one as follows.

*1) Sine function.* For $x \in [0, \frac{1}{2}\pi]$, $x^* = 0$ is the unique minimizer of $\sin(x)$, thus for $\eta \geq 1$, $\left(\mathrm{dist}(x, \mathcal{X}^*_{\sin})\right)^\eta = |x|^\eta \leq (\frac{\pi}{2})^\eta \sin(x)$ for all $x \in [0, \frac{1}{2}\pi]$. Therefore, $\sin(x)$ satisfies $(\varrho, \eta)$-HEB for $x \in [0, \frac{1}{2}\pi]$ with $\eta \geq 1$ and $\varrho = (\frac{\pi}{2})^\eta$.

*2) Polynomial function.* For $x \in [0,1]$, $x^* = 0$ is the unique minimizer of $x^p$. Thus $\left(\mathrm{dist}(x, \mathcal{X}^*_{\sin})\right)^p = |x|^p \leq x^p$ for all $x \in [0,1]$. Therefore, $x^p$ satisfies $(\varrho, \eta)$-HEB for $x \in [0,1]$ with $\eta = p$ and $\varrho = 1$.

*3) Exponential function.* For $x \in [0,1]$, $x^* = 0$ is the unique minimizer of $e^x$. Thus $\left(\mathrm{dist}(x, \mathcal{X}^*_{\sin})\right)^\eta = |x|^\eta \leq e^x - e^0$ for all $x \in [0,1]$, and $\eta \geq 1$. Therefore, $e^x$ satisfies $(\varrho, \eta)$-HEB for $x \in [0,1]$ with $\eta \geq 1$ and $\varrho = 1$.

*4) Logarithmic function.* For $x \in [0,1]$, $x^* = 0$ is the unique minimizer of $\ln(x+1)$. Thus $\left(\mathrm{dist}(x, \mathcal{X}^*_{\sin})\right)^\eta = |x|^\eta \leq \frac{1}{\ln(2)}\ln(x+1)$ for all $x \in [0,1]$, and $\eta \geq 1$. Therefore, $e^x$ satisfies $(\varrho, \eta)$-HEB for $x \in [0,1]$ with $\eta \geq 1$ and $\varrho = \frac{1}{\ln(2)}$.

The proof is complete. $\square$

*Remark* C.10. Note that, even though we have shown in the above that the exponential function satisfies HEB in a compact and subanalytic set $x \in [0,1]$, it is known that the exponential function is not globally subanalytic on $\mathbb{R}$. In the following proofs which require the HEB property, it suffices to show the global subanalyticity holds on a compact and subanalytic set constructed from the corresponding problem, which leads to the HEB property on the compact and subanalytic set. And HEB of $p(x)$ on a compact and subanalytic set within $\mathcal{X}$ is sufficient to show the relations of global/local/stationary solutions of the penalty problem to the bilevel problem as long as there exists bounded solutions to $\min_{x \in \mathcal{X}} p(x)$, even if $\mathcal{X}$ is not bounded.

## C.3. Relations of Hölderian error bound, quadratic growth, proximal PL, and proximal error bound

We further discuss the relations of Hölderian error bound (HEB), also known as Hölderian growth, with other commonly used conditions such as quadratic growth (QG), proximal error bound (EB), proximal PL inequality, and strong convexity (SC). Below, we consider a general function for the discussion.

$$\phi(x) \coloneqq f(x) + g(x) \tag{85}$$

with $f$ smooth and $g$ convex. In our specific problem, $g$ is an indicator function on $\mathcal{X}$. Let $\mathcal{X}^*_\phi$ denote the set of minimizers for $\phi$. We first review the formal definitions of the above conditions, and then discuss their relations.

**Definition C.11** (Proximal error bound). The function $\phi$ in (85) satisfies proximal error bound if for $t > 0$, there exists $c_\phi > 0$ that

$$\mathrm{dist}(x, \mathcal{X}^*_\phi) \leq c_\phi t^{-1}\|x - \mathrm{prox}_{tg}(x - t\nabla f(x))\|. \tag{86}$$

**Definition C.12** (Proximal PL). The function $\phi$ in (85) satisfies proximal PL inequality if for $t > 0$, there exists $c > 0$ that

$$\phi(x) - \phi(x^*) \leq c\mathcal{D}_g(x, t) \text{ with } x^* \in \mathcal{X}^*_\phi, \text{ and}$$

$$\mathcal{D}_g(x,t) := -2t \min_y \left[ \langle \nabla f(x), y - x \rangle + \frac{t}{2} \|y - x\|^2 + g(y) - g(x) \right]. \tag{87}$$

By definition, QG is a special case of HEB with the exponent $\eta = 2$. And similarly, proximal KL generalizes the concept of proximal PL. The relations of SC, proximal EB, proximal PL, QG have been studied in existing literature. We summarize these relations using the equation below.

$$
\begin{array}{ccccc}
f \text{ is SC} & \overset{(a)}{\Longrightarrow} & \text{proximal EB} & \overset{(b)}{\Longleftrightarrow} & \text{proximal PL} & \overset{(c)}{\Longrightarrow} & \text{QG} \\
& & & & \Downarrow & & \Downarrow \\
& & \text{proximal EB} & \Longleftrightarrow & \text{proximal KL} & \Longrightarrow & \text{HEB}
\end{array} \tag{88}
$$

where $(a)$ has been proved in e.g., (Karimi et al., 2016, Appendix F-2); $(b)$ has been proved in e.g., (Karimi et al., 2016, Appendix G); $(c)$ has been proved in e.g., (Liao et al., 2024, Theorem 3.1) with additional conditions that $\phi$ is closed and weakly convex, or in (Karimi et al., 2016, Theorem 2) with additional conditions that $g$ is a constant.

Furthermore, when the function $f$ is convex, we have that

$$
\begin{array}{ccc}
\text{proximal EB/PL} & \overset{(d)}{\Longleftrightarrow} & \text{QG} \\
\Downarrow & & \Downarrow \\
\text{proximal EB/KL} & \overset{(f)}{\Longleftrightarrow} & \text{HEB}
\end{array} \tag{89}
$$

where $(d)$ has been proved in e.g., (Drusvyatskiy & Lewis, 2018, Corollary 3.6); $(f)$ has been proved in e.g., (Bolte et al., 2016, Theorem 5). In this paper, we provide a proof of the relations between KL, HEB, and EB in (88) for nonconvex $f$ and constant $g$ in Lemma C.13 and Lemma C.14 below.

**Lemma C.13** (KL implies HEB). *Consider the function $\phi$ in (85) with $g(x) = 0$, and $\min \phi(x) = 0$. If $\phi$ satisfies the $(c_\phi, \alpha_\phi)$-KL inequality on $\Omega$ with exponent $\alpha_\phi > 1$, then it also satisfies the $(\varrho_\phi, \eta_\phi)$-HEB on $\Omega$ with exponent $\eta_\phi = \frac{\alpha_\phi}{\alpha_\phi - 1}$, and $\varrho_\phi = \left(1 - \frac{1}{\alpha_\phi}\right)^{-\frac{\alpha_\phi}{\alpha_\phi - 1}} \left(\frac{1}{c_\phi}\right)^{-\frac{1}{\alpha_\phi - 1}}$.*

*Proof of Lemma C.13.* We first define an auxiliary function

$$L(x) := \left(\phi(x)\right)^{1 - \frac{1}{\alpha_\phi}}. \tag{90}$$

Since $\phi$ satisfies the KL inequality, then for any $x \notin \mathcal{X}_\phi^*$, and thus $\phi(x) \neq 0$, we have

$$\|\nabla L(x)\|^2 = \left(1 - \frac{1}{\alpha_\phi}\right)^2 \left\| \frac{\nabla \phi(x)}{(\phi(x))^{\frac{1}{\alpha_\phi}}} \right\|^2 = \left(1 - \frac{1}{\alpha_\phi}\right)^2 \frac{\|\nabla \phi(x)\|^2}{(\phi(x))^{\frac{2}{\alpha_\phi}}} \geq \underbrace{\left(1 - \frac{1}{\alpha_\phi}\right)^2 \left(\frac{1}{c_\phi}\right)^{\frac{2}{\alpha_\phi}}}_{\mu_L}. \tag{91}$$

Also $\phi$ satisfies the KL inequality implies that $\phi$ is an invex function and thus $L$ is a non-negative invex function with a closed optimal solution set and zero optimal value. or any point $x_0 \notin \mathcal{X}_L^*$, consider solving the following differential equation for $x(t) \notin \mathcal{X}_\phi^*$:

$$\frac{dx(t)}{dt} = -\nabla L(x(t)) \tag{92}$$

$$x(t = 0) = x_0 \tag{93}$$

Following similar arguments for proving PL implies QG, see e.g., (Karimi et al., 2016, Theorem 2, Appendix A), there exists a $T$ such that $x(T) \in \mathcal{X}_L^*$ (and at this point the differential equation ceases to be defined). Then

$$L(x_0) - L(x_t) = \int_{x_t}^{x_0} \langle \nabla L(x), dx \rangle = -\int_{x_0}^{x_t} \langle \nabla L(x), dx \rangle = -\int_0^T \langle \nabla L(x(t)), \frac{dx(t)}{dt} \rangle dt \tag{94}$$

$$= \int_0^T \|\nabla L(x(t))\|^2 dt \geq \int_0^T \mu_L dt = \mu_L T. \tag{95}$$

As $L(x_t) \geq 0$, this shows we need to have $T \leq L(x_0)/\mu_L$, so there must be a $T$ with $x(T) \in \mathcal{X}_L^*$. The length of the orbit $x(t)$ starting at $x_0$, which we'll denote by $D(x_0)$, is given by

$$D(x_0) = \int_0^T \|dx(t)/dt\|dt = \int_0^T \|\nabla L(x(t))\|dt \geq \|x_0 - x_p\| \tag{96}$$

where $x_p$ is the projection of $x_0$ onto $\mathcal{X}_L^*$ and the inequality follows because the orbit is a path from $x_0$ to a point in $\mathcal{X}_L^*$ (and thus it must be at least as long as the projection distance).

Then we can further bound

$$L(x_0) = L(x_0) - L(x_T) \stackrel{(95)}{=} \int_0^T \|\nabla L(x(t))\|^2 dt \stackrel{(91)}{\geq} \sqrt{\mu_L} \int_0^T \|\nabla L(x(t))\|dt \stackrel{(96)}{\geq} \sqrt{\mu_L}\|x_0 - x_p\|. \tag{97}$$

The proof is complete. $\qquad\square$

**Lemma C.14** (KL and HEB imply EB). *For $\phi(x)$ in (85) with $g(x) = 0$ that satisfies the KL inequality with exponent $\alpha_\phi$, and the HEB with exponent $\eta_\phi$, i.e.,*

$$\text{KL: } c_\phi\|\nabla\phi(x)\|^{\alpha_\phi} \geq \phi(x), \quad \text{HEB: } \phi(x) \geq \varrho_\phi^{-1}\big(\text{dist}(x, \mathcal{X}_\phi^*)\big)^{\eta_\phi} \tag{98}$$

*Then it holds that*

$$\|\nabla\phi(x)\| \geq \varrho_h^{-1}\big(\text{dist}(x, \mathcal{X}_\phi^*)\big)^{\eta_h} \ \text{ with } \eta_h = \frac{\eta_\phi}{\alpha_\phi}, \text{ and } \varrho_h = \big(\varrho_\phi c_\phi\big)^{\frac{1}{\alpha_\phi}}. \tag{99}$$

*Proof of Lemma C.14.* The proof directly follows from combining the two inequalities from KL and HEB that

$$c_\phi\|\nabla\phi(x)\|^{\alpha_\phi} \geq \phi(x) \geq \varrho_\phi^{-1}\big(\text{dist}(x, \mathcal{X}_\phi^*)\big)^{\eta_\phi}. \tag{100}$$

Rearranging the above inequalities proves the result. $\qquad\square$

# D. Proof of relations of different formulations

In this section, we prove the relations of the solutions of the bilevel problem and the penalty reformulation. Recall that we denote the penalty function with exponent $\theta$ as

$$p(x) = \big(v_{l,\tau}(x) + \tau \ln M\big)^\theta. \tag{101}$$

Recall that we let $\mathcal{X}_{v_{l,\tau}}^* := \arg\min_{x\in\mathcal{X}} v_{l,\tau}(x)$. Similarly, we define $\mathcal{X}_{\varphi_\gamma}^* := \arg\min_{x\in\mathcal{X}} \varphi_\gamma(x)$. For $\epsilon \geq 0$, define the $\epsilon$-approximate solution set (level set) below

$$\mathcal{X}_\epsilon := \{x \in \mathcal{X} \mid p(x) \leq \epsilon\}. \tag{102}$$

Then $\mathcal{X}_{v_{l,\tau}}^* = \mathcal{X}_0 \subseteq \mathcal{X}_\epsilon$. We use $\mathcal{X}_C$ to denote a compact subanalytic set, $\mathcal{X}_S \subseteq \mathcal{X}$ to denote a closed subanalytic subset of $\mathcal{X}$, and define $z_\epsilon(x) \in \arg\min_{z\in\mathcal{X}_\epsilon\cap\mathcal{X}_C} \|z - x\|$.

### D.1. Proof of Theorem 3.7-1: the $\epsilon$-global/local solutions relation of the smoothed problem

*Proof of Theorem 3.7-1.* We use $u_l$ as an auxiliary merit function, defined below

$$u_l(x) := \sup_{y\in\mathcal{X}} \ \min_{m\in[M]} \ \{f_m(x) - f_m(y) - \frac{l}{2}\|y - x\|^2\}. \tag{103}$$

Under the conditions in Proposition 3.3-2, solving (OPS) is equivalent to solving

$$\min_{x\in\mathcal{X}} f_0(x), \ \text{ s.t. } x \in \mathcal{X}_{u_l}^* := \{x \in \mathcal{X} \mid u_l(x) \leq 0\}. \tag{104}$$

*Part 1: proof of global solutions relation.* Let $x_\delta$ be an $(\epsilon, \delta)$-global solution to (CP), then $x_\delta \in \mathcal{X}_\delta$, i.e., $v_{l,\tau}(x_\delta) + \tau \ln M \leq \delta$. From Proposition 3.3-2, we have that $u_l(x_\delta) \leq v_{l,\tau}(x_\delta) + \tau \ln M \leq \delta$.

Let $x^*$ be a global solution to (OPS), and let $x_p := \text{Proj}_{\mathcal{X}_{u_l}^*}(x_\delta)$. By the $(\varrho_u, \eta_u)$-HEB of the function $u_l$ in Corollary C.8, see Appendix C.1, we have $\|x_\delta - x_p\| = \text{dist}(x_\delta, \mathcal{X}_{u_l}^*) \leq (\varrho_u \delta)^{\frac{1}{\eta_u}}$.

For $\delta \geq \tau \ln M$, $\mathcal{X}_{u_l}^* \subseteq \mathcal{X}_\delta^*$, thus $f_0(x^*) - f_0(x_\delta) \geq 0$. By the $\ell_f$-local Lipschitz continuity of $f_0$, it holds that

$$f_0(x^*) - f_0(x_\delta) \leq f_0(x_p) - f_0(x_\delta) \leq \ell_f \|x_\delta - x_p\| \leq \ell_f (\varrho_u \delta)^{\frac{1}{\eta_u}}. \tag{105}$$

This proves that $x_\delta$ is an $(\epsilon', \delta)$-global solution to (OPS) with $\epsilon' = \ell_f (\varrho_u \delta)^{\frac{1}{\eta_u}}$.

Conversely, if $x'$ is an $(\epsilon, \delta)$-global optimal solution to (OPS), then by definition,

$$f_0(x') - \min_{x \in \mathcal{X}_{u_l, \delta}} f_0(x) \leq \epsilon, \text{ and } x' \in \mathcal{X}_{u_l, \delta}. \tag{106}$$

In other words, $u_l(x') \leq \delta$, which implies $v_{l,\tau}(x') + \tau \ln M \leq \delta + \tau \ln M$, i.e., $x' \in \mathcal{X}_{\delta'}$ with $\delta' = \delta + \tau \ln M$. Furthermore, since $\mathcal{X}_{u_l, \delta} \subseteq \mathcal{X}_{\delta'}$, $\min_{x \in \mathcal{X}_{\delta'}} f_0(x) \leq \min_{x \in \mathcal{X}_{u_l, \delta}} f_0(x)$, therefore

$$\begin{aligned} f_0(x') - \min_{x \in \mathcal{X}_{\delta'}} f_0(x) = & f_0(x') - \min_{x \in \mathcal{X}_{u_l, \delta}} f_0(x) + \min_{x \in \mathcal{X}_{u_l, \delta}} f_0(x) - \min_{x \in \mathcal{X}_{\delta'}} f_0(x) \\ \leq & \epsilon + \min_{x \in \mathcal{X}_{u_l, \delta}} f_0(x) - \min_{x \in \mathcal{X}_{\delta'}} f_0(x) \end{aligned} \tag{107}$$

where letting $x_{\delta',0}^* \in \arg\min_{x \in \mathcal{X}_{\delta'}} f_0(x)$, and $x_{p,\delta',0}^* = \text{Proj}_{\mathcal{X}_{u_l, \delta}}(x)$, then $\min_{x \in \mathcal{X}_{u_l, \delta}} f_0(x) - \min_{x \in \mathcal{X}_{\delta'}} f_0(x)$ can be further bounded by

$$\begin{aligned} & \min_{x \in \mathcal{X}_{u_l, \delta}} f_0(x) - \min_{x \in \mathcal{X}_{\delta'}} f_0(x) = \min_{x \in \mathcal{X}_{u_l, \delta}} f_0(x) - f_0(x_{\delta',0}^*) \\ \leq & f_0(x_{p,\delta',0}^*) - f_0(x_{\delta',0}^*) \leq \ell_f \|x_{\delta',0}^* - x_{p,\delta',0}^*\| \leq \ell_f \text{dist}(x_{\delta',0}^*, \mathcal{X}_{u_l}^*) \leq \ell_f (\varrho_u \delta')^{\frac{1}{\eta_u}}. \end{aligned} \tag{108}$$

Therefore, $x'$ is an $(\epsilon', \delta')$ solution to (CP) with $\epsilon' = \epsilon + \ell_f (\varrho_u \delta')^{\frac{1}{\eta_u}}$ and $\delta' = \delta + \tau \ln M$.

*Part 2: proof of local solutions relation.* Let $x_\delta$ be an $(\epsilon, \delta)$-local solution to (CP), then $x_\delta \in \mathcal{X}_\delta$, and $x_\delta$ is an $(\epsilon, \delta)$-global solution to (CP) on the set $\mathcal{X} \cap \mathbb{B}(x_\delta, r)$. Applying the results in Part 1 we get that $x_\delta$ is an $(\epsilon', \delta)$-global solution to (OPS) on the set $\mathcal{X} \cap \mathbb{B}(x_\delta, r)$, thus an $(\epsilon', \delta)$-local solution to (OPS) on the set $\mathcal{X}$. Similar arguments can be used to prove the converse statement.

Combining Part 1 and Part 2 completes the proof. $\qquad\square$

## D.2. Proof of Theorem 3.7-2: the $\epsilon$-global/local solutions relation

We first present some auxiliary lemmas below, then prove the main results.

**Lemma D.1.** *Let $\mathcal{X}_C \subseteq \mathbb{R}^q$ be a compact subanalytic set with $\mathcal{X}_C \cap \mathcal{X}_0 \neq \emptyset$, and thus $\mathcal{X}_C \cap \mathcal{X}_\epsilon \neq \emptyset$ for some $\epsilon \geq 0$. Suppose $f_0(x)$ is $\ell_f$-Lipschitz continuous on $\mathcal{X}_C \cap \mathcal{X}$ with some $\ell_f > 0$. If $v_{l,\tau}$ is globally subanalytic on $\mathcal{X}_C \cap \mathcal{X}$, then it satisfies the $(\varrho, \eta)$-HEB on $\mathcal{X}_C \cap \mathcal{X}$ with some $\varrho, \eta > 0$, and $p(x)$ satisfies the $(\varrho_p, \eta_p)$-HEB on $\mathcal{X}_C \cap \mathcal{X}$ with $\varrho_p = \varrho^\theta$ and $\eta_p = \theta\eta$. Given $x \in \mathcal{X}_C \cap \mathcal{X}$, it holds for any $\epsilon \geq 0$ that*

$$f_0(x) + \gamma p(x) - f_0(z_\epsilon(x)) \geq -\epsilon_\gamma := \begin{cases} \ell_f \left(\frac{\ell_f \varrho_p}{\gamma \eta_p}\right)^{\frac{1}{\eta_p - 1}} \left(\frac{1}{\eta_p} - 1\right), & \eta_p > 1, \gamma > 0; \\ 0, & \eta_p = 1, \gamma \geq \varrho_p \ell_f. \end{cases} \tag{109}$$

*Proof of Lemma D.1.* Since $v_{l,\tau}(x)$ and $p(x)$ are globally subanalytic on $\mathcal{X}_C \cap \mathcal{X}$, there exists $\varrho, \eta > 0$ that $v_{l,\tau}(x)$ satisfies the $(\varrho, \eta)$-HEB, thus $p(x)$ satisfies the $(\varrho_p, \eta_p)$-HEB with $\varrho_p = \varrho^\theta$ and $\eta_p = \theta\eta$ on $\mathcal{X}_C \cap \mathcal{X}$, which yields that for all $\epsilon \geq 0$,

$$\varrho_p p(x) \geq \|z_0(x) - x\|^{\eta_p} \geq \|z_\epsilon(x) - x\|^{\eta_p}. \tag{110}$$

Since $\mathcal{X}_C$ is bounded, we have that for all $x, x' \in \mathcal{X}_C$, there exists $\ell_f > 0$ such that $f_0(x) - f_0(x') \geq -\ell_f \|x - x'\|$. Combined with the above inequality, it holds that

$$f_0(x) + \gamma p(x) - f_0(z_\epsilon(x)) \geq \frac{\gamma}{\varrho_p} \|z_\epsilon(x) - x\|^{\eta_p} - \ell_f \|z_\epsilon(x) - x\| \geq \underbrace{\inf_{\zeta \in \mathbb{R}_+} \frac{\gamma}{\varrho_p} \zeta^{\eta_p} - \ell_f \zeta}_{-\epsilon_\gamma}. \tag{111}$$

Analyzing $-\epsilon_\gamma$ separately under $\eta_p > 1$ and $\eta_p = 1$ proves the result. When $\eta_p > 1$, the result is obtained by solving the optimal $\zeta$ through the first-order optimality condition. When $\eta_p = 1$, the optimal value is achieved at $\zeta = 0$. $\qquad\square$

**Lemma D.2.** *Given $x_\gamma \in \mathcal{X}_S \subseteq \mathcal{X}$ with $p(x_\gamma) = \epsilon_\gamma$, and that $x_\gamma$ is an $\epsilon$-global solution to* (PP$_\gamma$) *on $\mathcal{X}_S$, then*

$$f_0(x_\gamma) - \inf_{x \in \mathcal{X}_S \cap \mathcal{X}_{\epsilon_\gamma}} f_0(x) \leq \epsilon \tag{112}$$

*Proof of Lemma D.2.* Since $x_\gamma$ is an $\epsilon$-global solution of (PP$_\gamma$) on $\mathcal{X}_S$, by definition we have that for all $x \in \mathcal{X}_S \cap \mathcal{X}_{\epsilon_\gamma}$,

$$f_0(x_\gamma) + \gamma p(x_\gamma) \leq f_0(x) + \gamma p(x) + \epsilon. \tag{113}$$

Recall $p(x_\gamma) = \epsilon_\gamma$, and $p(x) \leq \epsilon_\gamma$ since $x \in \mathcal{X}_S \cap \mathcal{X}_{\epsilon_\gamma}$. Plugging these into the above inequality yields (112). $\qquad\square$

**Lemma D.3.** *Let $\mathcal{X}_C \subseteq \mathbb{R}^q$ be a compact subanalytic set with $\mathcal{X}_C \cap \mathcal{X}_0 \neq \emptyset$, and thus $\mathcal{X}_C \cap \mathcal{X}_\epsilon \neq \emptyset$ for some $\epsilon \geq 0$. Let $\mathbb{B}(x, r)$ denote the neighborhood of $x$ with radius $r > 0$ for some $x \in \mathcal{X}_C \cap \mathcal{X}$. If there exists $\bar{x} \in \mathbb{B}(x, r) \cap \mathcal{X} \cap \mathcal{X}_C$ such that $p(\bar{x}) \leq \epsilon$, then there exists $z_\epsilon(x) \in \arg\min_{z \in \mathcal{X}_\epsilon \cap \mathcal{X}_C} \|z - x\|$ such that $z_\epsilon(x) \in \mathbb{B}(x, r) \cap \mathcal{X}_\epsilon \cap \mathcal{X}_C$.*

*Proof of Lemma D.3.* As $\mathcal{X}_\epsilon \cap \mathcal{X}_C$ for $\epsilon \geq 0$ is closed and nonempty, there exists $z_\epsilon(x) \in \arg\min_{x \in \mathcal{X}_\epsilon \cap \mathcal{X}_C} \|z - x\|$. Also since $\bar{x} \in \mathbb{B}(x, r) \cap \mathcal{X}$, and $p(\bar{x}) \leq \epsilon$, thus $\bar{x} \in \mathbb{B}(x_\gamma, r) \cap \mathcal{X}_\epsilon \cap \mathcal{X}_C$, then

$$\|z_\epsilon(x) - x\| = \min_{z \in \mathcal{X}_\epsilon \cap \mathcal{X}_C} \|z - x\| \leq \|\bar{x} - x\| \leq r \tag{114}$$

which implies that $z_\epsilon(x) \in \mathbb{B}(x, r)$, combined with $z_\epsilon(x) \in \mathcal{X}_\epsilon \cap \mathcal{X}_C$ proves the result. $\qquad\square$

**Lemma D.4.** *Let $\mathcal{X}_S = \mathcal{X}$ or $\mathcal{X}_S = \mathcal{X} \cap \mathbb{B}(x, r)$. Given $x_{\epsilon_b} \in \mathcal{X}_S \subseteq \mathcal{X}$, which is also an $(\epsilon_b, \epsilon)$-global solution to* (CP) *on $\mathcal{X}_S$. Suppose $\mathcal{X}_S \cap \mathcal{X}_{\epsilon'} \neq \emptyset$ with $\epsilon \geq \epsilon' \geq 0$, and there exist bounded points in $\mathcal{X}_S \cap \mathcal{X}_{\epsilon'}$, then for $\epsilon_\gamma$ specified in* (109) *in Lemma D.1, we have*

$$f_0(x_{\epsilon_b}) + \gamma p(x_{\epsilon_b}) \leq \inf_{x \in \mathcal{X}_S} f_0(x) + \gamma p(x) + \epsilon_\gamma + \epsilon_b + \gamma \epsilon. \tag{115}$$

*Proof of Lemma D.4.* By the definition of $(\epsilon_b, \epsilon)$-global solution to (CP) on $\mathcal{X}_S$, we have

$$p(x_{\epsilon_b}) \leq \epsilon, \text{ and } f_0(x_{\epsilon_b}) \leq \inf_{x \in \mathcal{X}_\epsilon \cap \mathcal{X}_S} f_0(x) + \epsilon_b. \tag{116}$$

Therefore,

$$f_0(x_{\epsilon_b}) + \gamma p(x_{\epsilon_b}) \leq \inf_{x \in \mathcal{X}_\epsilon \cap \mathcal{X}_S} f_0(x) + \epsilon_b + \gamma \epsilon. \tag{117}$$

Recall that $\mathcal{X}_S \cap \mathcal{X}_{\epsilon'} \neq \emptyset$. Let $\mathcal{X}_C$ be a compact subanalytic set such that $\mathcal{X}_C \cap \mathcal{X}_S \cap \mathcal{X}_{\epsilon'} \neq \emptyset$, and $\mathcal{X}_C \cap \mathcal{X}_{S,\varphi_\gamma}^* \neq \emptyset$. The set $\mathcal{X}_C$ exists because there exist bounded points in $\mathcal{X}_S \cap \mathcal{X}_{\epsilon'}$ and $\mathcal{X}_{S,\varphi_\gamma}^*$, respectively. For $x \in \mathcal{X}_C \cap \mathcal{X}$, recall we define $z_\epsilon(x) \in \arg\min_{z \in \mathcal{X}_\epsilon \cap \mathcal{X}_C} \|z - x\|$. When $\mathcal{X}_S = \mathcal{X}$, $z_\epsilon(x) \in \mathcal{X}_S$; when $\mathcal{X}_S = \mathcal{X} \cap \mathbb{B}(x, r)$, $z_\epsilon(x) \in \mathcal{X}_S$ by Lemma D.3. Let $x_C^* \in \arg\min_{x \in \mathcal{X}_C \cap \mathcal{X}_S} \varphi_\gamma(x)$. Then $\inf_{x \in \mathcal{X}_\epsilon \cap \mathcal{X}_S} f_0(x)$ can be further bounded by

$$\begin{aligned}
\inf_{x \in \mathcal{X}_\epsilon \cap \mathcal{X}_S} f_0(x) &\leq \inf_{x \in \mathcal{X}_{\epsilon'} \cap \mathcal{X}_C \cap \mathcal{X}_S} f_0(x) &&\text{since } \mathcal{X}_{\epsilon'} \cap \mathcal{X}_C \cap \mathcal{X}_S \subseteq \mathcal{X}_\epsilon \cap \mathcal{X}_S \\
&\leq f_0(z_{\epsilon'}(x_C^*)) &&\text{since } z_{\epsilon'}(x_C^*) \in \mathcal{X}_{\epsilon'} \cap \mathcal{X}_C \cap \mathcal{X}_S \\
&\leq f_0(x_C^*) + \gamma p(x_C^*) + \epsilon_\gamma &&\text{from Lemma D.1} \\
&= \inf_{x \in \mathcal{X}_S} f_0(x) + \gamma p(x) + \epsilon_\gamma. &&\text{since } \mathcal{X}_C \cap \mathcal{X}_{S,\varphi_\gamma}^* \neq \emptyset
\end{aligned} \tag{118}$$

Plugging (118) into (117) yields

$$f_0(x_{\epsilon_b}) + \gamma p(x_{\epsilon_b}) \leq \inf_{x \in \mathcal{X}_S} f_0(x) + \gamma p(x) + \epsilon_\gamma + \epsilon_b + \gamma\epsilon. \tag{119}$$

The proof is complete. $\qquad\square$

*Proof of Theorem 3.7-2.* 1) Given $\gamma > 0$, let $x_\gamma$ be a bounded $\epsilon$-global solution of (PP$_\gamma$) on $\mathcal{X}_S$, with $\mathcal{X}_S = \mathcal{X}$ or $\mathcal{X}_S = \mathbb{B}(x_\gamma, r) \cap \mathcal{X}$, then for all $x \in \mathcal{X}_S$,

$$f_0(x_\gamma) + \gamma p(x_\gamma) \leq f_0(x) + \gamma p(x) + \epsilon. \tag{120}$$

Let $x^* \in \mathcal{X}_S \cap \mathcal{X}_{\epsilon^*}$ denote a bounded $(0, \epsilon^*)$-global solution of (CP) on $\mathcal{X}_S$. Then there exists a compact and subanalytic set $\mathcal{X}_C$ such that $x_\gamma, x^* \in \mathcal{X}_C$ and $\mathcal{X}_C \cap \mathcal{X}_{\epsilon^*} \neq \emptyset$. Let $z_{\epsilon^*}(x) \in \arg\min_{z \in \mathcal{X}_{\epsilon^*} \cap \mathcal{X}_C} \|z - x\|$. When 1) $\mathcal{X}_S = \mathcal{X}$, $z_{\epsilon^*}(x_\gamma) \in \mathcal{X}_S \cap \mathcal{X}_{\epsilon^*}$; when 2) $\mathcal{X}_S = \mathbb{B}(x_\gamma, r)$, $z_{\epsilon^*}(x_\gamma) \in \mathcal{X}_S \cap \mathcal{X}_{\epsilon^*}$ according to Lemma D.3. Then by Lemma D.1, given $\gamma' > 0$, in both cases we have that

$$f_0(x_\gamma) + \gamma' p(x_\gamma) - f_0(z_{\epsilon^*}(x_\gamma)) \geq -\epsilon_{\gamma'}. \tag{121}$$

Because $f_0(z_{\epsilon^*}(x_\gamma)) \geq f_0(x^*)$, (121) indicates that $f_0(x_\gamma) + \gamma' p(x_\gamma) \geq f_0(x^*) - \epsilon_{\gamma'}$. Plugging $x = x^*$ in (120), and combining with the above inequality, we obtain that

$$\begin{aligned} f_0(x_\gamma) + \gamma p(x_\gamma) &\leq f_0(x^*) + \gamma\epsilon^* + \epsilon && \text{since } p(x^*) \leq \epsilon^* \\ &\leq f_0(x_\gamma) + \gamma' p(x_\gamma) + \epsilon_{\gamma'} + \gamma\epsilon^* + \epsilon && \text{from (121)} \end{aligned} \tag{122}$$

which further implies

$$(\gamma - \gamma') p(x_\gamma) \leq \epsilon_{\gamma'} + \gamma\epsilon^* + \epsilon. \tag{123}$$

Define $\epsilon_\gamma := p(x_\gamma)$, then $x_\gamma \in \mathcal{X}_{\epsilon_\gamma} := \{x \in \mathcal{X} \mid p(x_{\epsilon_\gamma}) \leq \epsilon_\gamma\}$. The above inequality implies $\epsilon_\gamma \leq \frac{\epsilon_{\gamma'} + \gamma\epsilon^* + \epsilon}{\gamma - \gamma'}$. Further, from Lemma D.2 we have

$$f_0(x_\gamma) - \inf_{x \in \mathcal{X}_S \cap \mathcal{X}_{\epsilon_\gamma}} f_0(x) \leq \epsilon. \tag{124}$$

Combining (123) and (124) proves that $x_\gamma$ is an $(\epsilon, \epsilon_\gamma)$-approximate global solution to (CP) on $\mathcal{X}_S$ with $\epsilon_\gamma \leq \frac{\epsilon_{\gamma'} + \gamma\epsilon^* + \epsilon}{\gamma - \gamma'}$.

When $\eta_p > 1$, choosing $\gamma' = \frac{\gamma}{2}$, $\gamma \geq 2\ell_f \delta^{-\frac{\eta_p - 1}{\eta_p}} \left(\frac{\varrho_p}{\eta_p}\right)^{\frac{1}{\eta_p}} \left(1 - \frac{1}{\eta_p}\right)^{\frac{\eta_p - 1}{\eta_p}}$, and $\epsilon \leq \ell_f \delta^{\frac{1}{\eta_p}} \left(\frac{\varrho_p}{2\eta_p}\right)^{\frac{1}{\eta_p}} \left(1 - \frac{1}{\eta_p}\right)^{\frac{\eta_p - 1}{\eta_p}}$, we further have that

$$\epsilon_\gamma \leq \frac{\epsilon_{\gamma'} + \epsilon + \gamma\epsilon^*}{\gamma - \gamma'} = \left(\frac{2\ell_f}{\gamma}\right)^{\frac{\eta_p}{\eta_p - 1}} \left(\frac{\varrho_p}{\eta_p}\right)^{\frac{1}{\eta_p - 1}} \left(1 - \frac{1}{\eta_p}\right) + \frac{2\epsilon}{\gamma} + 2\epsilon^* \leq \delta + 2\epsilon^* \tag{125}$$

which proves that the $\epsilon$-global solution of (PP$_\gamma$) on $\mathcal{X}_S$ with $\gamma \geq 2\ell_f \delta^{-\frac{\eta_p - 1}{\eta_p}} \left(\frac{\varrho_p}{\eta_p}\right)^{\frac{1}{\eta_p}} \left(1 - \frac{1}{\eta_p}\right)^{\frac{\eta_p - 1}{\eta_p}}$ and $\epsilon \leq \ell_f \delta^{\frac{1}{\eta_p}} \left(\frac{\varrho_p}{2\eta_p}\right)^{\frac{1}{\eta_p}} \left(1 - \frac{1}{\eta_p}\right)^{\frac{\eta_p - 1}{\eta_p}}$, is an $(\epsilon, \delta + 2\epsilon^*)$-global solution to (CP) on $\mathcal{X}_S$.

When $\eta_p = 1$, from Lemma D.1, for $\gamma' \geq \varrho_p \ell_f$, $\epsilon_{\gamma'} = 0$. Choosing $\gamma \geq \varrho_p \ell_f + 1$, we have $\epsilon_\gamma \leq \epsilon$. Therefore, the $\epsilon$-global solution of (PP$_\gamma$) on $\mathcal{X}_S$ is an $(\epsilon, \epsilon + 2\epsilon^*)$-global solution to (CP) on $\mathcal{X}_S$. The set $\mathcal{X}_S$ can be $\mathcal{X}$ or $\mathbb{B}(x_\gamma, r) \cap \mathcal{X}$, which corresponds to the global solution on $\mathcal{X}$, or the local solution in the neighborhood of $x_\gamma$, respectively.

2) Next we prove the converse. Define $x_{\epsilon_b}$ to be a bounded $(\epsilon_b, \epsilon)$-global solution to (CP) on $\mathcal{X}_S$, and recall $\mathcal{X}_\epsilon := \{x \in \mathcal{X} \mid p(x) \leq \epsilon\}$. Then by Lemma D.4, we have

$$\begin{aligned} f_0(x_{\epsilon_b}) + \gamma p(x_{\epsilon_b}) &\leq \inf_{x \in \mathcal{X}_S} f_0(x) + \gamma p(x) + \epsilon_\gamma + \epsilon_b + \gamma\epsilon \\ &\leq \inf_{x \in \mathcal{X}_S} f_0(x) + \gamma p(x) + \delta \end{aligned} \tag{126}$$

where the last inequality holds by choosing $\epsilon \leq \frac{\delta}{3\gamma}$, $\epsilon_b \leq \frac{\delta}{3}$, and $\gamma = \frac{\ell_f \varrho_p}{\eta_p} \left(\frac{3\ell_f (1 - \frac{1}{\eta_p})}{\delta}\right)^{\eta_p - 1}$ when $\eta_p > 1$, $\gamma = \varrho_p \ell_f$ when $\eta_p = 1$. The set $\mathcal{X}_S$ can be $\mathcal{X}$ or $\mathbb{B}(x_{\epsilon_b}, r) \cap \mathcal{X}$, which corresponds to the global solution on $\mathcal{X}$, or the local solution in the neighborhood of $x_{\epsilon_b}$, respectively. The converse is proved. $\qquad\square$

**Lemma D.5.** *If $x_\gamma \in \mathcal{X}$ is an $\epsilon$-stationary solution to* $(PP_\gamma)$, *and $\|\nabla f_0(x_\gamma)\| \leq \ell_f$, then it is also an $\epsilon_\gamma$-stationary solution to $\min_{x \in \mathcal{X}} p(x)$ with $\epsilon_\gamma = \frac{\epsilon + \ell_f}{\gamma}$. Furthermore, for $\theta \geq 1$, let $c_v(x_\gamma) := \theta(v_{l,\tau}(x_\gamma) + \tau \ln M)^{\theta - 1}$. Then either one of the following two conditions holds.*
*1) $c_v(x_\gamma) = 0$ and $x_\gamma$ is an optimal solution, thus also a stationary solution to $\min_{x \in \mathcal{X}} v_{l,\tau}(x)$;*
*2) $c_v(x_\gamma) > 0$, and $x_\gamma$ is an $\epsilon_\gamma'$-stationary solution to $\min_{x \in \mathcal{X}} v_{l,\tau}(x)$ with $\epsilon_\gamma' = \frac{\epsilon + \ell_f}{\gamma c_v(x_\gamma)}$.*

*Proof of Lemma D.5.* Since $x_\gamma \in \mathcal{X}$ is an $\epsilon$-stationary solution to $(PP_\gamma)$, for $\alpha > 0$, we have

$$\frac{1}{\alpha}\|x_\gamma - \mathrm{Proj}_{\mathcal{X}}(x_\gamma - \alpha \nabla \varphi_\gamma(x_\gamma))\| \leq \epsilon. \tag{127}$$

By the definition that $\varphi_\gamma(x) = f_0(x) + \gamma p(x)$, we further have that

$$\begin{aligned}
&\|x_\gamma - \mathrm{Proj}_{\mathcal{X}}(x_\gamma - \alpha\gamma\nabla p(x_\gamma))\| \\
&\leq \|x_\gamma - \mathrm{Proj}_{\mathcal{X}}(x_\gamma - \alpha\nabla\varphi_\gamma(x_\gamma))\| + \|\mathrm{Proj}_{\mathcal{X}}(x_\gamma - \alpha\nabla\varphi_\gamma(x_\gamma)) - \mathrm{Proj}_{\mathcal{X}}(x_\gamma - \alpha\gamma\nabla p(x_\gamma))\| \\
&\leq \|x_\gamma - \mathrm{Proj}_{\mathcal{X}}(x_\gamma - \alpha\nabla\varphi_\gamma(x_\gamma))\| + \|\alpha\nabla\varphi_\gamma(x_\gamma) - \alpha\gamma\nabla p(x_\gamma)\| \\
&= \|x_\gamma - \mathrm{Proj}_{\mathcal{X}}(x_\gamma - \alpha\nabla\varphi_\gamma(x_\gamma))\| + \alpha\|\nabla f_0(x_\gamma)\|.
\end{aligned} \tag{128}$$

Dividing both sides by $\frac{1}{\alpha\gamma}$ of the above inequality yields

$$\frac{1}{\alpha\gamma}\|x_\gamma - \mathrm{Proj}_{\mathcal{X}}(x_\gamma - \alpha\gamma\nabla p(x_\gamma))\| \leq \frac{\epsilon + \ell_f}{\gamma} \tag{129}$$

which proves that $x_\gamma$ is an $\epsilon_\gamma$-stationary solution to $\min_{x \in \mathcal{X}} p(x)$ with $\epsilon_\gamma = \frac{\epsilon + \ell_f}{\gamma}$. By the definition that $p(x_\gamma) = (v_{l,\tau}(x_\gamma) + \tau \ln M)^\theta$, for $\theta \geq 1$, we further have

$$\frac{1}{\alpha\gamma}\|x_\gamma - \mathrm{Proj}_{\mathcal{X}}(x_\gamma - \alpha\gamma\underbrace{\theta(v_{l,\tau}(x_\gamma) + \tau \ln M)^{\theta - 1}}_{c_v(x_\gamma)}\nabla v_{l,\tau}(x_\gamma))\| \leq \frac{\epsilon + \ell_f}{\gamma} \tag{130}$$

where $c_v(x_\gamma) \geq 0$. This implies that either $v_{l,\tau}(x_\gamma) + \tau \ln M = 0$, or $c_v(x_\gamma) > 0$, and

$$\frac{1}{\alpha\gamma c_v(x_\gamma)}\|x_\gamma - \mathrm{Proj}_{\mathcal{X}}(x_\gamma - \alpha\gamma c_v(x_\gamma)\nabla v_{l,\tau}(x_\gamma))\| \leq \frac{\epsilon + \ell_f}{\gamma c_v(x_\gamma)}. \tag{131}$$

The proof is complete. $\square$

**Lemma D.6.** *Suppose Assumption 4 holds. Recall that $y_{l,\tau}^*(x) := \arg\min_{y \in \mathcal{X}} h_{l,\tau}(x, y)$. For $l - \ell_{f,1} > 0$, we have*

$$\frac{l - \ell_{f,1}}{2}\|x - y_{l,\tau}^*(x)\|^2 \leq v_{l,\tau}(x) + \tau \ln M \leq \frac{3\ell_{h_{l,\tau},1}^2 + 6\alpha^{-2}}{2(l - \ell_{f,1})}\|x - y_{l,\tau}^*(x)\|^2. \tag{132}$$

*Proof of Lemma D.6.* By definition, $v_{l,\tau}(x) = -h_{l,\tau}(x, y_{l,\tau}^*(x))$, and $h_{l,\tau}(x, x) = \tau \ln M$. Therefore,

$$v_{l,\tau}(x) + \tau \ln M = -h_{l,\tau}(x, y_{l,\tau}^*(x)) + h_{l,\tau}(x, x) \geq \frac{l - \ell_{f,1}}{2}\|x - y_{l,\tau}^*(x)\|^2 \tag{133}$$

where the last inequality follows from the $(l - \ell_{f,1})$-strong convexity of $h_{l,\tau}(x, \cdot)$, thus the quadratic growth. The first inequality in (132) is proved.

For the second inequality in (132), we have

$$\begin{aligned}
v_{l,\tau}(x) + \tau \ln M &= -h_{l,\tau}(x, y_{l,\tau}^*(x)) + h_{l,\tau}(x, x) \overset{(a)}{\leq} \frac{1}{2(l - \ell_{f,1})\alpha^2}\|x - \mathrm{Proj}_{\mathcal{X}}(x - \alpha\nabla_y h_{l,\tau}(x, x))\|^2 \\
&\overset{(b)}{\leq} \frac{1}{2(l - \ell_{f,1})\alpha^2}\|x - \mathrm{Proj}_{\mathcal{X}}(x - \alpha\nabla_y h_{l,\tau}(x, x)) - y_{l,\tau}^*(x) + \mathrm{Proj}_{\mathcal{X}}(y_{l,\tau}^*(x) - \alpha\nabla_y h_{l,\tau}(x, y_{l,\tau}^*(x)))\|^2
\end{aligned}$$

$$\overset{(c)}{\leq} \frac{3\ell_{h_{l,\tau},1}^2 + 6\alpha^{-2}}{2(l - \ell_{f,1})} \|x - y_{l,\tau}^*(x)\|^2 \tag{134}$$

where $(a)$ follows from the $(l - \ell_{f,1})$-strong convexity of $h_{l,\tau}(x, \cdot)$, thus the proximal-PL inequality, $(b)$ follows from the optimality condition of $\min_{y \in \mathcal{X}} h_{l,\tau}(x, y)$ at $y_{l,\tau}^*(x)$, and $(c)$ follows from the $\ell_{h_{l,\tau},1}$-smoothness of $h_{l,\tau}(x, \cdot)$, and the non-expansiveness of projection. $\qquad \square$

**Corollary D.7.** *Lemma D.6 implies that, for $l - \ell_{f,1} > 0$, $v_{l,\tau}(x)$ achieves its minimum value if and only if $y_{l,\tau}^*(x) = x$.*

*Proof of Proposition 3.9.* Since $x_\gamma$ is a local solution to (PP$_\gamma$), it is a stationary solution to (PP$_\gamma$), thus an $\epsilon$-stationary solution to $\min_{x \in \mathcal{X}} v_{l,\tau}$ by Lemma D.5.

We then show the $\epsilon$-stationary solution of $\min_{x \in \mathcal{X}} v_{l,\tau}$ is also an $\epsilon'$-optimal solution of $\min_{x \in \mathcal{X}} v_{l,\tau}$ under additional assumptions of $F$. By Lemma B.11, the directional derivative of $v_{l,\tau}$, denoted as $v_{l,\tau}'(x; z - x)$, can be computed by

$$v_{l,\tau}'(x; z - x) = \sum_{m=1}^M \pi_m(x, y_{l,\tau}^*(x)) f_m'(x; z - x) - l\Big(x - y_{l,\tau}^*(x)\Big)^\top (z - x) \geq -\epsilon \|z - x\| \quad \text{for all } z \in \mathcal{X}. \tag{135}$$

Plugging $z = y_{l,\tau}^*(x)$ into the above inequality yields

$$\sum_{m=1}^M \pi_m(x, y_{l,\tau}^*(x)) f_m'(x; y_{l,\tau}^*(x) - x) + l\|y_{l,\tau}^*(x) - x\|^2 \geq -\epsilon \|y_{l,\tau}^*(x) - x\|. \tag{136}$$

Furthermore, let $P(x)$ be the indicator function on $\mathcal{X}$, by the optimality condition of $h_{l,\tau}(x, \cdot)$ at $y_{l,\tau}^*(x)$, we have

$$0 \in \sum_{m=1}^M \pi_m(x, y_{l,\tau}^*(x)) \nabla f_m(y_{l,\tau}^*(x)) + l\big(y_{l,\tau}^*(x) - x\big) + \partial P(y_{l,\tau}^*(x)). \tag{137}$$

For analysis, we construct an auxiliary function $R(y) := \lambda^\top F(y) + P(y)$, where $\lambda = [\lambda_1; \lambda_2; \ldots; \lambda_M] \in \Delta^M$ is a hyperparameter. Then $R(y)$ is $(1, 0)$-point quasar convex at $y = y_{l,\tau}^*(x)$. By the definition of the subgradient of $R(y)$, and the $(1, 0)$-point quasar convexity, for all $z \in \mathcal{X}$, we have

$$\Big\langle \sum_{m=1}^M \lambda_m \nabla f_m(y_{l,\tau}^*(x)) + \partial P(y_{l,\tau}^*(x)), z - y_{l,\tau}^*(x) \Big\rangle \leq \sum_{m=1}^M \lambda_m \big(f_m(z) - f_m(y_{l,\tau}^*(x))\big). \tag{138}$$

Letting $\lambda_m = \pi_m(x, y_{l,\tau}^*(x))$ given that $x \in \mathcal{X}$ is a stationary point of $\min_{x \in \mathcal{X}} v_{l,\tau}(x)$, and plugging $y = y_{l,\tau}^*(x)$, $l(x - y_{l,\tau}^*(x)) \in \sum_{m=1}^M \pi_m(x, y_{l,\tau}^*(x)) \nabla_y f_m(y_{l,\tau}^*(x)) + \partial P(y_{l,\tau}^*(x))$ into the above inequality yield

$$l\langle x - y_{l,\tau}^*(x), z - y_{l,\tau}^*(x) \rangle \leq \sum_{m=1}^M \pi_m(x, y_{l,\tau}^*(x)) \Big(f_m(z) - f_m(y_{l,\tau}^*(x))\Big). \tag{139}$$

Substituting $z = x$ into the above inequality, then $l\|x - y_{l,\tau}^*(x)\|^2 \leq \sum_{m=1}^M \pi_m(x, y_{l,\tau}^*(x)) \Big(f_m(x) - f_m(y_{l,\tau}^*(x))\Big)$, which combined with (136) yields

$$\sum_{m=1}^M \pi_m(x, y_{l,\tau}^*(x)) \Big(f_m(y_{l,\tau}^*(x)) - f_m(x)\Big) - \epsilon \|y_{l,\tau}^*(x) - x\| \leq \sum_{m=1}^M \pi_m(x, y_{l,\tau}^*(x)) f_m'(x; y_{l,\tau}^*(x) - x)$$

$$\leq \sum_{m=1}^M \pi_m(x, y_{l,\tau}^*(x)) \Big(f_m(y_{l,\tau}^*(x)) - f_m(x)\Big) - \mu \|y_{l,\tau}^*(x) - x\|^2 \tag{140}$$

where the last inequality holds since $f_m(x)$ are $(1, \mu)$-strong point quasar convex for all $m \in [M]$ at $x$. The above inequality implies $\mu \|y_{l,\tau}^*(x) - x\| \leq \epsilon$.

Applying Lemma D.6 and letting $\alpha = O(1)$, we have

$$v_{l,\tau}(x) + \tau \ln M \leq \frac{3\ell^2_{h_{l,\tau},1} + 6\alpha^{-2}}{2(l - \ell_{f,1})} \|x - y^*_{l,\tau}(x)\|^2 \leq \frac{3\ell^2_{h_{l,\tau},1} + 6\alpha^{-2}}{2(l - \ell_{f,1})\mu^2} \epsilon^2 = O(\epsilon^2). \tag{141}$$

Then there exists $\bar{x} \in \mathcal{N}(x_\gamma, r)$ that $p(\bar{x}) = O(\epsilon^{2\theta})$. $\qquad\qquad\square$

### D.3. Proof of Theorem 3.11: the $\epsilon$-stationary solutions relation

We first discuss the stationary condition of (CP) when $\mathcal{X} = \mathbb{R}^q$, and the calmness condition that ensures the KKT condition is a necessary condition. Then we prove Theorem 3.11, the relation of $\epsilon$-stationary solutions to $(\text{PP}_\gamma)$ and (CP). Consider a general constrained problem below

$$\min_{x \in \mathbb{R}^q} f_0(x) \quad \text{s.t. } H(x) = 0 \tag{142}$$

where $f_0 : \mathbb{R}^q \to \mathbb{R}$, and $H : \mathbb{R}^q \to \mathbb{R}^{d_h}$ with $d_h \geq 1$.

**Definition D.8** (KKT condition of (142))**.** The KKT condition of (142) is

$$\underbrace{H(x) = 0}_{\text{feasibility}}, \quad \underbrace{\nabla f_0(x) + \nabla H(x)w = 0}_{\text{stationarity}}, \quad \text{with } w \in \mathbb{R}^{d_h}. \tag{143}$$

Correspondingly, the $(\epsilon', \epsilon)$-KKT condition of (142) is

$$\|\nabla f_0(x) + \nabla H(x)w\| \leq \epsilon', \quad \|H(x)\| \leq \epsilon, \quad \text{with } w \in \mathbb{R}^{d_h}. \tag{144}$$

**Definition D.9** (Calmness (Clarke, 1990, Definition 6.4.1))**.** Let $x^*$ be the global minimizer of (142). If there exist $\epsilon, c > 0$ such that for any $u \in \mathbb{R}^{d_h}$ with $\|u\| \leq \epsilon$ and any $x$ that $\|x - x^*\| \leq \epsilon$ which satisfies $H(x) + u = 0$, one has

$$f_0(x) - f_0(x^*) + c\|u\| \geq 0. \tag{145}$$

Then the problem (142) is said to be calm with $c$.

**Lemma D.10** ((Ye, 2000, Theorem 3.6))**.** *If the problem* (142) *is calm at a global solution* $x^*$*, then* $x^*$ *satisfies the KKT condition in Definition D.8.*

Below is a lemma to show that if the objective is Lipschitz and the constraint satisfies error bound with exponent no greater than one, then the calmness condition holds. Similar results have been discussed in (Ye, 2000, Proposition 4.2) with exponent equal to one. This result connects error bound with the calmness condition, and thus the necessity of KKT condition.

**Lemma D.11.** *Let* $x^*$ *be a global minimizer of problem* (142)*. For* $\epsilon < 1$*, consider any* $u \in \mathbb{R}^{d_h}$ *and* $\|u\| \leq \epsilon$*, and any* $x$ *that* $\|x - x^*\| \leq \epsilon$ *and* $H(x) + u = 0$*. Define* $x_p = \text{Proj}_{\mathcal{X}^*_H}(x)$*, where* $\mathcal{X}^*_H = \{x \in \mathbb{R}^q \mid H(x) = 0\}$*. If* $H(x)$ *satisfies an error bound that* $\|H(x)\| \geq \varrho_h \|x - x_p\|^{\eta_h}$ *and* $f_0$ *is* $\ell_f$*-Lipschitz for all* $x \in \mathbb{B}(x^*, 2\epsilon)$ *with* $\epsilon < 1$*, then the calmness condition in Definition D.9 for problem* (142) *holds.*

*Proof of Lemma D.11.* By definition, for $x \in \mathbb{B}(x^*, \epsilon)$ with $\epsilon \leq 1$,

$$\epsilon \geq \|u\| = \|H(x)\| \geq \varrho_h \|x - x_p\|^{\eta_h}. \tag{146}$$

Since $x^*$ is a global minimizer, and $\|u\| \leq \epsilon \leq 1$, for $\eta_h \leq 1$,

$$f_0(x) - f_0(x^*) \geq f_0(x) - f_0(x_p) \overset{(a)}{\geq} -\ell_f \|x - x_p\| \geq -\frac{\ell_f}{\varrho_h} \|u\|^{\frac{1}{\eta_h}} \geq -\frac{\ell_f}{\varrho_h} \|u\|. \tag{147}$$

where $(a)$ holds because of the $\ell_f$-Lipschitz continuity of $f_0$ on a bounded set that includes $x$ and $x_p$. Therefore, the calmness condition in Definition D.9 holds with $c = \frac{\ell_f}{\varrho_h}$. $\qquad\square$

**Proposition D.12.** *Recall that* $p(x) = \left(v_{l,\tau}(x) + \tau \ln M\right)^\theta$ *with* $\theta > 0$*. If* $v_{l,\tau}$ *satisfies the* $(c_v, \alpha_v)$*-KL inequality on* $\Omega$ *with* $\alpha_v > 1$*, then* $p$ *satisfies the* $(c_p, \alpha_p)$*-KL inequality on* $\Omega$ *with* $\alpha_p = \frac{\theta}{\theta - 1 + \frac{1}{\alpha_v}} > 1$*, and* $c_p = \theta^{-\frac{\theta}{\theta - 1 + \frac{1}{\alpha_v}}} \cdot c_v^{\frac{\theta}{(\theta - 1)\alpha_v + 1}}$*.*

*Proof of Proposition D.12.* For $x \in \Omega$ and that $v_{l,\tau}(x) + \tau \ln M > 0$, we have

$$
\begin{aligned}
\|\nabla p(x)\| =& \theta(v_{l,\tau}(x) + \tau \ln M)^{\theta - 1} \|\nabla v_{l,\tau}(x)\| \\
\geq& \theta(c_v)^{-\frac{1}{\alpha_v}} \left(v_{l,\tau}(x) + \tau \ln M\right)^{\theta - 1 + \frac{1}{\alpha_v}} = \theta(c_v)^{-\frac{1}{\alpha_v}} \left(p(x)\right)^{\frac{\theta - 1 + \frac{1}{\alpha_v}}{\theta}}.
\end{aligned}
\tag{148}
$$

Rearranging the above inequality proves the result. $\qquad\square$

Theorem 3.11 requires the assumption of the $\ell_{v,2}$-smoothness of $\nabla v_{l,\tau}$ on a bounded set. Below we provide a sufficient condition, which shows that under additional assumptions of $f_m, m \in [M]$, the $\ell_{v,2}$-smoothness of $\nabla v_{l,\tau}$ on a bounded set $\mathcal{X}_C$ can be justified.

**Lemma D.13** (Smoothness of $\nabla v_{l,\tau}$). *Suppose Assumption 2 holds, and $l + \mu > 0$. If $\nabla f_m$ is $\ell_{f,2}$-smooth for all $m \in [M]$ on a bounded set $\mathcal{X}_C$, and there exists $x' \in \mathcal{X}_C$ that $\nabla^2 f_m$ for all $m \in [M]$ is bounded, then $\nabla v_{l,\tau}$ is $\ell_{v,2}$-smooth on $\mathcal{X}_C$, with $\ell_{v,2} = \ell_{h_{xx},2}(1 + \ell_{y^*_{l,\tau}}) + \ell_{y^*_{l,\tau},1}\ell_{h_{xy},1} + \ell_{y^*_{l,\tau}}\ell_{h_{xy},2}(1 + \ell_{y^*_{l,\tau}})$.*

*Proof of Lemma D.13.* With similar arguments as Lemma E.1, since $\nabla f_m$ is $\ell_{f,2}$-smooth for all $m \in [M]$ on a bounded set $\mathcal{X}_C$, and there exists $x' \in \mathcal{X}_C$ that $\nabla^2 f_m(x')$ for all $m \in [M]$ is bounded, therefore, there exists $\ell_{f,1} > 0$ such that for $x \in \mathcal{X}_C$, and for all $m \in [M]$, $\|\nabla^2 f_m(x)\| \leq \ell_{f,1}$.

Under Assumption 2, and with $l + \mu > 0$, first recall from (5) that $\nabla v_{l,\tau}(x)$ can be computed by

$$
\nabla v_{l,\tau}(x) = -\nabla_x h(x,y) \mid_{y=y^*_{l,\tau}(x)} = \sum_{m=1}^M \pi_m(x,y)\nabla f_m(x) - l(x-y) \mid_{y=y^*_{l,\tau}(x)}.
\tag{149}
$$

Because of the twice continuous differentiability of $f_m$ for $m \in [M]$, $\nabla^2 v_{l,\tau}(x)$ exists and can be computed by

$$
\nabla^2 v_{l,\tau}(x) = -\nabla^2_{xx}h(x,y) - \nabla y^*_{l,\tau}(x)\nabla^2_{xy}h(x,y) \mid_{y=y^*_{l,\tau}(x)}.
\tag{150}
$$

For simplicity, we simplify $y^*_{l,\tau}(x)$ as $y^*(x)$, and $h_{l,\tau}(x,y)$ as $h(x,y)$ in the following derivations. Then $\|\nabla^2 v_{l,\tau}(x) - \nabla^2 v_{l,\tau}(x')\|$ can be bounded by

$$
\|\nabla^2 v_{l,\tau}(x) - \nabla^2 v_{l,\tau}(x')\| \leq \underbrace{\|\nabla^2_{xx}h(x,y^*(x)) - \nabla^2_{xx}h(x',y^*(x'))\|}_{J_1} + \underbrace{\|\nabla y^*(x) - \nabla y^*(x')\|\|\nabla^2_{xy}h(x,y^*(x))\|}_{J_2}
$$
$$
+ \underbrace{\|\nabla y^*(x)\|\|\nabla^2_{xy}h(x,y^*(x)) - \nabla^2_{xy}h(x',y^*(x'))\|}_{J_3}
\tag{151}
$$

where $J_1$ can be further bounded by

$$
J_1 \leq \ell_{h_{xx},2}\left(\|x - x'\| + \|y^*(x) - y^*(x')\|\right) \leq \ell_{h_{xx},2}(1 + \ell_{y^*_{l,\tau}})\|x - x'\|
\tag{152}
$$

with $\ell_{h_{xx},2}$ denoting the Lipschitz continuity of $\nabla^2_{xx}h(x,y)$ w.r.t. $[x;y]$. Similarly, with $\ell_{h_{xy},2}$ denoting the Lipschitz continuity of $\nabla^2_{xy}h(x,y)$ w.r.t. $[x;y]$, $J_3$ can be bounded by

$$
J_3 \leq \ell_{y^*_{l,\tau}}\ell_{h_{xy},2}(1 + \ell_{y^*_{l,\tau}})\|x - x'\|.
\tag{153}
$$

And $J_2$ can be bounded by

$$
J_2 \leq \ell_{y^*_{l,\tau},1}\ell_{h_{xy},1}\|x - x'\|.
\tag{154}
$$

Therefore,

$$
\|\nabla^2 v_{l,\tau}(x) - \nabla^2 v_{l,\tau}(x')\| \leq \underbrace{\left(\ell_{h_{xx},2}(1 + \ell_{y^*_{l,\tau}}) + \ell_{y^*_{l,\tau},1}\ell_{h_{xy},1} + \ell_{y^*_{l,\tau}}\ell_{h_{xy},2}(1 + \ell_{y^*_{l,\tau}})\right)}_{\ell_{v,2}} \|x - x'\|.
\tag{155}
$$

The derivation of $\ell_{y^*_{l,\tau}}$ is discussed in Lemma B.8. We next discuss the derivation for $\ell_{h_{xy},1}, \ell_{h_{xx},2}, \ell_{h_{xy},2}$ and $\ell_{y^*_{l,\tau},1}$.

To compute $\ell_{y^*_{l,\tau},1}$, from implicit differentiation, $\nabla y^*_{l,\tau}(x)$ can be computed by

$$\nabla y^*_{l,\tau}(x) = -\nabla^2_{xy} h(x, y^*(x)) \left[\nabla^2_{yy} h(x, y^*(x))\right]^{-1}. \tag{156}$$

Then $\|\nabla y^*_{l,\tau}(x) - \nabla y^*_{l,\tau}(x')\|$ can be bounded by

$$\|\nabla y^*_{l,\tau}(x) - \nabla y^*_{l,\tau}(x')\| \leq \|\nabla^2_{xy} h(x, y^*(x)) - \nabla^2_{xy} h(x', y^*(x'))\| \|\left[\nabla^2_{yy} h(x, y^*(x))\right]^{-1}\|$$
$$+ \|\nabla^2_{xy} h(x', y^*(x'))\| \|\left[\nabla^2_{yy} h(x, y^*(x))\right]^{-1}\| \|\left[\nabla^2_{yy} h(x', y^*(x'))\right]^{-1}\| \left\|\nabla^2_{yy} h(x, y^*(x)) - \nabla^2_{yy} h(x', y^*(x'))\right\|$$
$$\leq \underbrace{\left(\ell_{h_{xy},2}(l+\mu)^{-1} + \ell_{h_{xy},1}(l+\mu)^{-2}\ell_{h_{yy},2}\right)(1 + \ell_{y^*_{l,\tau}})}_{\ell_{y^*_{l,\tau},1}} \|x - x'\|. \tag{157}$$

We then proceed to bound $\ell_{h_{xy},1}$.

$$\nabla^2_{xy} h(x, y) = \sum_{m=1}^M \nabla_y \pi_m(x, y) \nabla f_m(x)^\top + lI_q$$
$$= \frac{1}{\tau} \nabla F(y) \left(\pi(x, y)\pi(x, y)^\top - \operatorname{diag}(\pi(x, y))\right) \nabla F(x)^\top - lI_q. \tag{158}$$

We have that

$$\|\nabla^2_{xy} h(x, y)\| \leq \ell_f + l := \ell_{h_{xy},1}. \tag{159}$$

Furthermore, to bound $\ell_{h_{xy},2}$, we have

$$\|\nabla^2_{xy} h(x, y) - \nabla^2_{xy} h(x', y')\| \leq \frac{1}{\tau} \Big( \|\nabla F(y) - \nabla F(y')\| \|\left(\pi(x, y)\pi(x, y)^\top - \operatorname{diag}(\pi(x, y))\right)\nabla F(x)^\top\|$$
$$+ \|\nabla F(y')\| \|\pi(x, y) - \pi(x', y')\| (\|\pi(x, y)\| + \|\pi(x', y')\| + 1)\|\nabla F(x)\|$$
$$+ \|\nabla F(y')\left(\pi(x', y')\pi(x', y')^\top - \operatorname{diag}(\pi(x', y'))\right)\| \|\nabla F(x) - \nabla F(x')\| \Big)$$
$$\leq \underbrace{\frac{1}{\tau}\left(2\ell_{f,1} + 3\ell_f\ell_\pi\right)M\ell_f}_{\ell_{h_{xy},2}} \left(\|x - x'\| + \|y - y'\|\right). \tag{160}$$

To compute $\ell_\pi$, recall that

$$\nabla_x \pi(x, y) = -\frac{1}{\tau} \nabla F(x)\left(\operatorname{diag}(\pi(x, y)) - \pi(x, y)\pi(x, y)^\top\right), \tag{161}$$

$$\nabla_y \pi(x, y) = \frac{1}{\tau} \nabla F(y)\left(\operatorname{diag}(\pi(x, y)) - \pi(x, y)\pi(x, y)^\top\right) \tag{162}$$

from which we have

$$\max\{\|\nabla_x \pi(x, y)\|, \|\nabla_y \pi(x, y)\|\} \leq \frac{2}{\tau}\sqrt{M}\ell_f := \ell_\pi. \tag{163}$$

Next we bound $\ell_{h_{xx},2}$, and $\ell_{h_{yy},2}$. The Hessian of $h(x, y)$ can be computed by

$$\nabla^2_{yy} h(x, y) = -\frac{1}{\tau} \nabla F(y)\left(\pi(x, y)\pi(x, y)^\top - \operatorname{diag}(\pi(x, y))\right)\nabla F(y)^\top + \sum_{m=1}^M \pi_m(x, y)\nabla^2 f_m(y) + lI_q, \tag{164}$$

$$\nabla^2_{xx} h(x, y) = -\frac{1}{\tau} \nabla F(x)\left(\pi(x, y)\pi(x, y)^\top - \operatorname{diag}(\pi(x, y))\right)\nabla F(x)^\top + \sum_{m=1}^M \pi_m(x, y)\nabla^2 f_m(x) + lI_q. \tag{165}$$

Then we have

$$\|\nabla^2_{yy} h(x, y) - \nabla^2_{yy} h(x', y')\| \leq \frac{1}{\tau}\Big(\|\nabla F(y) - \nabla F(y')\| \|\left(\pi(x, y)\pi(x, y)^\top - \operatorname{diag}(\pi(x, y))\right)\nabla F(y)^\top\|$$

$$
+ \|\nabla F(y')\|\|\pi(x,y) - \pi(x',y')\|(\|\pi(x,y)\| + \|\pi(x',y')\| + 1)\|\nabla F(y)\|
$$

$$
+ \|\nabla F(y')\big(\pi(x',y')\pi(x',y')^\top - \mathrm{diag}(\pi(x',y'))\big)\|\|\nabla F(y) - \nabla F(y')\|\big)
$$

$$
+ \|\nabla^2 F(y) - \nabla^2 F(y')\|\|\pi(x,y)\| + \|\nabla^2 F(y')\|\|\pi(x,y) - \pi(x',y')\|
$$

$$
\leq \underbrace{\frac{1}{\tau}\Big( (2\ell_{f,1} + 3\ell_f\ell_\pi)M\ell_f + \sqrt{M}\ell_{f,2} + \sqrt{M}\ell_{f,1}\ell_\pi \Big)}_{\ell_{h_{yy},2}} \big(\|x - x'\| + \|y - y'\|\big). \tag{166}
$$

With similar derivations as the above, we have $\ell_{h_{xx},2} = \ell_{h_{yy},2}$.

Collecting the results in (155), (157), (159), (160), (163), (166) completes the proof. $\qquad\square$

*Proof of Theorem 3.11.* Since $x_\gamma$ is an $\epsilon$-stationary solution to (PP$_\gamma$), thus

$$
\|\nabla f_0(x_\gamma) + \gamma\nabla p(x_\gamma)\| \leq \epsilon. \tag{167}
$$

By Lemma D.5, it is also an $\epsilon_\gamma$-stationary solution to $\min_{x\in\mathbb{R}^q} p(x)$, and thus

$$
\|\nabla p(x_\gamma)\| \leq \epsilon_\gamma = \frac{\epsilon + \ell_f}{\gamma}. \tag{168}
$$

By Lemma C.14, the KL condition implies that $\varrho_h\big(\mathrm{dist}(x_\gamma, \mathcal{X}_p^* \cap \mathcal{X}_C)\big)^{\eta_h} \leq \|\nabla p(x_\gamma)\|$. And since $\mathcal{X}_p^* \cap \mathcal{X}_C$ is closed, the above implies that there exists $x^* \in \mathcal{X}_p^* \cap \mathcal{X}_C$ such that

$$
\|x_\gamma - x^*\| = \mathrm{dist}(x_\gamma, \mathcal{X}_p^* \cap \mathcal{X}_C) \leq \varrho_h^{-\frac{1}{\eta_h}}\|\nabla p(x_\gamma)\|^{\frac{1}{\eta_h}} = (\varrho_h^{-1}\epsilon_\gamma)^{\frac{1}{\eta_h}} = O(\epsilon_\gamma^{\frac{1}{\eta_h}}). \tag{169}
$$

Taking Taylor expansion of $\nabla p(x)$ at $x^*$ and by the $\ell_{p,2}$-smoothness of $\nabla p(x)$ on $\mathcal{X}_C$, we have

$$
\|\nabla p(x_\gamma) - \nabla^2 p(x^*)(x_\gamma - x^*)\| = \|\nabla p(x_\gamma) - \nabla p(x^*) - \nabla^2 p(x^*)(x_\gamma - x^*)\|
$$

$$
\leq \ell_{p,2}\|x_\gamma - x^*\|^2 \leq \ell_{p,2}(\varrho_h^{-1}\epsilon_\gamma)^{\frac{2}{\eta_h}} = O(\epsilon_\gamma^{\frac{2}{\eta_h}}). \tag{170}
$$

Plugging the above inequality into (167), we have

$$
\|\nabla f_0(x_\gamma) + \gamma\nabla^2 p(x^*)(x_\gamma - x^*)\| \leq \epsilon + \gamma\ell_{p,2}(\varrho_h^{-1}\epsilon_\gamma)^{\frac{2}{\eta_h}} = O(\epsilon + \gamma^{1-\frac{2}{\eta_h}}). \tag{171}
$$

Letting $w = \gamma(x_\gamma - x^*)$, then $\|w\| = O(\gamma^{1-\frac{1}{\eta_h}}) \leq O(1)$ is bounded since $\eta_h \leq 1$. We can further bound $\|\nabla f_0(x_\gamma) + \nabla^2 p(x_\gamma)w\|$ by

$$
\|\nabla f_0(x_\gamma) + \nabla^2 p(x_\gamma)w\| \leq \|\nabla f_0(x_\gamma) + \nabla^2 p(x^*)w\| + \|\nabla^2 p(x_\gamma) - \nabla^2 p(x^*)\|\|w\|
$$

$$
\leq \|\nabla f_0(x_\gamma) + \nabla^2 p(x^*)w\| + \ell_{p,2}\gamma\|x_\gamma - x^*\|^2
$$

$$
\leq \epsilon + 2\gamma\ell_{p,2}(\varrho_h^{-1}\epsilon_\gamma)^{\frac{2}{\eta_h}} = O(\epsilon + \gamma^{1-\frac{2}{\eta_h}}). \tag{172}
$$

Recall that $\eta_h \leq 1$, thus choosing $\gamma = \Omega(\delta^{-1})$, and $\epsilon \leq \ell_f$, we have $\epsilon_\gamma \leq \delta$, and $\|\nabla f_0(x_\gamma) + \nabla^2 p(x_\gamma)w\| \leq \epsilon + \delta$, which proves the result. In this case, since we require

$$
1 \geq \eta_h = \frac{\eta_p}{\alpha_p} = \frac{1}{\alpha_p - 1} \tag{173}
$$

which implies $\alpha_p \geq 2$, and thus $\frac{\theta}{\theta - 1 + \frac{1}{\alpha_v}} \geq 2$ from Proposition D.12, implying $\theta < 2$. $\qquad\square$

*Remark* D.14. Note that, the HEB and KL exponents may not be unique (c.f. Appendix C.2). In such cases, there may exist $0 < \eta_p \neq \frac{\alpha_p}{\alpha_p - 1}$. And the above theorem still requires $\eta_h = \frac{\eta_p}{\alpha_p} \leq 1$, thus $\alpha_p \geq \eta_p$. If we further have that $p(x)$ is smooth around $x^*$, then the smoothness implies $\eta_p \geq 2 \geq \alpha_p$, combining which with $\alpha_p \geq \eta_p$, implies that we require $\alpha_p = \eta_p = 2$. This condition can be relaxed, or the exponent can take a wider range if the local Lipschitz continuity of $f_0$ is replaced by the Hölder continuity with larger exponent as in (Ye et al., 1997, Definition 2.8). We leave a detailed discussion to future work.

# E. Proof of convergence of algorithms

In this section, we prove the convergence of the proposed algorithms with a specific instantiation of $\theta = 1$, thus $p(x) = v_{l,\tau}(x) + \tau \ln M$. For convenience, we define $\varphi_{fh,\gamma}(x,y) := f_0(x) - \gamma h_{l,\tau}(x,y)$.

The details of other oracles including Nesterov's acceleration and Adam updates are given below.

**Nesterov's acceleration.** Define $U(w, \Delta w; \alpha_t, t) = \text{Proj}_{\mathcal{X}}(v_{t+1} + \tilde{\alpha}(v_{t+1} - v_t))$, where $v_{t+1} = w - \alpha_t \Delta w$ and $\tilde{\alpha}$ is the lookahead coefficient.

**Adam updates.** Define $U(w, \Delta w; \alpha, t) = \text{Proj}_{\mathcal{X}}(w - \alpha \frac{\hat{m}_t}{\sqrt{\hat{v}_t} + \epsilon})$, where $m_t = \beta_1 m_{t-1} + (1 - \beta_1)\Delta w$ denotes the moving average of gradients, $v_t = \beta_2 v_{t-1} + (1 - \beta_2)(\Delta w^2)$ denotes moving average of squared gradients, $\hat{m}_t = \frac{m_t}{1 - \beta_1^t}$ is bias-corrected first moment, $\hat{v}_t = \frac{v_t}{1 - \beta_2^t}$ is bias-corrected second moment, with $\beta_1, \beta_2 \in (0, 1)$, $\epsilon$ is the small error.

In our convergence analysis, we use the PGD algorithm as an oracle.

## E.1. Auxiliary lemmas

We first present the auxiliary lemmas to prove the convergence of the algorithms.

**Assumption 4** (Smoothness of functions). For all $m \in \{0, \ldots, M\}$, $f_m$ is $\ell_{f,1}$-smooth on $\mathcal{X}$.

**Assumption 5.** The sequence $\{x_t\}$ generated by Algorithm 1 is bounded on the trajectory.

The above assumption combined with Lemma E.1, implies that $\{f_m(x_t)\}, m = 0, \ldots, M$ are $\ell_f$-Lipschitz on the trajectory.

**Lemma E.1.** *Suppose Assumption 4 holds. Given a bounded set $\mathcal{X}_C \subseteq \mathcal{X}$ such that $\|x\| \le \ell_x$ for all $x \in \mathcal{X}_C$, if there exists $x' \in \mathcal{X}_C$ that $\|\nabla f_m(x')\| \le \bar{\ell}_f$, and $|f_m(x')| < \bar{c}_f$ for all $m = 0, \ldots, M$. Then $f_m(x)$ is bounded and $\ell_f$-Lipschitz continuous for all $x \in \mathcal{X}_C$ with $\ell_f = \bar{\ell}_f + 2\ell_{f,1}\ell_x$.*

*Proof of Lemma E.1.* We first prove that $f_m$ is $\ell_f$-Lipschitz continuous on $\mathcal{X}_C$. For all $x \in \mathcal{X}_C$, it holds that

$$\|\nabla f_m(x)\| \le \|\nabla f_m(x')\| + \|\nabla f_m(x) - \nabla f_m(x')\| \le \bar{\ell}_f + \ell_{f,1}\|x - x'\| \le \underbrace{\bar{\ell}_f + 2\ell_{f,1}\ell_x}_{\ell_f}. \tag{174}$$

Then we can further bound $|f_m(x)|$ by

$$|f_m(x)| \le |f_m(x')| + |f_m(x) - f_m(x')| \le \bar{c}_f + \ell_f\|x - x'\| \le \bar{c}_f + 2\ell_f\ell_x. \tag{175}$$

The above holds for all $m = 0, \ldots, M$, the proof is complete. $\qquad\square$

**Lemma E.2** (Smoothness of $h_{l,\tau}$ and $v_{l,\tau}$). *Under the same settings as Lemma E.1, then $h_{l,\tau}(x,y)$ is $\ell_{h_{l,\tau},1}$-smooth on $\mathcal{X}_C$ w.r.t. both $x$ and $y$ with $\ell_{h_{l,\tau},1} = l + \ell_{f,1} + 2e^{\frac{\ell_f \ell_x}{\tau}} \frac{\ell_f^2}{\tau}$. And $v_{l,\tau}(x)$ is $\ell_{v_{l,\tau},1}$-smooth on $\mathcal{X}_C$ with $\ell_{v_{l,\tau},1} = \ell_{h_{l,\tau},1}(1 + \ell_{y_{l,\tau}^*})$.*

*Proof of Lemma E.2.* The gradient of $h_{l,\tau}(x,y)$ w.r.t. $x$ can be computed by

$$\nabla_x h_{l,\tau}(x,y) = -\sum_{m=1}^{M} \pi_m(x,y)\nabla f_m(x) + l(x - y). \tag{176}$$

Given $x, x', y \in \mathcal{X}$, we can bound $\|\nabla_x h_{l,\tau}(x,y) - \nabla_x h_{l,\tau}(x',y)\|$ by

$$\begin{aligned}
&\|\nabla_x h_{l,\tau}(x,y) - \nabla_x h_{l,\tau}(x',y)\| \\
\le &l\|x - x'\| + \sum_{m=1}^{M} \pi_m(x,y)\|\nabla f_m(x) - \nabla f_m(x')\| + \sum_{m=1}^{M} \big\|\pi_m(x,y) - \pi_m(x',y)\big\|\|\nabla f_m(x')\| \\
\le &l\|x - x'\| + \ell_{f,1}\|x - x'\| + \ell_f \sum_{m=1}^{M} \big\|\pi_m(x,y) - \pi_m(x',y)\big\|
\end{aligned} \tag{177}$$

where we bound $\|\pi_m(x,y) - \pi_m(x',y)\|$ by

$$
\begin{aligned}
\|\pi_m(x,y) - \pi_m(x',y)\| &= \left\| \frac{e^{\frac{f_m(y)-f_m(x)}{\tau}}}{\sum_{m=1}^M e^{\frac{f_m(y)-f_m(x)}{\tau}}} - \frac{e^{\frac{f_m(y)-f_m(x')}{\tau}}}{\sum_{m=1}^M e^{\frac{f_m(y)-f_m(x')}{\tau}}} \right\| \\
&\leq \left\| \frac{e^{\frac{f_m(y)-f_m(x)}{\tau}}}{\sum_{m=1}^M e^{\frac{f_m(y)-f_m(x)}{\tau}}} - \frac{e^{\frac{f_m(y)-f_m(x')}{\tau}}}{\sum_{m=1}^M e^{\frac{f_m(y)-f_m(x)}{\tau}}} \right\| + \left\| \frac{e^{\frac{f_m(y)-f_m(x')}{\tau}}}{\sum_{m=1}^M e^{\frac{f_m(y)-f_m(x)}{\tau}}} - \frac{e^{\frac{f_m(y)-f_m(x')}{\tau}}}{\sum_{m=1}^M e^{\frac{f_m(y)-f_m(x')}{\tau}}} \right\| \\
&\leq \left\| \frac{e^{\frac{f_m(y)-f_m(\tilde{x})}{\tau}}}{\sum_{m=1}^M e^{\frac{f_m(y)-f_m(x)}{\tau}}} \right\| \frac{\ell_f}{\tau}\|x-x'\| + \left\| \frac{e^{\frac{f_m(y)-f_m(x')}{\tau}} \sum_{m=1}^M e^{\frac{f_m(y)-f_m(\tilde{x})}{\tau}}}{\left(\sum_{m=1}^M e^{\frac{f_m(y)-f_m(x)}{\tau}}\right)\left(\sum_{m=1}^M e^{\frac{f_m(y)-f_m(x')}{\tau}}\right)} \right\| \frac{\ell_f}{\tau}\|x-x'\|
\end{aligned}
\tag{178}
$$

where $\tilde{x}$ is on the line segment of $x$ and $x'$. Taking $\sum_{m=1}^M$ of the above inequality yields

$$
\begin{aligned}
&\sum_{m=1}^M \|\pi_m(x,y) - \pi_m(x',y)\| \\
&\leq \frac{\sum_{m=1}^M e^{\frac{f_m(y)-f_m(\tilde{x})}{\tau}}}{\sum_{m=1}^M e^{\frac{f_m(y)-f_m(x)}{\tau}}} \cdot \frac{\ell_f}{\tau}\|x-x'\| + \frac{\sum_{m=1}^M e^{\frac{f_m(y)-f_m(\tilde{x})}{\tau}}}{\sum_{m=1}^M e^{\frac{f_m(y)-f_m(x)}{\tau}}} \cdot \frac{\ell_f}{\tau}\|x-x'\| \\
&\leq 2e^{\frac{\ell_f \ell_x}{\tau}} \frac{\ell_f}{\tau}\|x-x'\|
\end{aligned}
\tag{179}
$$

where the last inequality uses the fact that $\|f_m(\tilde{x}) - f_m(x)\| \leq \ell_f \ell_x$. Combining the above arguments yields

$$
\|\nabla_x h_{l,\tau}(x,y) - \nabla_x h_{l,\tau}(x',y)\| \leq \ell_{h_{l,\tau},1}\|x-x'\|
\tag{180}
$$

with $\ell_{h_{l,\tau},1} = l + \ell_{f,1} + 2e^{\frac{\ell_f \ell_x}{\tau}} \frac{\ell_f^2}{\tau}$.

Similarly, given $x, y, y' \in \mathcal{X}$, we can bound $\|\nabla h_{l,\tau}(x,y) - \nabla h_{l,\tau}(x,y')\|$ by

$$
\|\nabla_x h_{l,\tau}(x,y) - \nabla_x h_{l,\tau}(x,y')\| \leq \ell_{h_{l,\tau},1}\|y-y'\|
\tag{181}
$$

$$
\|\nabla_y h_{l,\tau}(x,y) - \nabla_y h_{l,\tau}(x,y')\| \leq \ell_{h_{l,\tau},1}\|y-y'\|.
\tag{182}
$$

The gradient of $v_{l,\tau}(x)$ can be computed by

$$
\nabla v_{l,\tau}(x) = \sum_{m=1}^M \pi_m(x, y_{l,\tau}^*(x))\nabla f_m(x) + l(x - y_{l,\tau}^*(x)).
\tag{183}
$$

Given $x, x' \in \mathcal{X}$, we can bound $\|\nabla v_{l,\tau}(x) - \nabla v_{l,\tau}(x')\|$ by

$$
\|\nabla v_{l,\tau}(x) - \nabla v_{l,\tau}(x')\| \leq \ell_{h_{l,\tau},1}(\|x-x'\| + \|y_{l,\tau}^*(x) - y_{l,\tau}^*(x')\|) \leq \ell_{h_{l,\tau},1}(1 + \ell_{y_{l,\tau}^*})\|x-x'\|.
\tag{184}
$$

The proof is complete. $\qquad\square$

**Lemma E.3** (Smoothness of the penalized function). *Suppose Assumptions 4, 5 hold. Then $\varphi_{fh,\gamma}$ is $\ell_{\varphi_{fh,\gamma},1}$-smooth w.r.t. $x$ and $y$ on the trajectory, with $\ell_{\varphi_{fh,\gamma},1} = \ell_{f,1} + \gamma\ell_{h_{l,\tau},1}$.*

*Proof of Lemma E.3.* Given $x, x', y, y' \in \mathcal{X}$, we can bound the difference $\|\nabla_x \varphi_{fh,\gamma}(x,y) - \nabla_x \varphi_{fh,\gamma}(x',y)\|$ by

$$
\begin{aligned}
\|\nabla_x \varphi_{fh,\gamma}(x,y) - \nabla_x \varphi_{fh,\gamma}(x',y)\| &\leq \|\nabla f_0(x) - \nabla f_0(x')\| + \gamma\|\nabla h_{l,\tau}(x,y) - \nabla h_{l,\tau}(x',y)\| \\
&\leq \ell_{f,1}\|x-x'\| + \gamma\ell_{h_{l,\tau},1}\|x-x'\|.
\end{aligned}
\tag{185}
$$

Similarly, we can bound the difference $\|\nabla_y \varphi_{fh,\gamma}(x,y) - \nabla_y \varphi_{fh,\gamma}(x,y')\|$ by

$$
\|\nabla_y \varphi_{fh,\gamma}(x,y) - \nabla_y \varphi_{fh,\gamma}(x,y')\| \leq \gamma\|\nabla h_{l,\tau}(x,y) - \nabla h_{l,\tau}(x,y')\| \leq \gamma\ell_{h_{l,\tau},1}\|y-y'\|.
\tag{186}
$$

The proof is complete. $\qquad\square$

**Lemma E.4** (Contraction of $y_{t,k}$). *Suppose Assumptions 4, 5 hold, and $l - \ell_{f,1} \geq \mu_{h_y} > 0$. Recall that $y_{l,\tau}^*(x) := \arg\min_y h_{l,\tau}(x, y)$. The sequence $\{y_{t,k}\}_{k=1}^K$ produced by Algorithm 1 satisfies*

$$\|y_{t,k+1} - y_{l,\tau}^*(x_t)\|^2 \leq (1 - \mu_{h_y}\beta_{t,k})\|y_{t,k} - y_{l,\tau}^*(x_t)\|^2. \tag{187}$$

*Proof of Lemma E.4.* Recall that the update of $y_{t,k}$ in (13a) takes the projected gradient descent (PGD) on $h_{l,\tau}(x, y)$. By Corollary B.4, for $l + \min_{m \in [M]} \mu_m \geq \mu_{h_y} > 0$, the function $h_{l,\tau}(x, y)$ is $\mu_{h_y}$-strongly convex w.r.t. $y$.

Leveraging the convergence result of PGD on strongly convex functions, we have

$$\|y_{t,k+1} - y_{l,\tau}^*(x_t)\|^2 \leq (1 - \mu_{h_y}\beta_{t,k})\|y_{t,k} - y_{l,\tau}^*(x_t)\|^2. \tag{188}$$

The proof is complete. $\qquad\square$

**Corollary E.5.** *Suppose Assumptions 4, 5 hold, and $l - \ell_{f,1} \geq \mu_{h_y} > 0$. Recall that $y_{l,\tau}^*(x) := \arg\min_y h_{l,\tau}(x, y)$. The sequence $\{y_{t,k}\}_{k=1}^K$ produced by Algorithm 1 satisfies*

$$\|y_{t,K} - y_{l,\tau}^*(x_t)\|^2 \leq \prod_{k=0}^{K-1}(1 - \mu_{h_y}\beta_{t,k})\|y_{t,0} - y_{l,\tau}^*(x_t)\|^2. \tag{189}$$

*Proof of Corollary E.5.* The result directly follows from the update of $y_{t,k}$ in (13a), and by applying Lemma E.4 iteratively from $k = 0, \ldots, K - 1$. $\qquad\square$

### E.2. Convergence of the meta algorithm

**Theorem E.6** (Convergence of Algorithm 1 with projected gradient descent). *Suppose Assumptions 4 and 5 hold. The sequence $\{x_t, y_t\}_{t=0}^T$ produced by Algorithm 1 with $\alpha_t = \alpha = \Theta(1)$, $\beta_t = \beta = \Theta(1)$, $\gamma_t = O(1 + t)$, $K_t = O(1 + t)$ satisfies*

$$\frac{1}{T}\sum_{t=0}^{T-1}\frac{1}{\alpha_t^2}\left\|x_t - \text{Proj}_{\mathcal{X}}\left(x_t - \alpha_t\nabla\varphi_{\gamma_t}(x_t)\right)\right\|^2 = O\left(\frac{1}{T}\right). \tag{190}$$

*Proof of Theorem E.6.* We first prove (190), the convergence of the penalty reformulation (PP$_\gamma$). Recall that at each outer-loop iteration, Algorithm 1 does the following update

$$x_{t+1} = \text{Proj}_{\mathcal{X}}\left(x_t - \alpha_t(\nabla f_0(x_t) - \gamma_t\nabla_x h_{l,\tau}(x_t, y_{t+1}))\right) \tag{191}$$

where $y_{t+1} = y_{t,K}$ approximates $y_{l,\tau}^*(x_t)$ with sufficiently large $K$ based on Corollary E.5. Choosing $\beta_{t,k} = \beta_t \leq 1/\mu_{h_y}$ for all $k = 0, \ldots, K - 1$, it then follows that

$$\|y_{t+1} - y_{l,\tau}^*(x_t)\|^2 = \|y_{t,K} - y_{l,\tau}^*(x_t)\|^2 \leq (1 - \mu_{h_y}\beta_t)^K\|y_{t,0} - y_{l,\tau}^*(x_t)\|^2. \tag{192}$$

Let $\ell_{fh,1,t}$ denote the smoothness constant for $f_0(x) - \gamma_t h_{l,\tau}(x, y)$. Define the Lyapunov function $\mathbb{V}_t$ to be

$$\mathbb{V}_t := f_0(x_t) - \gamma_t h_{l,\tau}(x_t, y_{t+1}). \tag{193}$$

Applying the convergence of PGD for general nonconvex smooth objective yields

$$\mathbb{V}_{t+1} - \mathbb{V}_t \leq \langle\nabla f_0(x_t) - \gamma_t\nabla h_{l,\tau}(x_t, y_{t+1}), x_{t+1} - x_t\rangle + \frac{\ell_{fh,1,t}}{2}\|x_{t+1} - x_t\|^2. \tag{194}$$

By the property of projection, and the update of $x_t$, we further have

$$\langle\nabla f_0(x_t) - \gamma_t\nabla h_{l,\tau}(x_t, y_{t+1}), x_{t+1} - x_t\rangle \leq -\frac{1}{\alpha_t}\|x_{t+1} - x_t\|^2 \tag{195}$$

Since $\alpha_t \leq 1/\ell_{fh,1,t}$, plugging the above inequality back into (194) and rearranging yield

$$\mathbb{V}_{t+1} - \mathbb{V}_t \leq -\frac{1}{2\alpha_t}\|x_{t+1} - x_t\|^2 \tag{196}$$

Recall from (191) and (192) that $x_{t+1}$ is an approximation to $\text{Proj}_{\mathcal{X}}\big(x_t - \alpha_t \nabla \varphi_{\gamma_t}(x_t)\big) = \text{Proj}_{\mathcal{X}}\big(x_t - \alpha_t(\nabla f_0(x_t) - \gamma_t \nabla_x h_{l,\tau}(x_t, y^*_{l,\tau}(x_t)))\big)$, since $y_{t+1}$ is an approximation to $y^*_{l,\tau}(x_t)$. Therefore, the term $\big\|x_t - \text{Proj}_{\mathcal{X}}\big(x_t - \alpha_t \nabla \varphi_{\gamma_t}(x_t)\big)\big\|^2$ can be further decomposed as

$$\big\|x_t - \text{Proj}_{\mathcal{X}}\big(x_t - \alpha_t \nabla \varphi_{\gamma_t}(x_t)\big)\big\|^2 = \Big\|x_t - \text{Proj}_{\mathcal{X}}\big(x_t - \alpha_t(\nabla f_0(x_t) - \gamma_t \nabla_x h_{l,\tau}(x_t, y^*_{l,\tau}(x_t)))\big)\Big\|^2$$

$$\leq 2\Big\|x_t - \text{Proj}_{\mathcal{X}}\big(x_t - \alpha_t(\nabla f_0(x_t) - \gamma_t \nabla_x h_{l,\tau}(x_t, y_{t+1}))\big)\Big\|^2 + 2\gamma_t^2\|\nabla_x h_{l,\tau}(x_t, y_{t+1}) - \nabla_x h_{l,\tau}(x_t, y^*_{l,\tau}(x_t))\|^2$$

$$\leq 2\|x_t - x_{t+1}\|^2 + 2\gamma_t^2 \ell_{h_{l,\tau},1}^2 \|y_{t+1} - y^*_{l,\tau}(x_t)\|^2$$

$$\leq 2\|x_t - x_{t+1}\|^2 + 2\gamma_t^2 \ell_{h_{l,\tau},1}^2 \epsilon_{y,t}^2 \tag{197}$$

where $\epsilon_{y,t}^2 = (1 - \mu_{h_y}\beta_t)^{K_t}\|y_{t,0} - y^*_{l,\tau}(x_t)\|^2$ .

Plugging (197) into (196) and rearranging, we have

$$\big\|x_t - \text{Proj}_{\mathcal{X}}\big(x_t - \alpha_t \nabla \varphi_{\gamma_t}(x_t)\big)\big\|^2 \leq 2\|x_t - x_{t+1}\|^2 + 2\gamma_t^2 \ell_{h_{l,\tau},1}^2 \epsilon_{y,t}^2$$

$$\leq 4\alpha_t(\mathbb{V}_t - \mathbb{V}_{t+1}) + 2\gamma_t^2 \ell_{h_{l,\tau},1}^2 \epsilon_{y,t}^2. \tag{198}$$

Taking telescoping sum of the above inequality, and choosing $\alpha_t = \alpha = \Theta(1)$, $\beta_t = \beta = \Theta(1)$, $\gamma_t = O(1 + t)$, $K_t = O(1 + t)$ prove the result. $\qquad\square$

# F. Implementation details and additional experiments

In this section, we report the additional implementation details and additional experimental results and discussion omitted from the main text.

**Computation.** All experiments were conducted on a server with an Intel i9-7920X CPU, and one NVIDIA A5000 GPU. Some experiments require CPU only.

For all the experiments reported in the main text except for the multi-lingual speech recognition experiment, we exactly follow the settings from (Mahapatra & Rajan, 2020). For the multi-lingual speech recognition experiment, we follow the settings from (Chen et al., 2024a). The implementations of the baselines including LS, PMTL, and EPO are from the official code of the EPO (Mahapatra & Rajan, 2020) and that of FERERO (Chen et al., 2024a) is from its official code with their default hyperparameters. The results of XWC-MGDA are directly referenced from the paper.

**Synthetic data.** For the results in both Figure 4 and Figure 6, the model parameter $x$ has dimension $q = 20$, the number of objectives is $M = 2$. The angles between the preference vectors and the horizontal axis are generated between $[\frac{1}{20}\pi, \frac{9}{20}\pi]$ with equal angular distance. The optimization methods are all deterministic in this experiment. We use the default parameters for all baseline methods. For the FOOPS method, we use hyperparameters $\theta = 1, l = 1, \tau = 0.01$. The penalty parameter $\gamma_t = \min\{0.05 + 0.01t, 1.5\}$. The inner-loop parameters are $K = 100, \beta = 0.1$.

*Table 7.* Summary of hyper-parameters for the synthetic data experiments in Figure 6.

| Hyperparameters | LS | MGDA | PMTL | EPO | FERERO | FOOPS (ours) |
|---|---|---|---|---|---|---|
| step size $\alpha_t$ | 0.1 | 0.2 | 0.2 | 0.1 | 0.05 | 0.2 |
| max iterations | 150 | 150 | 150 | 100 | 100 | 100 |

In Figure 6, we include additional results from LS and MGDA for comparison. For all preferences and all methods, the initial model parameter $x_0$ is randomly generated from a Gaussian distribution $\mathbb{B}(0, 1)$ for each dimension. In Table 7, we provide a summary of the hyperparameters for the baselines and our methods for the experiments in Figure 6.

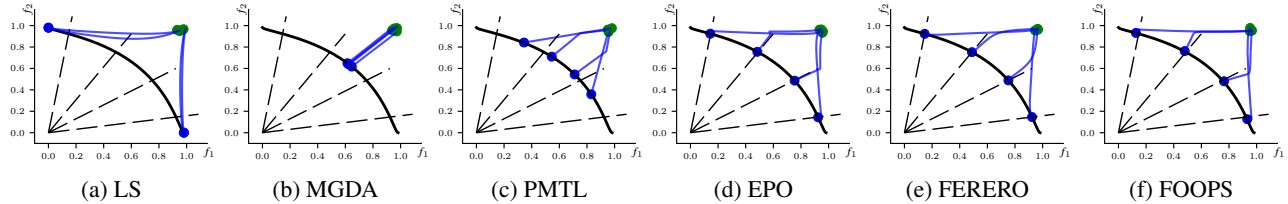

*Figure 6.* Converging solutions (blue dots) and optimization trajectories (blue lines) on the objective space of different methods on synthetic objectives given in (12a). Dashed black arrows represent pre-specified preference vectors. The green dots represent initial objective values.

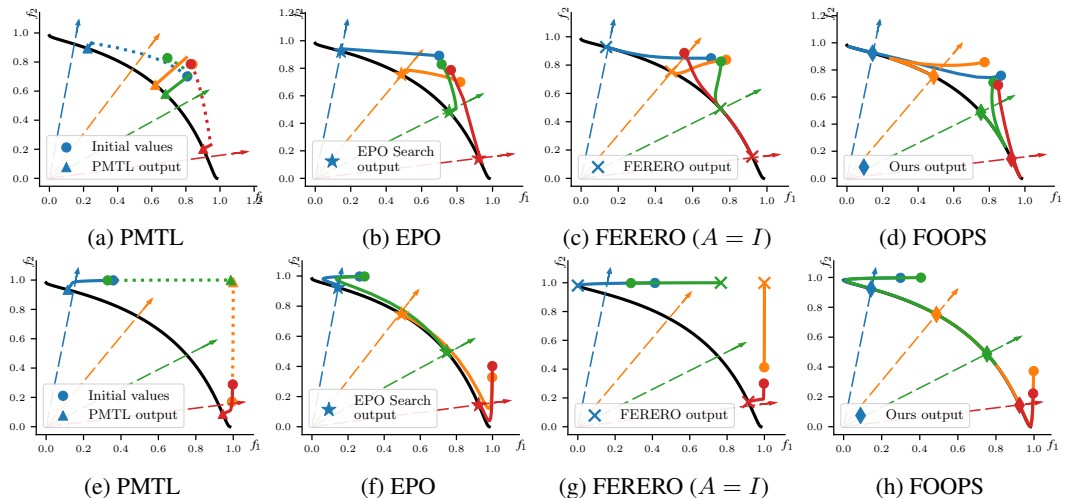

*Figure 7.* Extension of Figure 4. Outputs (colored markers) and optimization trajectories (colored lines) of different methods when initial objectives are near the Pareto front. Different colors represent different preferences. For FERERO, $A = I$ represents choosing the partial order cone $C_A = \mathbb{R}^M_+$ therein for vector optimization, which corresponds to Pareto dominance.

Figure 7 is an extension of Figure 4 under the same experiment objectives but with additional results from EPO and by different initializations. In Figures 7a-7d, the initial model parameters are randomly generated from a uniform distribution between $[-0.3, 0.3]$ for each dimension. While in Figures 4a-4c and Figures 7e-7h, the initial model parameters are randomly generated from a uniform distribution between $[-0.5, -0.15]$ or $[0.15, 0.5]$ for each dimension. Table 8 summarizes the hyperparameters for the experiments in Figure 4. Compared to other baselines, our method is more robust to initializations and requires the least number of iterations for the hard initialization in Figures 7e-7h.

*Table 8.* Summary of hyper-parameters for the synthetic data experiments in Figure 7.

| Hyperparameters | Figures 7a-7d | | | | Figures 7e-7h | | | |
|---|---|---|---|---|---|---|---|---|
| | PMTL | EPO | FERERO | FOOPS | PMTL | EPO | FERERO | FOOPS |
| step size $\alpha_t$ | 0.25 | 0.10 | 0.60 | 0.20 | 0.50 | 0.20 | 0.60 | 0.20 |
| max iterations | 100 | 60 | 10 | 100 | 200 | 120 | 200 | 100 |

**Multi-patch image classification.** For a fair comparison, we follow the same data splitting and processing procedures as (Mahapatra & Rajan, 2020). In each of the three datasets, there are 120k samples for training and 20k samples for testing. There are two tasks on each dataset: 1) classifying the top-left image, and 2) classifying the bottom-right image. For all methods, we use the SGD optimizer with batch size 256. The step sizes for updating the model parameters of all methods are $10^{-3}$. The number of epochs for all methods are 100. The parameters for other methods are chosen as default. For the FOOPS method, we use hyperparameters $\theta = 1, l = 0.6, \tau = 0.01$. The penalty parameter $\gamma_t$ is set initially to 0.1 and increased by 0.1 after every 10 epochs until it reaches 2. The inner-loop parameters are $K = 5, \beta = 10^{-3}$.

We use the Pymoo 0.6.1 library to compute the hypervolume. The Nadir points for the hypervolume computation are given in Table 9. For a fair comparison, the Nadir points we use are the same with (Momma et al., 2022; Chen et al., 2024a).

Table 9. Nadir points for the hypervolume computation

| Dataset and metrics | Nadir points, metrics on objective $[1, \ldots, M]$ |
|---|---|
| Multi-MNIST loss | [0.500, 0.450] |
| Multi-Fashion loss | [0.840, 0.800] |
| Multi-F+M loss | [0.625, 0.575] |
| Multi-MNIST accuracy | [0.830, 0.848] |
| Multi-Fashion accuracy | [0.680, 0.710] |
| Multi-F+M accuracy | [0.790, 0.785] |

Table 10. Hypervolumes $\uparrow \left( \times 10^{-2} \right)$ in multi-patch image classification of different methods including PNG and PB-PDO.

| Datasets | LS | PMTL | EPO | XM | PB-PDO | PNG | FERERO | FOOPS |
|---|---|---|---|---|---|---|---|---|
| Mt-M loss | 1.68 | 1.41 | 1.35 | 1.42 | 1.89 | 1.93 | $1.95 \pm 0.21$ | $\mathbf{2.62 \pm 0.21}$ |
| Mt-F loss | 6.75 | 5.90 | 6.02 | 6.77 | 7.82 | 7.79 | $7.76 \pm 0.18$ | $\mathbf{8.32 \pm 0.37}$ |
| Mt-F+M loss | 3.63 | 3.03 | 3.76 | 3.89 | 3.77 | 3.85 | $3.82 \pm 0.21$ | $\mathbf{4.80 \pm 0.45}$ |
| Mt-M accuracy | 0.19 | 0.15 | 0.15 | 0.16 | 0.23 | 0.25 | $0.25 \pm 0.04$ | $\mathbf{0.33 \pm 0.02}$ |
| Mt-F accuracy | 0.99 | 0.87 | 0.87 | 0.99 | 1.16 | 1.15 | $1.13 \pm 0.07$ | $\mathbf{1.22 \pm 0.07}$ |
| Mt-F+M accuracy | 0.48 | 0.40 | 0.50 | 0.52 | 0.51 | 0.56 | $0.53 \pm 0.04$ | $\mathbf{0.72 \pm 0.06}$ |

**Ablation studies.** In the multi-patch image classification problem, we further conduct ablation studies to test the sensitivity of hypervolumes (HV) for the proposed FOOPS algorithm under different hyperparameters $\tau$ and $l$. Results are plotted in Figure 8 below. They show that choosing $\tau$ or $l$ to be too large or too small could degrade the performance. Nevertheless, the performances of FOOPS under suboptimal choice of the hyperparameters are still better than the baselines.

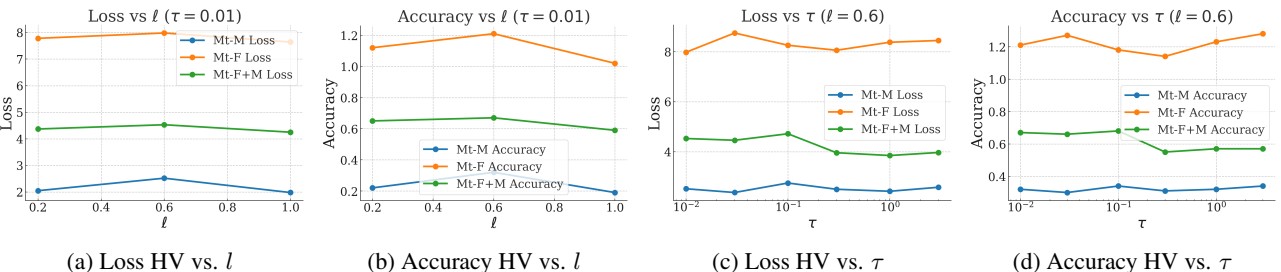

| (a) Loss HV vs. $l$ | (b) Accuracy HV vs. $l$ | (c) Loss HV vs. $\tau$ | (d) Accuracy HV vs. $\tau$ |
|---|---|---|---|

Figure 8. Ablation studies for multi-patch image classification on $l$ and $\tau$.

**Multi-lingual speech recognition.** We follow the same experiment settings in (Chen et al., 2024a). We use two datasets, Librispeech and AISHELL v1. Librispeech is an English speech dataset that consists of 960 hours of labeled audio data. For our experiments, we use the "train-clean-100" subset of the Librispeech dataset for supervised training, which contains 100 hours of clean training data. Additionally, we use the full 960 hours of data for self-supervised training. AISHELL v1 is a 178-hour Mandarin speech corpus designed for various speech and speaker processing tasks. We use the full AISHELL v1 dataset for both self-supervised and supervised training. We combine these two datasets for our multi-lingual speech recognition experiments.

We use the conformer (Gulati et al., 2020) model with 8 conformer blocks as the encoder. Each block contains 512 hidden units and 8 attention heads. Each attention head has dimension 64. The convolutional kernel size is 31. Two classification heads are used. They contain two linear layers, one with 1000 output size for English, and another with 5000 output size for Chinese. The total number of parameters is around 64.5M with 58.4M encoder layer parameters and the rest being the classification layer parameters.

The loss functions we use include the Contrastive Predictive Coding (CPC) loss, and the Connectionist Temporal Classification (CTC) loss. The *CPC loss* (Oord et al., 2018) is a self-supervised loss to learn robust representations from unlabeled speech data. The CPC loss is designed to maximize the probability of a future sample given a contextual representation generated from the current speech sequence. The *CTC loss* is defined as the negative log-likelihood of the model parameter given the input sequence and the label sequence.

For all methods including the baselines, we use the step sizes $\alpha_{t,1} = 5 \times 10^{-4}$ for training the backbone conformer parameters and $\alpha_{t,2} = 5 \times 10^{-5}$ for training the classification head parameters.

**Comparison of run time and memory cost.** In Table 11 we summarize the average run time and number of iterations or epochs of different methods on different datasets. The results show that FOOPS generally requires shorter run time than FERERO, but longer run time than LS. Furthermore, we summarize the memory cost in Table 12. It shows slightly higher memory cost than LS or FERERO on smaller-scale experiments. This is because although FOOPS does not compute $M$ gradients per-iteration while FERERO does, which saves some memory, FOOPS requires storing the model parameters for both $x$ and $y$ while FERERO does not, which introduces extra memory cost. We leave it for future work to further reduce the memory cost and improve the efficiency of the algorithms.

*Table 11.* Summary of average run time in seconds (s), minutes (m), or hours (h) and number of iterations or epochs of different methods on different datasets.

| Datasets | Metrics | LS | PMTL | EPO | FERERO | FOOPS |
|---|---|---|---|---|---|---|
| Synthetic, Figures 3(a-c) | Iterations | 100 | 100 | 60 | 10 | 100 |
| | Per-iteration run time | 3.50E-4s | 7.67E-4s | 4.93E-3s | 7.50E-4s | 7.61E-4s |
| | Total run time | 0.035s | 0.0767s | 0.296s | 0.0075s | 0.0761s |
| Synthetic, Figures 3(d-f) | Iterations | 100 | 200 | 80 | 200 | 100 |
| | Per-iteration run time | 3.10E-4s | 7.65E-4s | 4.93E-3s | 7.30E-4s | 7.43E-4s |
| | Total run time | 0.031s | 0.153s | 0.394s | 0.146s | 0.074s |
| Multi-MNIST/Fashion/F+M | Epochs | 100 | 100 | 100 | 100 | 100 |
| | Per-epoch run time | 3.54s | 11.88s | 9.66s | 7.02s | 6.89s |
| | Total run time | 5.9m | 19.8m | 16.1m | 11.7m | 11.5m |
| Multi-lingual ASR | Finetuning time | 4.2 h | - | - | 13.5 h | 12.7 h |

*Table 12.* Summary of memory consumption and running time of different methods on different datasets. "M" is short for "Megabytes", "G" is short for "Gigabytes".

| Datasets | Metrics | LS | PMTL | EPO | FERERO | FOOPS |
|---|---|---|---|---|---|---|
| Multi-M/F/F+M | GPU memory (LeNet) | 1356 M | 1378 M | 1358 M | 1380 M | 1395 M |
| | GPU memory (ResNet18) | 1738 M | 2386 M | 2060 M | 2196 M | 2224 M |
| Multi-lingual ASR | GPU memory | 38 G | - | - | 40 G | 40 G |

