# OpenReview forum: "Efficient First-Order Optimization on the Pareto Set for Multi-Objective Learning under Preference Guidance"
_ICML.cc/2025/Conference — ICML 2025 spotlightposter_

### Official Review · Reviewer_SqKr · 2025-03-06

**Overall Recommendation:** 3

**Summary:**

This paper considers the problem of preference-guided multi-objective optimization. The authors first formulate it as a semivectorial bilevel optimization problem, which optimizes the upper-level preference objective, subject to the constraint that the model parameters are weakly Pareto optimal or Pareto stationary for the lower-level multi-objective optimization problem. The authors further propose an algorithm to solve this semivectorial bilevel optimization problem, which first converts the lower-level constraint to a single-objective constraint through a merit function, then solves the transformed problem by a penalty-based reformulation. Theoretically, the authors analyze the relations of solutions for different formulations and the convergence of the proposed method. Empirical results on various synthetic and real-world problems demonstrate the effectiveness of the proposed method.

**Claims And Evidence:**

There are two key claims in this work:
- The preference-guided multi-objective optimization problem can be solved through a semivectorial bilevel optimization problem
- The proposed method FOOPS can effectively solved the obtained semivectorial bilevel optimization problem

While the second claim is supported by sound theoretical analysis and empirical results, the first claim seems not sufficiently supported from my perspective. I am a bit confused on how the original formulation for preference-guided multi-objective optimization:

$\min_x F(x), s.t., G(x) \le 0, H(x)=0$

can be re-formulated as

$\min_x f_0(x), s.t., x \in \arg\min F(x)$

I suppose $F(x)$ remains the same, while how are $G(x)$ and $H(x)$ related to $f_0(x)$? Some more explanations are necessary here. Without such explanations, it is even unclear how existing preference-guided multi-objective optimization methods are implemented and if they are solving the same problem.

**Essential References Not Discussed:**

The references are generally complete, and I do not have any works that are strongly recommended to be included.

**Experimental Designs Or Analyses:**

Generally the experiments are sufficient with sound analysis.

**Methods And Evaluation Criteria:**

While the proposed method is easy to understand and the evaluation is sufficient, I am a bit curious on whether the proposed method can be combined with other bi-level optimization methods, as is also mentioned in Appendix A.1. Specifically, after converting the lower-level constraint to a single-objective constraint, I suppose the converted problem can be also solved by some bi-level optimization methods other than the penalty-based formulation? Some empirical comparisons can be useful here.

**Other Comments Or Suggestions:**

The authors seem to have modified margins in some places, e.g., line 110-116 and line 432-439. This should be strictly forbidden and some reasonable explanation is necessary here.

**Other Strengths And Weaknesses:**

All mentioned in previous parts

**Questions For Authors:**

Please see the mentioned points of weakness in the **Claims And Evidence** and **Methods And Evaluation Criteria** part, as well as the possible violation of paper format template.

**Relation To Broader Scientific Literature:**

This paper proposes a novel formulation of preference-guided multi-objective optimization as well as a novel method (based on smoothed merit functions and existing gradient-based bi-level optimization methods) to solve the optimization problem under this formulation.

**Theoretical Claims:**

Theoretical analysis in this work is sound and interesting. I have checked the proofs in Appendix B-E and found they are clearly organized and easy to understand without significant errors.

---

> ### Author Rebuttal · Authors · 2025-03-31
>
> Thanks for acknowledging that **we propose a novel formulation and an easy-to-understand novel method for preference-guided multi-objective optimization, our proof is sound and interesting, and the experiments are sufficient with sound analysis**.
>
> Below we address your concerns point by point. The link to additional results is https://anonymous.4open.science/r/FOOPS-F746/ICML3045_rebuttal.pdf
>
> >**Claims And Evidence:** "The preference-guided multi-objective optimization problem can be solved through a semivectorial bilevel optimization problem" is unclear... How are $G(x)$ and $H(x)$ related to $f_0(x)$? Whether they are solving the same problem?
>
> Yes, $F(x)$ remains the same. The relation of $G,H$ to $f_0$ is that $f_0$ can be chosen as $f_0(x) = ||H(x)||^2 + ||[G(x)]_+||^2$. $f_0$ is minimized when $H(x)=0$ and $G(x)\leq 0$. In our experiments, we show the example for equality-constrained problems without $G$ and with $f_0(x) = ||H(x)||^2$. See Section 6, and Eq. (16) for the speech experiment.
>
> The two formulations are not equivalent mathematically. But both can be applied to preference-guided multi-objective learning with different emphasis on either satisfying preference or achieving weak Pareto optimality. As discussed in Section 3.3 and Section 6 in our paper, the constrained formulation in e.g., PMTL and FERERO with preference modeled by constraints $G$ and $H$ puts more emphasis on satisfying the preference, while the FOOPS formulation with preference modeled by $f_0$ puts more emphasis on achieving weak Pareto optimality.
>
> >**Methods And Evaluation Criteria:** While the proposed method is easy to understand and the evaluation is sufficient, I am a bit curious on whether the proposed method can be combined with other bi-level optimization methods, as is also mentioned in Appendix A.1. Specifically, after converting the lower-level constraint to a single-objective constraint, I suppose the converted problem can be also solved by some bi-level optimization methods other than the penalty-based formulation? Some empirical comparisons can be useful here.
>
> - *Other bilevel methods.*
> Thanks for the suggestion. Indeed, the penalty method used in this paper is not the only way to solve a bilevel problem. However, the existing methods listed in Appendix A.1, Table 5 require **different assumptions** which cannot be satisfied by our converted problem. Therefore, the methods in Table 5 cannot be directly applied to our converted problem.
> However, we provide a discussion on how some other methods could be applied under certain additional assumptions.
> For example, the recent concurrent AGILS (Bai et al., 2024) method can be applied when the merit function $v_{l,\tau}$ has HEB with $\eta \geq 1$.
>
> - *Empirical comparisons.*
> Since other existing bilevel methods require different assumptions, they cannot be directly applied to our converted problem. Further investigation is needed to check whether the methods still work under weaker assumptions. The concurrent AGILS (Bai et al., 2024) method may be applied to specific problems satisfying the assumptions therein, which we leave for future work. We provide a comparison to other OPS methods such as PNG in Figure A and Table B in the link.
>
> >**Other Comments Or Suggestions:** The authors seem to have modified margins in some places, e.g., lines 110-116 and lines 432-439. This should be strictly forbidden and some reasonable explanation is necessary here.
>
> Thanks for spotting this. We would like to clarify that **we did not intentionally or explicitly alter the margins**. Instead, we used the {\small } environment in LaTeX to reduce the size of Equations (5) and (16) so they would better fit the space. This inadvertently caused the line spacing for lines 110-116 and 432-439, immediately before the equations, to appear smaller, despite our attempt to limit the {\small } environment to the equations themselves.
> In response to the reviewer’s feedback, we will modify the paper to remove this unintended spacing issue.
>
> ---
> We hope our rebuttal resolves the reviewer's concerns and the reviewer can reconsider the rating of our paper. Thanks!

---

> > ### Comment · Reviewer_SqKr · 2025-04-09
> >
> > I would like to first thank the authors for their detailed reply. Most of my previous concerns are addressed and the authors are encouraged to add the additional clarification in the revised version, as well as fixing the formatting issue. I have also increased my score.

---

> > > ### Author Response · Authors · 2025-04-09
> > >
> > > Thank you very much for acknowledging our response, engaging in the discussion, and updating your score. Yes, we will incorporate the promised revisions and fix the formatting issue.
> > >
> > > Sincerely, authors.

---

### Official Review · Reviewer_7wXS · 2025-03-06

**Overall Recommendation:** 3

**Summary:**

This paper studies multi-objective optimization with user-specified preferences. The authors formulate the problem as a bilevel optimization problem, where the upper-level is a preference function, and the lower-level problem is the minimization of a smoothed version of merit function. Merit function usually serves as an objective whose solutions are a set of weak Pareto optimal points. The authors provide a comprehensive analysis for the proposed problem. In particular, they first provide some properties of the smoothed merit functions under the assumption that f_m is quasar-convex function. Then, they consider a penalized reformulation of the original bilevel problem. And then they characterize the equivalence between the two problems in terms of global or local solutions, based on the assumption on the global subanalyticity, Holderian error bound and KL inequality. Experiments validate the effectiveness of the proposed method.

**Claims And Evidence:**

Yes.

**Essential References Not Discussed:**

Yes. it includes the most important works on preference-based MOO. However, more related works on differentiable MOO in ML could be provided. For example, PCGrad, CAGrad, NashMTL, MoCo, SDMGrad, where CAGrad and SDMGrad also introduce preference-based regularization or constraints. They should be discussed.

**Experimental Designs Or Analyses:**

A running time comparison and confidence intervals could be provided. More details

**Methods And Evaluation Criteria:**

A running time comparison and confidence intervals could be provided.

**Other Comments Or Suggestions:**

see the strengths and weaknesses.

**Other Strengths And Weaknesses:**

Strengths:

1.	The studied problem is very important in multi-objective optimization, because we often tend to find a preferred point on the Pareto front. The bilevel optimization perspective seems to be new.

2.	The authors analyze the smoothed merit function and the equivalence between the bilevel problem and the panelized problem under some assumptions. Experiments seem to support that the proposed method can get higher accuracy.

Weakness:

1.	The paper is not well written. The analysis part makes multiple assumptions. Some theorem requires the convexity-like assumptions, some needs HEB assumption, and later a KL inequality is needed to guarantee the penalty function is less than \epsilon. I suggest explicitly point out all assumptions in before the theorems.

2.	The motivation of smoothing the merit function is not very clear to me. I understand that the original merit function could be non-differentiable. It may not be a problem in real-world problems. Can the authors validate this in the experiments to see if smoothing is necessary?
3.	Another issue for this smoothing is that it introduces two more hyperparameters \tau and l. Can the authors explain how they select these hyperparameters in practice? Ablation studies should be provided.

4.	The assumptions could be quite strong. The point strong quasar-convexity assumption is hard to be satisfied in practical problems. Although the authors mention some examples such as linear models with leaky ReLU, for general setups, it may be hard to validate this assumption. In addition, the subanalyticity and the KL inequality assumptions further make the analysis less applicable to the practical cases. It would be great if the authors could provide some justification that the problems in the experiments could satisfy these assumptions (it is ok if just partially)?

5.	In lemma 3.4, what does it mean by X_v^* \cap X_C is globally subanalytic. The assumption 2 is made on the function rather than the set. Perhaps a definition on subanalyticity should be provided.

6.	In theorem 3.5, what is the definition of $(\epsilon,\delta)$-global solution for (CP)?

7.	The algorithm is complex, containing multiple hyperparameters like $\tau, l, \gamma_t, K_t, \alpha_t,\beta_t$, as well as two projections in steps 5 and 8. Compared to previous methods like EPO, FERERO, this is less appealing.

8.	Experiments results are not convincing. In table 4, it is very close to FERERO-FT. Then, multiple seeds with confidence interval should be provided. A running time comparison could be provided to further justify the efficiency compared to FERERO.

**Questions For Authors:**

see the strengths and weaknesses.

**Relation To Broader Scientific Literature:**

The idea of using bilevel optimization for preference-based multi-objective optimization is new to the literature.

**Theoretical Claims:**

Check some proofs and did not find main problems.

---

> ### Author Rebuttal · Authors · 2025-04-01
>
> Thanks for acknowledging that the problem is important and the bilevel perspective is new. We would like to emphasize that the bilevel problem in this paper with *non-convex vector-valued lower-level objective* is much more challenging and nontrivial, as pointed out by Reviewer XkmK.
>
> Below we address your concerns. The link to additional results is https://anonymous.4open.science/r/FOOPS-F746/ICML3045_rebuttal.pdf
>
> >W1.Writing&assumptions not clear... Explicitly point out all assumptions before the theorems.
>
> It might be a misunderstanding that the HEB and KL conditions are assumed. Instead of directly assuming them, **we prove them** based on the properties of $F$.
> In our theorems, we list all the required assumptions in the beginning and then discuss other conditions that can be proved. We will clarify this.
>
> >W2.Motivation of smoothing the merit function ...Validate this in the experiments?
>
> Smoothing is commonly used for nonsmooth optimization problems, whose motivation is well-studied in prior works, e.g. [R1] and references therein. [R1] shows that smoothing is necessary for finite-time convergence of nonconvex nonsmooth problems using any algorithm. Therefore, we use LSE to smooth the max-min nonsmooth $\bar{u}$ to ensure finite-time convergence, which is widely used in prior works, e.g. [R2]. In our experiments, we find that without smoothing, it could return NaN (not a number) errors and the algorithm could diverge.
>
> [R1] Deterministic Nonsmooth Nonconvex Optimization. M.I. Jordan, et. al. COLT 2023.
>
> [R2] An alternative softmax operator for reinforcement learning. Kavosh Asadi, et. al. ICML 2017.
>
> >W3.Smoothing introduces hyperparams. \tau and l. How to select them? Provide ablation studies.
>
> We choose $\tau,l$ to be small and to ensure differentiability. We use grid search to tune them. See more details in Appendix F. We provide an ablation study in Table C in the link.
>
> >W4.The assumptions could be strong...The subanalyticity&KL assumptions make the analysis less applicable...Justification of problems in the experiments satisfying these assumptions (ok if partially)?
>
> Note that compared to existing works for OPS (Table 2) and bilevel optimization (Table 5), which usually require convexity or PL assumptions, our assumptions are actually much weaker. The subanalylticity and KL conditions are weaker than PL, see more discussions in Appendix C with examples justifying the assumptions.
>
> >W5.In lemma 3.4, meaning of global subanalyticity?...A definition should be provided.
>
> We have mentioned in the main paper, lines 204-216 that, the definition of (global) subanalyticity, including both subanalytic functions and sets is provided in Appendix C, Definitions C.1 and C.2.
>
> >W6.In theorem 3.5, what is the definition of $(\epsilon,\delta)$-global solution for (CP)?
>
> The $(\epsilon,\delta)$-global solution for (CP) is that, $f_0(x) - \min_{x\in {\cal X}_{\delta}} f_0(x) \leq \epsilon, x\in {\cal X}_{\delta} = \{x \in {\cal X} \mid v_{l,\tau}(x) + \tau\ln M \leq \delta \}.$ This is a widely used definition for constrained optimization.
>
> >W7.The algorithm is complex, containing multiple hyperparameters like $\tau, l, \gamma_t, K_t, \alpha_t,\beta_t$, as well as two projections in steps 5 and 8. Compared to previous methods like EPO, FERERO, this is less appealing.
>
> This might be a misunderstanding. The algorithm is general such that it can be applied to both the cases when $\cal X$ is compact and when ${\cal X} = R^q$.
> When ${\cal X} = R^q$, the two projections are not needed, which is the case in our experiments. $K_t$ can be chosen to be small like 1 or 2. As a comparison, FERERO also requires choosing hyperparameters such as $\alpha_t,\gamma_t,K,c_g,c_h$. So the number of hyperparameters is similar. They do not make FOOPS more complex or inefficient.
>
> We do provide a run time in Appendix F, Table 10 in the paper, and Table A in the link to show FOOPS is comparable or more efficient than FERERO.
>
> >W8&**Methods&Evaluation&Experiment**: Experiments are not convincing. In Table 4, it is very close to FERERO-FT ...confidence interval... A running time comparison could be provided to further justify the efficiency compared to FERERO.
>
> We respectively disagree. We show in Table 3 that FOOPS achieves much better hypervolume compared to FERERO and other baselines. In Table 4, it achieves comparable average performance as FOOPS. So we believe our result demonstrates FOOPS is effective, as *acknowledged by all other reviewers*. This experiment takes much longer time. We will run additional experiments add the confidence in the revision.
>
> Running time comparison is given in Appendix F, Table 10 in the paper, and Table A in the linked PDF. Results show FOOPS can be faster than FERERO.
>
> >**References:** More works on differentiable MOO could be discussed.
>
> Thanks, we will include a discussion in the revision.
>
> ---
> We hope we have addressed your concerns and you can reconsider the rating of our paper. Thanks!

---

> > ### Comment · Reviewer_7wXS · 2025-04-09
> >
> > Apologize for the late response. I thank the authors for the detailed response. My concerns have been resolved and I increase my score accordingly. I highly suggest the authors add the related works and the discussion I mentioned in the final revision.
> >
> > Best,
> > Reviewer

---

> > > ### Author Response · Authors · 2025-04-09
> > >
> > > Thank you very much for acknowledging our response, engaging in the discussion, and updating your score. Yes, we will incorporate the promised revisions and other related works.
> > >
> > > Sincerely, authors.

---

### Official Review · Reviewer_3nrf · 2025-03-10

**Overall Recommendation:** 3

**Summary:**

In this work, the authors frame preference-guided multi-objective learning as an optimization problem on the Pareto set and propose a first-order penalty method to address it, where the penalty function is a polynomial of a smoothed merit function. They begin by establishing key properties of the merit function, including its connection to weak Pareto optimality and the Hölderian error bound. Next, they examine the relationship between solutions of the penalty reformulation and those of the original problem. Finally, they present algorithms for solving the penalty problem and analyze their convergence guarantees.

**Claims And Evidence:**

I find it unclear how the authors incorporate Preference Guidance in this paper. If Preference Guidance were replaced with another objective function, the overall formulation would remain largely unchanged, raising questions about its specific role and impact.

**Essential References Not Discussed:**

I did not find any essential references that are missing.

**Experimental Designs Or Analyses:**

The baselines used in the experimental section are incomplete. I suggest that the authors compare their algorithms with those listed in Table 2 for a more comprehensive evaluation.

**Methods And Evaluation Criteria:**

I believe there is still a missing component in the problem reformulation. The authors only establish the connection between CP and PP, but it would be helpful if they could also provide insights into the relationship between (1) and CP.

**Other Comments Or Suggestions:**

No

**Other Strengths And Weaknesses:**

Strengths:

1. This paper improves the theoretical results of optimization on the Pareto set by removing the strong convexity assumption in (Roy, 2023) and provides stronger convergence guarantees compared to (Ye, 2022).
2. The applications of this type of optimization problem are broad. The paper presents an algorithm to solve such problems, which has the potential to be scaled up to some large-scale applications.

Weaknesses:

1. How should $\tau$ and $l$ be chosen both theoretically and in practice for the merit function?
2. Theorem 3.6, 3.7, and 3.9 appear to be direct adaptations from (Shen, 2023) with only minor modifications.
3. The convergence analysis is relatively trivial, as it primarily leverages results from well-known optimization algorithms.

References:

[1] Roy, A., So, G., and Ma, Y.-A. Optimization on Pareto sets: On a theory of multi-objective optimization. arXiv preprint arXiv:2308.02145, 2023.

[2] Ye, M. and Liu, Q. Pareto navigation gradient descent: a first-order algorithm for optimization in Pareto set. In Uncertainty in Artificial Intelligence, pp. 2246–2255, 2022.

[3] Shen, H., Xiao, Q., and Chen, T. On penalty-based bilevel gradient descent method. arXiv preprint arXiv:2302.05185, 2023.

**Questions For Authors:**

See Theoretical Claims, Experimental Designs Or Analyses, and Other Strengths And Weaknesses.

**Relation To Broader Scientific Literature:**

The experimental results appear promising. After incorporating comparisons with additional baselines, it would be beneficial to scale up the approach and explore its potential applications in larger models.

**Theoretical Claims:**

In Proposition 3.2, where $\epsilon = \tau \ln M$, $\epsilon$ can become quite large as $M$ increases. Would it be better to choose $\tau$ as a function of $M$ rather than a constant, ensuring that $\epsilon$ remains sufficiently small?

---

> ### Author Rebuttal · Authors · 2025-04-01
>
> Thanks for acknowledging that we consider BLO with *non-convex vector-valued lower-level objective* with stronger guarantees, which is different from prior works, and the experiment results are promising.
>
> We would like to emphasize that the *non-convex vector-valued lower-level objective* is much more challenging and nontrivial, as also pointed out by Reviewer XkmK.
>
> Below we address all your concerns. The link to additional results is
> https://anonymous.4open.science/r/FOOPS-F746/ICML3045_rebuttal.pdf
>
> >**Claims:** How to incorporate Preference Guidance...
>
> This might be a misunderstanding. In our paper, the preference guidance was not replaced with another objective from $F$, but was enforced via a new upper-level preference objective $f_0$.
>
> Our experiments include the preference vector guided problems with $f_0(x) = ||H(x)||^2$, and $H(x)=0$ describing the preference vector; see Section 6. See also the response to **Reviewer SqKr-Claims**. We will clarify this.
>
> >**Broader Literature:** ...scale up to applications in larger models.
>
> We have experiments for larger models for speech recognition. See Section 6, Table 4. The model size is around 64.5M, with more details provided in Appendix F.
>
> >**Methods&Evaluation:** Relationship between (1) and CP.
>
> In our paper, below Eq. (CP), we have discussed that as $l,\tau \downarrow 0$, (1) and (CP) are equivalent. For $\tau \downarrow 0, l>0$, (1) and (CP) are equivalent under conditions in Proposition 3.2-2-b). Moreover, by Proposition 3.2, the approximate solutions to (CP) with $v_{l,\tau}(x) + \tau \ln M \leq \epsilon$ are $\epsilon'$-weak Pareto optimal. Therefore, an $\epsilon$-global/local solution $x$ to (CP) satisfies that it is global/local $\epsilon'$-weakly Pareto optimal, and that $f_0(x)\leq f_0(x^*)$, with $x^*$ being the global/local solution to (1). We will add this discussion in the modified paper.
>
> >**Theoretical Claims:** In Proposition 3.2, $\epsilon=\tau\ln M$,... choose $\tau$ as a function of $M$...?
>
> Yes. In our experiments, $M$ is fixed for a problem, and $\tau$ is chosen according to $M$, so we can ensure $\epsilon$ to be sufficiently small.
>
> >**Experimental Designs/Analyses:** ...Compare with methods listed in Table 2.
>
> It is hard to compare with the methods since they do not provide open-source code. Furthermore, these methods require different assumptions that cannot be satisfied in our case. For example, for the lower-level objective, PMM and TAWT require strong convexity or invertible Hessian, which does not hold in our problems. So these methods cannot be applied.
>
> As per the reviewer's request, we implement PB-PDO and PNG. Figure A and Table B in the linked PDF summarize the results, which show they sometimes cannot converge to the Pareto front, and they achieve worse hypervolumes compared to FOOPS.
>
> >W1.How should $\tau$ and $l$ be chosen theoretically and in practice?
>
> As discussed in Section 3.1, theoretically, smaller $\tau$ approximates $\bar{u}$ better, but also increases the smoothness constant. We need to choose $l \geq \ell_{f,1}$ to ensure $\nabla v_{l,\tau}$ exists and can be computed. In practice, we choose relatively smaller $\tau$ and $l$ but not smaller such that $v_{l,\tau}$ is not differentiable. Detailed choices are provided in Appendix F.
>
> >W2.Thm 3.6, 3.7, and 3.9 appear to be direct adaptations from (Shen, 2023)...
>
> We respectively disagree. The theorems compared to (Shen, 2023) are not minor. In (Shen, 2023), the focus is on a scalar lower-level objective satisfying the QG condition (a special case of HEB with $\eta=2$). Directly applying the results from (Shen, 2023) is impossible as we can only make assumptions on objective $F$, but not on $v_{l,\tau}$, and $v_{l,\tau}$ does not satisfy this condition even if $F$ satisfies. In comparison, our theories require much weaker assumptions, as discussed in Appendix A.1, lines 749-756. In addition, (Shen, 2023) assumes the QG holds on the entire domain. However, the exponential function in our case is not globally subanalytic on the whole domain (Remark C.9). Instead, we can prove that $v_{l,\tau}$ is globally subanalytic on any bounded subanalytic set given that the objective $F$ is subanalytic. Thus, the HEB holds on the bounded subanalytic set, which is the base of the proof for Thm 3.6, 3.7, and 3.9.
>
> >W3.Convergence analysis is relatively trivial...
>
> We respectively disagree. Our main contribution is to **convert the challenging semivectorial bilevel optimization problem to a simpler penalty problem** with guarantees on their relations, and easy-to-evaluate gradients for the penalty problem. Building upon this,  the convergence of the penalty problem can be analyzed using existing tools.
>
> By converting challenging problems to simpler ones, we have a simple convergence analysis. This actually shows the *strength rather than weakness* of our method.
>
> ---
> We hope the concerns are addressed and the reviewer can reconsider the rating of our paper. Thanks!

---

> > ### Comment · Reviewer_3nrf · 2025-04-08
> >
> > Thanks to the authors for the rebuttal. I have a follow-up question regarding "Relationship between (1) and CP".
> > I am wondering why you do not have a definition of $\epsilon$-stationary solution for CP. Why is the gradient-based formula (10) needed here? It seems like (10) is only needed for the definition of an $\epsilon$-stationary solution. The relation between CP and (10) is also not clear to me.
> >
> > Since the authors have addressed most of my concerns, I’ve updated my score accordingly.

---

> > > ### Author Response · Authors · 2025-04-09
> > >
> > > Thank you very much for engaging in the discussion and the follow-up questions. Indeed, a proper definition of a stationary condition of CP is a delicate issue in our studied problem and one of our contributions. We will answer your questions point by point as follows.
> > >
> > > **1. Why not use $\epsilon$-stationary solution for CP.** A commonly used stationarity condition is the $(\epsilon,\epsilon')$-KKT condition for the constrained problem CP, which is defined as $v_{l,\tau}(x)\leq \epsilon, ||\nabla f_0(x) + \nabla v_{l,\tau}(x) w || \leq \epsilon'$, with $w$ being a bounded scalar. However, it has been proved in prior works [Liu et al. 2022, Section 4.1; Xiao et al. 2023, Example 1] that for the bilevel problem, the KKT condition of CP is not a necessary optimality condition when the KL exponent of $v_{l,\tau}$ is in a certain range, e.g. exponent $\alpha_v=2$, which reduces to the PL condition. Therefore, it is not suitable to directly use the KKT condition of CP as a stationarity measure.
> > >
> > > **2. The need of formula (10).** The above discussion shows KKT condition for CP cannot be used. Nevertheless, when the lower-level objective satisfies certain conditions such as the KL condition, then the KKT condition of the reformulated problem (10) is a necessary optimality condition. This is proved in our paper in Appendix D.4, Lemma D.7, which shows that the calmness constraint qualification in Definition D.6 holds under the KL condition, justifying that the KKT condition of the reformulated problem (10) is a necessary optimality condition to (10). This type of reformulation exists and has been justified in prior works [Liu et al. 2022, Eq(3); Xiao et al. 2023, Eq(3)], but only for PL lower-level objective. And (10) is equivalent to CP under the KL condition as detailed in the next point.
> > >
> > >
> > > **3. Relation between CP and (10).** (10) is an equivalent reformulation of CP when the lower-level objective $v_{l,\tau}(x)$ or $p(x)$ satisfies the KL condition. This is because under the KL condition, $\nabla p(x)=0$ is equivalent to $p(x)=0$, and thus equivalent to $v_{l,\tau}(x) + \tau\ln M=0$. Similar discussions exist in prior works [Liu et al. 2022, Section 4.1; Xiao et al. 2023, Section 2.1], but only under the PL condition (i.e. exponent $\alpha_p=2$).
> > >
> > >
> > > Due to limited space, a brief discussion is provided below Theorem 3.9, and a detailed discussion is provided in Appendix D.4 in the paper. We will further clarify these questions in the revision. We hope our answers address your questions and that you will reconsider the rating of our paper. We are willing to address any follow-up questions the reviewer may have.
> > >
> > >
> > > ---
> > >
> > > Thank you very much for acknowledging our response, engaging in the discussion, and updating your score. We will incorporate the promised revisions to improve our paper.
> > >
> > > Sincerely, authors
> > >
> > > >References
> > >
> > > >Liu, B., Ye, M., Wright, S., Stone, P., and Liu, Q. "BOME! Bilevel optimization made easy: A simple first-order approach." NeurIPS, 2022.
> > >
> > > >Xiao, Q., Lu, S., and Chen, T. "An alternating optimization method for bilevel problems under the Polyak Łojasiewicz condition." NeurIPS, 2023.

---

### Official Review · Reviewer_XkmK · 2025-03-14

**Overall Recommendation:** 4

**Summary:**

This paper proposes a new method for solving the semivector bilevel optimization problem. The authors first reformulate the multi-objective subproblem as a single objective constraint and then use a penalty-based method to solve the reformulated optimization problem. The results demonstrate the effectiveness of the proposed method in finding preference-guided optimal solutions to the multi-objective problem.

**Claims And Evidence:**

I think all the claims have been well supported by clear and convincing evidence.

**Essential References Not Discussed:**

I think the author should discuss some related work. (See Comments 2)

**Experimental Designs Or Analyses:**

I have checked the experimental designs.

**Methods And Evaluation Criteria:**

I think the proposed methods are well-evaluated.

**Other Comments Or Suggestions:**

1. Lacking analysis on computational cost. The authors should provide the order of computational cost and memory cost per iteration, and provide the table of real running time for each method.
2. In Appendix A, the authors discuss several works on bi-level optimization (BLO). However, I suggest they also provide a discussion on multi-objective bi-level optimization (MOBLO) problems, as explored in [1-5]. The key difference between semivector bi-level optimization and multi-objective bi-level optimization is that the former involves solving multiple objectives at the lower level, making it significantly more challenging.
3. Typo: The author should pay attention to the consistency of the punctuation at the end of the formulas; some formulas end with a period, while others do not.
4. The experiment on the multi-patch image classification problem (using Multi-Fashion+MNIST) is relatively simple and may not fully demonstrate the capabilities of the proposed method. Could the authors conduct experiments on more challenging and widely used datasets, such as Office-31 or Office-Home?
5. It seems the number of objectives in the LL subproblem is small in the experiments. Could the authors conduct experiments on more challenging settings, $M>20$ ?




[1] Gu et al. Min-max multi-objective bilevel optimization with applications in robust machine learning. ICLR, 2023.

[2] Ye et al. AFirst-Order Multi-Gradient Algorithm for Multi-Objective Bi-Level Optimization, ECAI 2024.

[3] Ye et al. Multi-Objective Meta-Learning, AIJ 2024.

[4] Yang et al. Gradient-based algorithms for multi-objective bi-level optimization, Science China Mathematics, 2024

[5] Fernando et al. Mitigating gradient bias in multi-objective learning: A provably convergent approach. ICLR, 2023.

**Other Strengths And Weaknesses:**

Strengths:
1. I think this paper is clearly written and easy to understand.
2. The solved semivector bilevel optimization is very challenging. This paper proposes an efficient method to solve it with a strong theoretical guarantee.
3. The experimental result is intuitive and demonstrates the authors' claims.

**Questions For Authors:**

No other questions.

**Relation To Broader Scientific Literature:**

The key contribution of this paper is proposing a new method for solving the challenging semivector bilevel optimization problem.

**Theoretical Claims:**

I have checked the proof.

---

> ### Author Rebuttal · Authors · 2025-03-31
>
> Thanks for supporting our work, acknowledging that **we propose a new efficient method to solve a very challenging semivectorial bilevel optimization problem,  with a strong theoretical guarantee, and with intuitive experimental result demonstrating the authors' claims**.
>
> We address your other comments below. The link to the additional results is
> https://anonymous.4open.science/r/FOOPS-F746/ICML3045_rebuttal.pdf
>
> >1.Lacking analysis on computational cost. Provide the order of computational cost and memory cost per iteration, and the real running time for each method.
>
> Thanks for the suggestion. In Appendix F, Table 10, we have provided the real running time for each method. We add the memory cost per-iteration in Table A in the link.
>
> >2.In Appendix A, the authors discuss several works on bi-level optimization (BLO). I suggest they also provide a discussion on multi-objective bi-level optimization (MOBLO) problems, as explored in [1-5]...
>
> Thanks for the suggestion. We will incorporate these works in our paper.
>
> >3.Typo: consistency of the punctuation at the end of the formulas.
>
> Thanks for the suggestion. We will check to ensure the consistency of the punctuation.
>
> >4.The experiment on the multi-patch image classification problem (using Multi-Fashion+MNIST) is relatively simple and may not fully demonstrate the capabilities of the proposed method. Could the authors conduct experiments on more challenging and widely used datasets, such as Office-31 or Office-Home?
>
> We have included the experiment for larger models in our paper for speech recognition; see e.g., Table 4 in Section 6. The model size is around 64.5M, much larger than the image classification problem. More details are provided in Appendix F. The benchmarks Office-31 and Office-Home suggested by the reviewer are designed for multi-task learning, but not for preference-guided multi-objective learning studied in this paper, thus not very suitable for our problem. To do such experiments, we need to first define a preference objective $f_0$, then conduct the experiments using our method. We will include the results in the revision.
>
> >5.It seems the number of objectives in the LL subproblem is small in the experiments. Could the authors conduct experiments on more challenging settings, $M>20$?
>
> Theoretically, our method can be used for a larger $M$ without significantly increasing complexity. We test this on a toy problem with $M=25$ and report the running time till convergence in Table A in the linked PDF, the last row, which shows the run time can be much shorter compared to FERERO. The problem is defined as
> $f_m(x) = (x - m)^2, m\in [M]$, $H(x) = f_1(x) - f_2(x)$, and $f_0(x) = ||H(x)||^2$. $x$ is initialized to be $100$. We will find some other real-world problems with more objectives and include the experiments in the revised paper.
>
> ---
> We hope the concerns are addressed and the reviewer can continue to support our paper. Thanks!

---

> > ### Comment · Reviewer_XkmK · 2025-04-04
> >
> > Thanks for the authors' responses. My concerns have been solved, and I will keep my positive score.
> >
> > Here are some other suggestions.
> >
> > In some BLO papers, such as "A Generic Descent Aggregation Framework for Gradient-Based Bi-Level Optimization," they call this kind of BLO a simple BLO where there is only one variable. I wonder whether we can call this problem (Problem 1) semivector bilevel optimization, because naturally, we will have UL variables and LL variables, but the solved problem only has one variable.
> >
> > The authors should write the convergence metric in a separate section and include the discussion of theoretical analysis challenges in the paper.

---

> > > ### Author Response · Authors · 2025-04-07
> > >
> > > Thank you for the follow-up reply and the consistent support!
> > >
> > > Thank you for the suggestions.
> > >
> > > 1. Yes, it would be more accurate to call (Problem 1) a semivector simple bilevel optimization problem. We will revise the paper accordingly.
> > >
> > > 2. We have highlighted some theoretical analysis challenges in the introduction and remarks. Following your suggestion, we will write the convergence metric in a separate section and include a more detailed discussion of theoretical analysis challenges in the main paper.
> > >
> > > Since you are an expert in the field, it would be great if you could champion our paper! Much appreciated!
> > >
> > > Sincerely, Authors

---

### Decision · Program_Chairs · 2025-05-01

**Decision:**

Accept (spotlight poster)

**Comment:**

This paper explores the problem of multi-objective learning under preference guidance. It formulates the problem as a semivectorial bilevel optimization task and proposes a first-order penalty approach to solve it. The paper further establishes the theoretical foundations of the proposed method and demonstrates its practical effectiveness on several machine learning tasks.

The reviewers agree that this paper addresses a challenging and important problem in multi-objective optimization, that the proposed method is novel, and that the theoretical foundation is solid. Given its important contributions to the field, I recommend accepting this paper to ICML.